# Open Vocabulary Compositional Explanations for Neuron Alignment

## Abstract

Compositional explanations leverage logical relationships between concepts to express the spatial alignment between neuron activations and human knowledge. However, these explanations rely on human-annotated datasets, restricting their applicability to specific domains and predefined concepts. This paper addresses this limitation by introducing a framework for the vision domain that allows users to probe neurons for arbitrary concepts and datasets. Specifically, the framework leverages open vocabulary semantic segmentation models to compute open vocabulary compositional explanations. The proposed framework consists of three steps: identifying concept sets, generating semantic segmentation masks using open vocabulary models, and deriving compositional explanations from these masks. The paper also proposes a process that leverages semantic knowledge graphs to analyze and compare compositional explanations computed by different methods sharing the same setup. The paper compares the proposed framework with previous methods for computing compositional explanations both in terms of quantitative metrics and human interpretability, analyzes the differences in explanations when shifting from human-annotated data to model-annotated data, and showcases the additional capabilities provided by the framework in terms of flexibility of the explanations with respect to the tasks and properties of interest.

## 1 Introduction

The black-box nature of deep neural networks (DNNs) remains an important limitation for their adoption in fields, such as healthcare, finance, and autonomous systems, where understanding the rationale behind model behaviors is essential for trust and accountability (Ching et al., 2018). In particular, the opacity of the learning process in DNNs makes it difficult to gain insights into what these models learn and to guarantee the correctness of their behavior. To address this problem, several works have focused on methods to study the knowledge encoded in DNNs and on what individual neurons learn during the training process. Among them, this paper focuses on methods that study the spatial alignment between neuron activations and human knowledge in the vision domain. These methods aim to understand whether artificial neurons encode and parse information similarly to humans. The state-of-the-art in this area is represented by compositional explanations (Mu & Andreas, 2020), which express the alignment between the locations of a given neuron activation range and the location of concepts through propositional logic formulas. For example, ((Cat AND White) OR Dog) can be associated with a neuron whose activations overlap with the locations of white cats or dogs within the images. This approach has been improved over time, including more complex spatial relations (Harth, 2022), knowledge bases (Massidda & Bacciu, 2023), and multiple activation ranges (La Rosa et al., 2023).

Despite progress, one of the main limitations of this family of methods is their dependency on concept-annotated datasets (Ramaswamy et al., 2023; Mu & Andreas, 2020). Specifically, for each concept, these explanations require annotations that identify its precise locations in all samples within the probing dataset. This annotation process is conducted by humans, making it both costly and prone to inconsistencies. On a practical level, only a limited number of concept-annotated datasets are available in the literature. This scarcity imposes several limitations, such as the closed-world assumption, where the model can only be

evaluated on concepts present in these few datasets. Consequently, concepts that are not annotated or concepts with a different level of granularity may be ignored.

**This paper addresses the dependency on human annotations by proposing a framework for the vision domain** that leverages end-to-end open vocabulary semantic segmentation models. These models have recently been proposed to segment *any* object in images, even those not seen during training, by combining traditional segmentation architectures with foundational models (Radford et al., 2021). Specifically, our framework is training-free and relies only on a user-specified list of concepts, without requiring any manual annotations. Based on these concepts, optionally organized into different concept sets, the proposed framework generates segmentation masks by using open vocabulary semantic segmentation models and computes compositional explanations based on the generated masks. This framework offers several advantages, such as enabling explanation generation independent of human-annotated data, supporting explanations at varying levels of granularity, improving explanations through iterative refinements, and compatibility with the open-world assumption, where there are no constraints on the concepts a user can probe the neurons for. Additionally, the paper proposes a process to compare and analyze the differences between compositional explanations computed by different methods sharing the same concept set.

In detail, the paper's contribution is threefold:

- it proposes the first framework that supports open vocabulary compositional explanations in the vision domain. Compared to previous methods, the framework achieves comparable performance on datasets with human annotations, while also offering greater flexibility, and better quantitative and qualitative results on datasets without human annotations.

- it proposes a process to compare explanations computed over the same concept sets by leveraging semantic knowledge graphs. By using this process, it investigates the differences between explanations derived from human and model-annotated data and analyzes the sources of these differences in terms of misalignment and granularity levels.

- it showcases, through two application scenarios, the advantages of the proposed framework in supporting multiple explanation granularity levels and iterative improvement of explanations through refinements.

We will release the code upon acceptance.

## 2 Related Work

**Open Vocabulary Semantic Segmentation** The task of semantic segmentation aims to identify semantic regions in an image based on predefined classes of interest. Open vocabulary semantic segmentation aims to achieve this goal by replacing pre-defined classes with textual descriptions, including ones not encountered during training (Liang et al., 2023b). Existing approaches can be categorized into two main groups: zero-shot segmentation approaches (Bucher et al., 2019; Xian et al., 2019), which typically rely on word embeddings to align image features with unseen classes, and approaches that leverage pre-trained multi-modal models (Radford et al., 2021) to encode both text and images in a shared embedding space and identify the combination of segmented regions and text that maximizes their alignment (Li et al., 2022; Xu et al., 2023c; Liu et al., 2024; Xie et al., 2024). Within the second group, we can further distinguish between two-stage approaches (Liang et al., 2023b; Xu et al., 2022b), which first generate class-agnostic masks and then assign labels to them using multi-modal models, and end-to-end approaches (Cho et al., 2024; Xu et al., 2023c), which integrate multi-modal models earlier in the pipeline to simultaneously identify regions of interest and assign labels. These approaches differ in the placement of the multi-modal model within the pipeline and the training procedures (e.g., alignment losses) used to adapt the models for the segmentation task. Our framework is agnostic to the specific open vocabulary segmentation model employed. However, in this paper, we focus primarily on end-to-end approaches, as they offer greater flexibility in adapting masks to different concept granularity (e.g., whole objects versus object parts).

**Explanations for Neuron Alignment** In this paper, we focus on a specific family of neuron explanations: logic and alignment-based explanations. These explanations aim to find the combination of concepts that maximizes the **alignment between the locations of a given neuron activation range and the locations of those concepts**. These combinations aim to capture a high degree of polysemantic behavior (i.e., the phenomenon where neurons can fire for multiple unrelated concepts (Elhage et al., 2022)). The seminal work in this area is Network Dissection (Bau et al., 2017; 2020), which was later extended by Mu & Andreas (2020) through compositional explanations. These explanations map neuron activations to logical connections between recognized concepts, expressing relationships between them. Prior work has used compositional explanations to analyze the degree of polysemanticity within neuron activations (Mu & Andreas, 2020), show that neurons may recognize different concepts at different activation ranges (La Rosa et al., 2023), study the extent to which convolutional neural networks exploit relative spatial information between objects (Harth, 2022), investigate whether neurons encode specialized or abstract knowledge (Massidda & Bacciu, 2023), and analyze the relationship between representation interpretability and model performance (Mu & Andreas, 2020).

Despite the progress, one of the main limitations of this family of methods is their dependency on concept-annotated datasets (Ramaswamy et al., 2023), limiting their applicability (Mu & Andreas, 2020) and making them costly in terms of human labor. The framework proposed in this paper aims to address the dependency on human annotations of this family of explanations. Our approach is related to Bau et al. (2020), which employs a segmentation model trained on the probing dataset to identify the individual concept (among the ones it has been trained on) that maximizes the overlap between annotations and activations. Differently from them, we leverage open segmentation models, thus removing the requirement to train on the probing dataset, support multiple granularities, and extract logical combinations of concepts. The support for different concept granularity also generalizes the approach proposed in Massidda & Bacciu (2023), which leverages an ontology to infer partial annotations at a higher level of granularity (e.g., from "*cat*" to "*animal*"). In contrast, our framework supports refinements in granularity toward both higher and lower levels.

**Other Neuron and Concept-Based Explanations** While this paper focuses on methods targeting neuron spatial alignment, the literature offers a wide range of explanation approaches that use concepts to derive explanations (Casper et al., 2023; Gilpin et al., 2018) or to decode other types of neurons (Hesse et al., 2025; Srinivas et al., 2025; Bau et al., 2020; O'Mahony et al., 2025; Bykov et al., 2023) and layer behaviors (Gao et al., 2025; Bricken et al., 2023). Some recent studies have also explored the use of foundational models to enable open-vocabulary explanations, for instance within the Concept Bottleneck framework (Oikarinen et al., 2023; Tan et al., 2024) or in approaches that estimate correlations between concept presence and neuron activations (Oikarinen & Weng, 2023). Our work is inspired by this growing interest in open-vocabulary explanations. However, the similarities end there: these approaches belong to distinct families of interpretability methods that differ fundamentally from compositional explanations across multiple dimensions, such as their goal (e.g., alignment vs. correlation), scope (e.g., neuron clusters vs. layers), assumptions (e.g., access to internals and availability of masks), representation (e.g., logical vs. statistical explanations), and phase (post-hoc vs. ante-hoc). Because these families are not able to measure the spatial alignment between activations and the locations of concepts, they have traditionally been regarded as complementary rather than competing categories with respect to compositional explanations. Consequently, they are not compared against them, and no established protocols exist for such comparisons. Given these differences, and because our work specifically focuses on improving and extending compositional explanations without altering their established formulation, we leave the exploration of connections and complementarities with other interpretability paradigms for future research.

## 3 Framework

Let $\mathbb{D} = x_1, x_2, ..., x_n$ be a probing dataset, where each input image $x \in \mathbb{R}^{3,h,w}$ has (variable) height $h$ and width $w$. Let $z$ be a neuron in a probed model. While $z$ is in general arbitrary, in this paper, we consider $z$ as the feature map associated with neurons within convolutional layers. Let $\mathbb{C}$ be a concept set specified by the user, including concepts that may or may not be present in the probing dataset. In vision tasks, a "concept" is the semantic label (e.g., "wheel") identifying a group of pixels that are semantically related (e.g., pixels

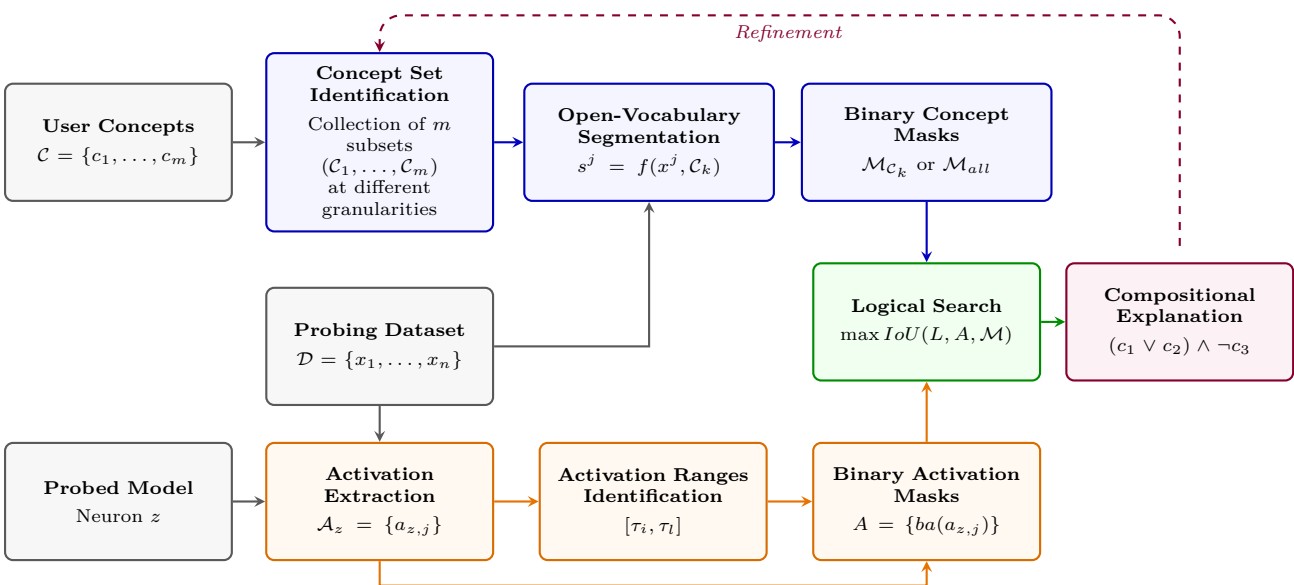

Figure 1: Pipeline of the proposed open-vocabulary compositional explanation framework. Users specify multiple concept subsets representing different levels of granularity. The framework generates concept masks through open-vocabulary semantic segmentation and identifies meaningful neuron activation ranges from the probed model. The compositional explanation algorithm searches for the logical formula that maximizes the alignment between binarized neuron activations and combinations of concept masks. After inspecting the resulting explanation, users can optionally refine the concept subsets and regenerate the masks associated with the affected concept subset.

corresponding to a wheel). Let $\mathfrak{L}^n$ be the set of all possible logical connections of arity at most $n$ between concepts in the concept set $\mathbb{C}$, where concepts are chained by propositional logic connectives. Compositional explanations aim to assign to $z$ the logical combination $L \in \mathfrak{L}^n$ of concepts in $\mathbb{C}$ (e.g., ((Cat OR Dog) AND Brown)) that maximizes the alignment between the localization of a given neuron's activation range and the localization of combinations of concepts within the probing dataset. The activation range can be extracted either using a quantile (Mu & Andreas, 2020) (i.e., highest activations) or using clustering (La Rosa et al., 2023).

The goal of our framework is to generate compositional explanations without requiring humans to manually annotate the location of every concept in the probing dataset while offering more flexibility to the user. We can distinguish three steps: identifying the concept set, generating segmentation masks, and generating compositional explanations.

**Concept Set Identification.** In our framework, the concept set $\mathbb{C}$ corresponds to a collection of $m$ concept subsets

$$\mathbb{C} = \{C_1, \ldots, C_m\} \tag{1}$$

where each subset $C_k$ consists of a list of $n_k$ concepts

$$C_k = \{c_{k,1}, \ldots, c_{k,n_k}\}, \ \forall k \in \{1, \ldots, m\} \tag{2}$$

$$\text{subject to} \quad C_i \cap C_j = \emptyset, \ \forall i \neq j \tag{3}$$

Equation (3) represents the constraint that the concept sets do not share concept names, and it is necessary to avoid inconsistency in mask generation. The concepts are arbitrary and specified by the users. Each concept subset can be used to describe different levels of concept granularity. For example, if $\mathbb{C}$ includes concepts related to cars, its subsets may be identified as $\mathbb{C} = \{brand, color, parts\}$ and each subset $C_k$ includes concepts related to each granularity (e.g., $color = \{yellow, white, black\}$).

**Masks Generation.** Given the probing dataset $\mathbb{D}$, a pretrained open vocabulary segmentation model $f(\cdot,\cdot)$, and a concept subset $C_k \in \mathbb{C}$, the framework generates a set of segmentation masks

$$S_k = \{ s^j, \forall j \in \mathbb{D} : s^j = f(x^j, C_k)\} \tag{4}$$

where each element in $s^j$ corresponds to the concept most likely represented by the pixel at the same position in $x^j$ (e.g., $s^0 = [[white, white, ..., blue], ..., [orange, ..., black]]$. The specific operations performed by the function $f(\cdot,\cdot)$ depend on the implementation of the open vocabulary segmentation model. Our framework is agnostic with respect to this implementation. The only assumption is that $f(\cdot,\cdot)$ can assign an arbitrary specified concept to each pixel.

To satisfy the requirement of the compositional explanation algorithm (Mu & Andreas, 2020), these masks are upsampled (or downsampled) to have the same dimensions. Each segmentation mask $s^j$ is then transformed into a set of binarized masks $M_{C_k}^j$, one for each concept $c \in C_k$:

$$M_{C_k}^j = \{b_s(s^j, c_{k,i}), \forall c_{k,i} \in C_k\} \tag{5}$$

where $b_s(s^j, c_{k,i})$ is a function that returns a binary mask where the pixels assigned to the concept $c_{k,i}$ are set to 1, and the others are set to 0 (e.g., $b_s(s^0, white) = [1, 1, ..., 0], ..., [0, ..., 0]]$). For each concept subset, the binarized masks are then grouped into a **single-granularity** binary mask set:

$$\mathbb{M}_{C_k} = \{M_{C_k}^j, \forall j \in \mathbb{D}\} \tag{6}$$

By aggregating the single-granularity sets for all of the desired granularities, we can obtain the **multi-granularity** binary masks set:

$$\mathbb{M}_{all} = \{M_{C_k}, \forall C_k \in \mathbb{C}\} \tag{7}$$

**Alignment Computation.** The first step to compute the alignment for a neuron $t$ is to collect its activations $A_t$ over the probing dataset:

$$A_t = \{a_{t,j}, \forall j \in \mathbb{D}\} \tag{8}$$

The shape of $a_{t,j}$ depends on the neuron type. In general, this shape differs from that of the input and segmentation masks, and an additional function is needed to project the activation into the proper annotation space. Our framework is agnostic to the specific projection. In this paper, we follow the established literature on the topic (Bau et al., 2020; Mu & Andreas, 2020) by considering bidimensional neurons in the convolutional layers (e.g., $a_{t,j} = [[2.3, 0.0, ..., 0.7], ..., [0.9, 2.0, ..., 0.0]]$ and using bilinear interpolation to reshape the activations. Given $A_t$ we apply K-Means to identify distinct activation ranges (e.g., $[\tau_1, \tau_2] = [0.5, 1.2]$) that correspond to different semantic behaviors of the neuron (La Rosa et al., 2023). However, the framework is independent of the specific technique used to extract the activation ranges, and other options are supported (e.g., quantile (Mu & Andreas, 2020)). Then, for each activation range $[\tau_i, \tau_l]$, the framework computes the binarized activations $\mathbb{A}$ as:

$$\mathbb{A} = \{b_a(a_{t,j}, [\tau_i, \tau_l]), \forall j \in \mathbb{D}\} \tag{9}$$

where $b_a(a_{t,j}, [\tau_i, \tau_l])$ is a function that sets to 1 all activation values within the specified range and to 0 otherwise (e.g., $b_a(a_{t,j}, [\tau_1, \tau_2]) = [[0, 0, ...., 1], ..., [1, 0, ..., 0]]$.

Finally, for each of the identified clusters, the framework computes compositional explanations by finding the concepts that maximize the alignment between the binarized masks $\mathbb{A}$ and the concepts' segmentation masks in $\mathbb{M}$. To compute these explanations, we apply beam search guided by the MMESH spatial heuristic (La Rosa et al., 2023) (see Section C for more details). Formally, the beam search identifies the label $L \in \mathfrak{L}^n$ that maximizes the following objective:

$$\underset{L \in \mathfrak{L}^n}{\arg\max} \, IoU(L, \mathbb{A}, \mathbb{M}) \tag{10}$$

where the Intersection Over Union ($IoU$) measures the overlap between label annotations and neuron activations, and it is defined as:

$$IoU(L, \mathbb{A}, \mathbb{M}) = \frac{|^1\mathbb{A} \, \cap \, ^1\theta(\mathbb{M}, L)|}{|^1\mathbb{A} \, \cup \, ^1\theta(\mathbb{M}, L)|} \tag{11}$$

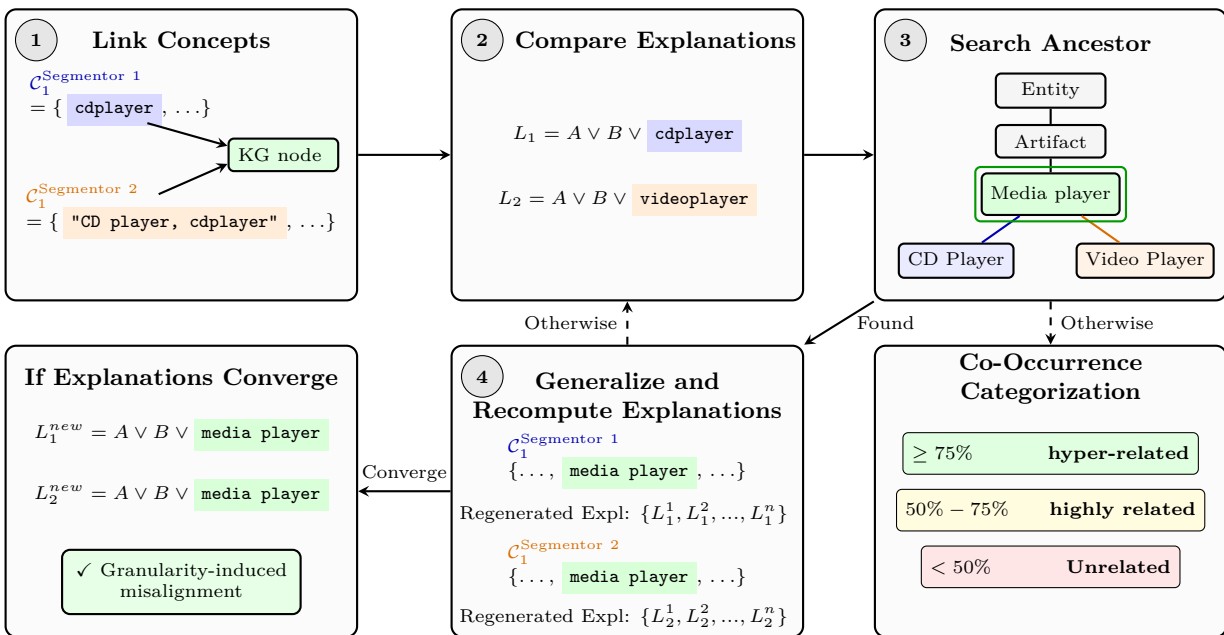

Figure 2: Illustration of knowledge-graph guided alignment analysis. Concept sets from different segmentors are first linked to semantic knowledge-graph nodes. Then, given two explanations, the process extracts the differing concepts and searches for their lowest common ancestor in the knowledge graph. If the ancestor is found, the concept sets are then updated, masks are regenerated, and explanations are recomputed. If the resulting explanations converge, the disagreement is classified as a granularity-induced misalignment. If an ancestor is not found, the disagreement is categorized according to concept co-occurrence.

where the notation $^1\cdot$ indicates the set of locations equal to 1 in a binary matrix, and $\theta(\mathbb{M}, L)$ is a function that returns the logical combination of the masks in $\mathbb{M}$ of the concepts involved in the label $L$. Following Mu & Andreas (2020), we consider AND, OR, and AND NOT as logical connectives, computed by standard bitwise logical operators between the binary matrices in $\mathbb{M}$. Setting $\mathbb{M} = \mathbb{M}_{C_k}$ in Equation (11) results in single-granularity compositional explanations, equivalent to those computed in previous work, but based on model annotations instead of human ones. Conversely, setting $\mathbb{M} = \mathbb{M}_{all}$ enables the usage of concepts from all of the granularities. In this case, the algorithm **automatically selects the granularity level that is most aligned with each neuron**.

After inspecting the compositional explanations computed in this step, the user can **optionally refine** the concept set by adding or removing concepts of interest, thus providing more flexibility during the analysis. Since the framework treats the concept subsets as independent, it regenerates the masks only for the specific subsets affected by the refinement (i.e., those to which the concepts are added or removed).

## 4 Knowledge-Graph Guided Alignment Analysis

This section describes the multi-step process we propose to analyze and characterize the misalignment between explanations computed by different methods. This process leverages a semantic knowledge graph (KG) that supports the hypernym relationship (i.e., "cat" is a "feline"). We assume each node in the graph represents a word sense (e.g., cat as an animal), and that each word sense is associated with one or multiple lemmas (e.g., $lemmas(cat) = \{true\ cat, domestic\ cat, puss, house\ cat\}$). Because the hypernym relationship is unidirectional, it defines a tree in the KG. The proposed process leverages this tree to analyze explanation differences. In this paper, we use WordNet (Miller, 1995) as the underlying KG because it is publicly available and several computer vision datasets already include information related to it. However, the proposed process is not restricted to WordNet and can also leverage alternative semantic knowledge graphs, such as BabelNet (Navigli et al., 2021), or task-specific ontologies tailored to the analyzed domain.

Given two explanations $L_1$ and $L_2$ computed from two different sets of annotations, the goal of this process is to establish whether there is a difference (**misalignment**) between them and characterize the origin of such misalignment (e.g., due to differences in granularity, such as "coca-cola" vs "soft drink"). At a high level, as shown in Figure 2, the process identifies the nodes in the KG corresponding to the concepts in the explanations and verifies whether there is a higher-level concept that could generalize both of them. If such a generalization is found, the concept set is modified by replacing the original concepts with their common generalization, a new set of annotations is generated, and the explanations are recomputed. If the resulting explanations converge (i.e., $L_1^{new} = L_2^{new}$), the disagreement is attributed to a difference in granularity. Otherwise, the disagreement is characterized based on the co-occurrence of the differing concepts. More in detail, the process consists of the following steps:

**Step 1: Mapping each Concept to a Node in the KG**  The goal of this step is to associate each concept in $\mathbb{C}$ with one and only one node in the KG. To compute this association, we propose a semi-automatic process that exploits synonyms of each concept. The list of synonyms can be retrieved from the dataset, if available, or it can be computed by retrieving synonyms from online repositories (Miller, 1995; Navigli et al., 2021) or by leveraging large language models and exploiting the contextual information (e.g., the dataset domain). Given a concept, we select the node whose lemmas have the maximum overlap with the list of concept's synonyms. For concepts without available synonyms, we extract the most common sense in the KG (i.e., the first sense returned by WordNet). In the case where this information is not available in the KG, one may select the most common sense based on the frequency of the sense in a sense-annotated corpus (Scarlini et al., 2020). The output of this step is a complete one-to-one mapping between concepts in $\mathbb{C}$ and nodes in the KG.

**Step 2: Extracting the Explanation Differences**  This step focuses on identifying differences between $L_1$ and $L_2$. This step deals with two tasks: identifying differing concepts and accounting for the logical meaning induced by logical operations. When both explanations are derived from the same concept set, identifying differing concepts is straightforward. When this is not the case, we rely on the synonyms of each concept to identify equivalences. In this case, two concepts are considered equivalent if they share at least one synonym (e.g., the concept "road" is equivalent to "road, route"). For dealing with the logical meaning, we consider two explanations equivalent if they satisfy logical equivalences (e.g., A OR B is equivalent to B OR A). In computing such equivalences, we ignore the negative side of explanations, such as the concept C in the explanation (A AND NOT C) since the granularity analysis focuses on the positively recognized concept (i.e., the generalization of "cats that are not white" is still "cats", independently of the concept being negated). The output of this step is the list of differing concepts between $L_1$ and $L_2$.

**Step 3: Identifying a Meaningful Ancestor**  For each differing concept identified in Step 2, we use the mapping generated in Step 1 to identify its corresponding node in the graph. Then, we search for the lowest (in the hierarchy defined by the hypernym tree) meaningful common ancestor between that concept and any concept in the explanation retrieved by the alternative method. The ancestor is identified by tracing the path of hypernyms from the concept's node to the root of the hypernym tree. Although any two nodes in the tree always share at least one common ancestor (i.e., the root node), ancestors located high in the hierarchy may be too abstract to provide meaningful insight (e.g., *entity, thing, abstraction*). To address this, we consider an ancestor "found" only if it is not one of the highest-level nodes in the hypernym tree. Because the notion of meaningful abstraction depends on the analysis objective and the ontology being used, the specific excluded nodes are ontology-dependent. For WordNet, we report the exact excluded nodes and rationale in Section E. The output of this step is a mapping between each differing concept and its generalization, if any.

**Step 4: Generating new Concept Sets, Annotations, and Explanations**  Given the mapping generated at the end of step 3, we revisit all the concepts in the concept set, replace them with their identified generalizations (if applicable), and generate a new concept set based on this updated mapping. Once the new concept set is defined, we generate updated segmentation masks using it, and we recompute the explanations based on the updated segmentation masks.

At the end of Step 4, if the updated explanations are different, the process restarts from Step 2, searching for differences in the updated explanations and then for additional common ancestors. This iterative process (from Step 2 to Step 4) continues until no further generalization can be identified between different explanations or the explanations converge.

**Misalignment Characterization:** Once the process terminates, the explanations can be compared before and after the generalization procedure. Explanations that become equivalent after the replacement of concepts with their identified ancestors are considered **misaligned due to differences in granularity**. Consequently, the reduction in the number of differing concepts provides a measure of the extent to which granularity contributes to explanation disagreement. Among the remaining misaligned concepts, we distinguish three categories: **hyper-related** concepts that co-occur in more than 75% of the samples activating the explanation, **highly related** concepts with co-occurrence above 50%, **unrelated** concepts with low or no co-occurrence. These categories help characterize the residual disagreement and support the analysis of differences between alternative annotation sources (e.g., human annotations and open-vocabulary segmentation models).

## 5 Experiments

This section introduces the experimental setup (Section 5.1), evaluates the proposed framework (Section 5.2), and analyzes the difference between compositional explanations computed over human annotations and by our framework (Section 5.3).

### 5.1 Setup

In the following sections, we use CAT-Seg (Cho et al., 2024) with its default parameters (Section J) as the backbone open vocabulary segmentation model of our framework. We follow prior work and consider the following approaches for computing spatial alignment: the heuristic-guided approach proposed in La Rosa et al. (2023) (*Human*), which relies on human-annotated data, and the closed vocabulary approach proposed in Bau et al. (2020). The term "closed vocabulary" refers to segmentation models trained on a specific dataset and able to recognize only concepts included in that dataset. Unlike our framework, the user cannot specify the concepts of interest and this baseline will generate segmentation masks related exclusively to the concept dataset used during the training stage. For the closed vocabulary approach, we update their proposal by extending it to the compositional explanation case and replacing their model with a state-of-the-art segmentation model (Mask2Former (Cheng et al., 2022)) trained on COCO (Lin et al., 2014). We do not include SAE or other open-vocabulary explanation methods (Section 2) as competitors, as they pursue different goals and are not designed to capture localization alignment. Evaluating them fairly would require substantial adaptations beyond the scope of this work.

All competitors share the same experimental settings and hyperparameters, selected as the best found by prior work (see Section J). Namely, we focus on the neurons of the last convolutional layer of the probed models, we set the maximum explanation length to 3, and the beam size to 5 as in Mu & Andreas (2020).

### 5.2 Quantitative and Qualitative Evaluation

#### 5.2.1 Quantitative Evaluation

The first set of experiments evaluates our proposed framework by measuring the quality of its generated compositional explanations by using the per-pixel metrics adopted by previous literature and a user study. Specifically, we employ $IoU$, as defined in Equation (11); Detection Accuracy (Makinwa et al., 2022) ($DetAcc$), which quantifies the percentage of label annotations recognized within the activation range and defined as $DetAcc(L, \mathbb{A}, \mathbb{M}) = \frac{|\mathbb{A} \cap \theta(\mathbb{M}, L)|}{|\theta(\mathbb{M}, L)|}$; and Activation Coverage (La Rosa et al., 2023) ($ActCov$), which measures the percentage of neuron activations within the annotated label regions and defined as $ActCov(L, \mathbb{A}, \mathbb{M}) = \frac{|\mathbb{A} \cap \theta(\mathbb{M}, L)|}{|\mathbb{A}|}$. Additional quantitative metrics and their results, as well as further details about these metrics, can be found in Section A.3.

Table 1: Avg. scores for compositional explanations computed by the competitors for a model trained on the Place365 dataset probed using multiple datasets.

| Method | Metrics | | |
|---|---|---|---|
| | IoU | ActCov | DetAcc |
| Ade20k-150 | | | |
| Human (La Rosa et al., 2023) | 0.083 | 0.197 | 0.129 |
| Closed (Bau et al., 2020) | 0.071 | 0.211 | 0.120 |
| Ours | 0.093 | 0.214 | 0.178 |
| Ade20K-Extended (847 classes) | | | |
| Human (La Rosa et al., 2023) | 0.098 | 0.196 | 0.242 |
| Closed (Bau et al., 2020) | 0.071 | 0.221 | 0.119 |
| Ours | 0.103 | 0.208 | 0.236 |
| Mapillary Vistas | | | |
| Human (La Rosa et al., 2023) | 0.053 | 0.266 | 0.080 |
| Closed (Bau et al., 2020) | 0.050 | 0.254 | 0.082 |
| Ours | 0.052 | 0.271 | 0.078 |
| Cityscapes | | | |
| Human (La Rosa et al., 2023) | 0.050 | 0.308 | 0.068 |
| Closed (Bau et al., 2020) | 0.048 | 0.277 | 0.068 |
| Ours | 0.045 | 0.342 | 0.060 |
| Pascal-Context | | | |
| Human (La Rosa et al., 2023) | 0.077 | 0.280 | 0.103 |
| Closed (Bau et al., 2020) | 0.077 | 0.301 | 0.102 |
| Ours | 0.079 | 0.286 | 0.109 |
| Coco-Stuff | | | |
| Human (La Rosa et al., 2023) | 0.078 | 0.197 | 0.127 |
| Closed (Bau et al., 2020) | 0.078 | 0.225 | 0.120 |
| Ours | 0.080 | 0.221 | 0.123 |
| VOC 2012 | | | |
| Human (La Rosa et al., 2023) | 0.077 | 0.247 | 0.106 |
| Closed (Bau et al., 2020) | 0.084 | 0.296 | 0.114 |
| Ours | 0.051 | 0.357 | 0.058 |

We begin our analysis by comparing quantitatively compositional explanations for 50 randomly selected neurons in a ResNet18 (He et al., 2016) model trained on Place365 (Zhou et al., 2017a). In this section, we focus on the highest activations, and therefore we report the results only for the highest cluster, identified as explained in Section 3. However, similar results are observed across all the clusters and are reported in Section A.1. As probing datasets, we use the validation split of the following datasets: Mapillary Vistas (Neuhold et al., 2017), Cityscapes (Cordts et al., 2016), Pascal VOC (Everingham et al., 2012), PASCAL-Context-459 (Mottaghi et al., 2014), Ade20k in both its 150 and 847 classes versions (Zhou et al., 2017b), and COCO-Stuff (Caesar et al., 2018). This setup includes datasets commonly used in the literature to evaluate compositional explanations or segmentation models, for which human annotations are available, that include concepts relevant to the task learned by the probed model (Bau et al., 2020), and that cover a wide range of concept-set complexities (ranging from 19 concepts in Cityscapes to 847 concepts in Ade20K in its full version). We use human annotations as masks for the "*human*" baseline and their labels as a concept set for our framework. The goal of this experiment is to investigate whether there is a *degradation* in explanation quality when transitioning from human-annotated data to model-annotated data. This potential degradation could arise due to imprecision in the segmentation masks returned by the models or errors in the masks' labeling process. As shown in Table 1, our framework achieves scores comparable to those of the competitors

Table 2: Avg. scores for compositional explanations computed by the competitors for a model trained on CUB.

| Method | IoU | ActCov | DetAcc |
|---|---|---|---|
| Human (La Rosa et al., 2023) | N/A | N/A | N/A |
| Human (La Rosa et al., 2023)$_{Ade20k}$ | 0.052 | 0.144 | 0.100 |
| Closed (Bau et al., 2020) | 0.029 | 0.674 | 0.033 |
| Ours | 0.077 | 0.188 | 0.131 |

(with std. dev. reported in Section A.1) across all datasets except VOC2012, where we observe a slight degradation in alignment. Consequently, we consider the results in these settings satisfactory, and **we do not observe significant degradation in explanation quality when using model-annotated data to compute compositional explanations**. Although the "*human*" baseline is applicable when the dataset includes human annotations, our framework remains useful in these scenarios for generating explanations at a different granularity and providing a deeper and more flexible interpretation.

Table 2 shows the results for 2048 neurons in a ResNet50 model (Song et al., 2021) trained on CUB (Wah et al., 2011) for bird species classification and using the validation split of CUB as a probing dataset. **This setting represents the task our framework is targeting**: we consider the case where no human-annotated relevant masks are available[1]. For our framework, we identify a multi-granularity concept set obtained through refinements and task-specific information. Namely, we use as concepts bird species, colors, shapes, patterns associated with bird parts, the background class, and a class "other" to provide the segmentation model with default choices. This information is available in the dataset as classes or additional attribution information. As granularities, we identify three levels: the first includes bird species, bird shapes, and colors; the second includes parts; and the third includes all remaining subsets. We also include the set of concepts annotated in the Ade20K dataset as an additional subset. For a more detailed discussion about the process used to identify the concept set for CUB, we invite the reader to refer to Section D. Because there are no human-annotated data, the "*human*" baseline could not be applied, and our framework aims to address this limitation. However, one could alternatively attempt to probe the model using a different dataset where annotations are available. To explore this, we consider using Ade20K as a probing dataset for their approach (Human$_{Ade20k}$). While this provides a point of comparison, we argue this strategy is not optimal and should be avoided due to several drawbacks (e.g., hallucinations and concept misalignment). In this case, **our framework represents a significantly better choice** than alternatives in terms of IoU and DetAcc. A qualitative analysis reveals even more significant differences. For the "*human*" baseline, compositional explanations are often computed over hallucinations of the probed model when parsing objects in Ade20K not available in CUB (i.e., the dataset used to train the probed model), leading to artifact alignments. This issue is evident when inspecting the most aligned concepts, where we observe hallucinated concepts such as "pool table" (IoU=0.328) and "car" (IoU=0.22), which are absent and not relevant in CUB. These findings confirm the limitations of the human-based approaches when applied to datasets lacking annotations. Regarding the *Closed* approach (Bau et al., 2020), its compositional explanations are associated with abnormally high ActCov and low DetAcc, suggesting that they fail to detect the alignment of more specific concepts. Indeed, the resulting explanations (Section K) are associated with concepts (e.g., bird or animal) that are too general for the given task and fail to highlight relevant alignment exhibited by the probed model (e.g., species, colors).

### 5.2.2 Qualitative Evaluation

To qualitatively validate our results and changes in **human interpretability**, we conducted a user study in which 100 participants were asked to rate, on a scale from 1 (none) to 5 (all), how many individual concepts in the each compositional explanations generated by each method were aligned, precise, and relevant, as defined below. Given a randomly sampled set of activation masks produced by a neuron within a specific activation range, we define a concept as aligned if it appears in at least a subset of the activated masks;

---

[1]As a result, we do not include the additional data provided by Farrell (2022) in our experiments.

Table 3: Average Alignment, Precision, and Relevance scores attributed by users to compositional explanations computed by the competitors. The superscript[*] indicates that the results are computed on a different probing dataset.

| Scores | Align | Prec | Relev |
|---|---|---|---|
| Places365 Probed Model | | | |
| Human (La Rosa et al., 2023) | 3.53 | 2.98 | 3.13 |
| Closed (Bau et al., 2020) | 3.10 | 2.60 | 3.25 |
| Ours | 3.53 | 3.19 | 3.34 |
| CUB Probed Model | | | |
| Human (La Rosa et al., 2023) | 3.17[*] | 3.08[*] | 1.51[*] |
| Closed (Bau et al., 2020) | 3.83 | 3.22 | 2.59 |
| Ours | 3.32 | 3.27 | 4.30 |

precise if its level of granularity matches that of the concepts included in the activation masks; and relevant if it is perceived as discriminative for the given task. A more detailed discussion of the design and setup of this user study can be found in Section G.

Table 3 reports the average scores (with standard deviations and p-values reported in Section G) for ResNet18 trained on Place365 and probed on ADE20K, and ResNet50 trained and probed on CUB. Specifically, for the Place365 model, we are interested in precision and alignment, since all competitors use concept sets relevant to the task. In this setting, these two metrics can reveal hallucinations or concepts that are too coarse or too specific for the dataset. The results show our framework matches the performance of the human baseline in alignment scores, suggesting that hallucinations are not a significant issue in this setting. In contrast, the closed approach is the worst performer, likely because its training dataset does not include all the concepts present in ADE20K and is therefore more prone to hallucinations. Similarly, in terms of precision, our framework achieves scores comparable to the human baseline and significantly better than the closed approach, further highlighting the limitations of the latter. In the CUB dataset, on the other hand, relevance is the primary metric of interest, since in fine-grained classification, we aim to determine whether the identified concepts are relevant to the specific task. In this metric, both the human and the closed baselines perform poorly because their explanations are based on probing datasets that contain coarse concepts that are not sufficiently informative for the task. Conversely, in terms of alignment and precision, all methods achieve similar scores. However, these two metrics are more challenging for participants to evaluate in this setting, since concepts such as "bird" can still receive high ratings for both alignment and precision if the participant is not a domain expert and does not have access to the alternative concepts available to the model (this was a deliberate design choice to avoid bias in the user study, as discussed in Section G).

Overall, these results suggest that our framework is the only approach that demonstrates consistent performance across both datasets, further supporting its effectiveness.

## 5.3 Explanations Analysis

After validating the explanation quality of the proposed framework, this section analyzes the differences between explanations computed using open vocabulary and human-annotated data by leveraging the process introduced in Section 4. Here, we focus on Ade20K, as it is the most extensively studied dataset in the literature of compositional explanations and both approaches share the same concept set under these settings. The analysis is performed over the full set of neurons (512) of the same ResNet18 model probed in Section 5.2.

The first question we aim to address is whether the differences in explanation scores arise from the segmentation masks (e.g., due to segmentation errors) while converging on the same explanation or whether the approaches converge on entirely different explanations. To explore this aspect, we measure the overlap in the explanations' concepts between the two approaches. We find that they share 86%, 91%, 82%, 70%, and 56% of the labels across five different activation ranges (from the lowest to the highest activations, identified

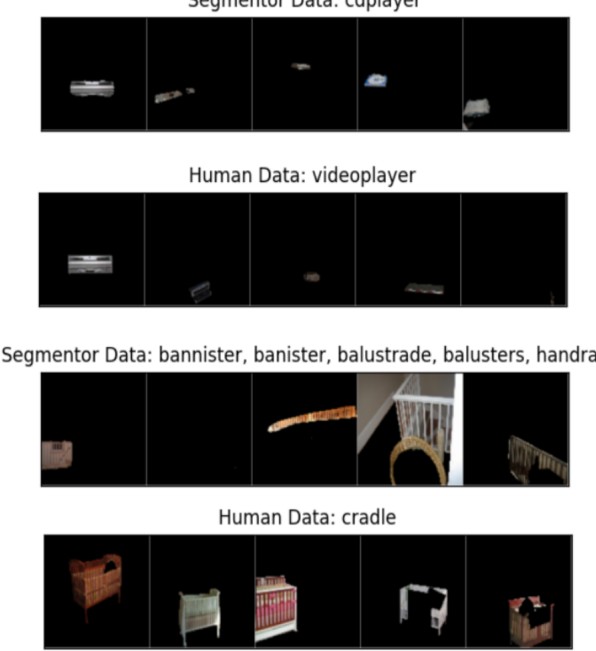

Figure 3: Examples of misalignment between human and model-annotated data due to different granularity in annotations (top) and the lack of concepts capturing patterns (bottom) in the concept set.

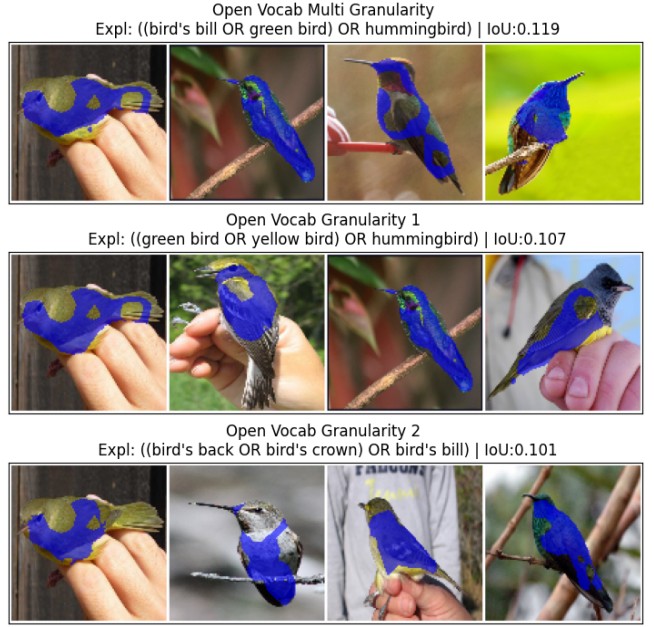

Figure 4: Explanations associated with neuron #19 and cluster 4 generated by our framework using different levels of granularity: color and species concepts (middle row), part-level concepts (bottom row), and a combination of both granularities (top row). Areas of neuron activation within the considered range are highlighted in blue.

as explained in Section 3), respectively. Differences in the lower clusters are due to the algorithm converging differently on activations that do not align with any concept (La Rosa et al., 2023). More interesting, however, is the case of the highest activations, where almost half of the explanations differ. In this case, we

observe that the differences stem from **misalignment**. In some cases, this misalignment can be attributed to hallucinations (e.g., vertical tanks often labeled as arcade machines). However, these cases are easy to identify by visually inspecting the samples that activate the explanations. More subtle and frequent cases of misalignment arise from differences in the concept set and the granularity of segmentations and annotations. For instance, as shown at the top of Figure 3, a neuron associated with the concept "cdplayer" by the first approach is associated with "videoplayer" by explanations computed over model-annotated data. Although these two labels are closely related and likely represent the same underlying object (e.g., a generic "media player"), the difference in annotation and segmentation granularity results in divergent explanations. Differently, at the bottom of the same figure, the two approaches converge on different samples and concepts. However, by visual inspection of these samples, they share highly similar patterns that are not available, as concepts, in the concept set (see Section 6.2).

To measure the extent of these two kinds of misalignment, we leverage the process proposed in Section 4, reducing the total number of concepts from 150 to 101. Due to the incompleteness of the ontology, some misaligned concepts (e.g., cushion and pillow) cannot be unified through this approach. However, the extent of this problem is limited to a few concepts. We leave the resolution of the completeness problem for future work. Then, we measure the extent of co-occurrence between misaligned concepts. Through this process, we observe that granularity impacts 12% of the total concepts, with 4% *unifiable* through the ontology and 8% hyper-related. The latter includes concepts whose annotations and segmentations are inconsistent or not aligned in granularity (e.g., traffic light vs road or mountain vs hill). Finally, 17% are highly related and 19% exhibit low or no co-occurrence. These represent cases where both approaches struggle due to the limitations of the concept set (similarly to Figure 3). While this limitation could potentially be mitigated through refinements, some areas of misalignment (e.g., patterns) need further advancements in semantic segmentation to support concepts that are highly relevant for explainability but remain underexplored in standard semantic segmentation settings. In this direction, **we identify and discuss these limitations and potential research directions in Section B** .

## 6 Application Scenarios

In this section, we show how we can exploit the proposed framework to improve the explanations associated with neurons and improve our understanding of what they recognize.

### 6.1 Supporting Custom Granularity

As described in Section 3, our framework supports multiple granularities through the use of concept subsets. These sets allow the algorithm to adjust explanations to the most aligned granularity. However, the framework can also be used to study individual neurons at different granularities, guided by the user. This capability is important because, due to superposition (Elhage et al., 2022; O'Mahony et al., 2023; Dreyer et al., 2024) and the fixed maximum length of explanations, some concepts aligned to the neuron may not be included in the explanation if they are weaker than those selected by the algorithm or do not add enough value to the previously selected concepts. Figure 4 shows multi-granularity explanations and two single-granularity explanations for a neuron in the CUB model probed in Section 5.2. The first individual granularity represents bird-level concepts (i.e., shapes, colors, and species), while the second one represents birds' parts. Although the explanation that includes all of the granularities achieves the highest score, the analysis of individual granularities provides further insights into the neuron's recognition power. In this example, we can derive that the neuron recognizes species and colored birds as well as specific parts of these birds. **This analysis offers the user a more complete picture of the concepts learned by neurons**. Notably, this analysis cannot be supported by the *Closed* approach (Bau et al., 2020) because it uses only one concept set and can be only partially supported (from lower to higher granularity) when combining ontologies and human-annotated data. Thus, this flexibility represents an additional advantage of our framework.

Figure 5: An example of how iterative refinements of the concept set can improve open vocabulary explanations.

### 6.2 Improving Explanations via Refinements

This section showcases how to improve misaligned explanations by correcting the concept set. In particular, the goal is to **analyze neurons' activations and explanations, identify possible misalignments due to the concept set, and fix them by refining the concept set**. Specifically, given an explanation of length $n$, we isolate the effect of a given concept into the explanation and we visually compare it with the neuron's activations not captured by the non-isolated part of the explanation (see Section F for the procedure). Here, we focus on the misaligned labels identified in Section 5.3. For example, as shown in Figure 5, when examining neuron 1, we observed that this neuron appears to represent concepts such as "shop" or "window shop". However, the probing dataset (Ade20K) does not include labels for these concepts, causing both the "*human*" baseline and our method to converge on related concepts (e.g., trader name). To address this problem, we added the missing concepts to the concept set and re-generated the masks for our framework.

It is important to note that in this process, *the user is not correcting the explanations but the concept set*. This means that when the user suggests a concept not aligned with the neuron's activation, the segmentation model will still identify the new concept, but the compositional algorithm will discard it because it would be less aligned with the activation than the previous concepts. This ensures that the neuron explanation is faithful even if the concept set is modified. Figure 5 shows that, after the refinement, the framework includes new concepts in the explanations and the updated explanations reach higher IoU scores than before. This means that the updated explanations are better aligned with the neuron activations or, equivalently, that the framework more accurately captures the alignment of the neuron activations. Finally, note that these improvements are not possible when using closed vocabulary segmentation models and require extensive and costly human labor to both annotate and fix the consistency of annotations in the human-based approaches.

## 7 Conclusion

In this paper, we introduced a novel framework for computing open vocabulary compositional explanations, addressing one of the main limitations of compositional explanations: their dependency on human-annotated datasets, and introduced a process to analyze compositional explanations computed by different methods. We demonstrated that our framework produces explanations that are comparable to those produced by previous approaches when human annotations are available, while offering greater flexibility and broader

applicability, and outperform previous approaches when human annotations are not available. We also call for further research in semantic segmentation to better support explainability tasks. Finally, future research directions could explore more advanced relationships between concepts, adapt the framework to different domains, and develop adaptive mechanisms to automatically identify the most suitable concept set for a given task.

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

# A   Extended Quantitative Evaluation

## A.1   Complete Results

This section complements the quantitative evaluation of our proposed frameworks by providing results computed using additional configurations. Results include both average and standard deviation. It is important to emphasize that the **explanations and the metrics used to measure their quality are not expected to exhibit low variance**. This variability arises from the overparameterization and the learning process of deep neural networks. Indeed, as observed by Bau et al. (2017) and Mu & Andreas (2020), not all neurons within the network are aligned with specific concepts, leading to high variability in the degree of alignment. This effect is especially pronounced for compositional explanations and the metrics computed over pixel-level data, where the alignment between neuron activations and labeled concepts can fluctuate significantly.

Specifically, this section broadens the experimental settings and demonstrates the robustness of our framework by testing multiple activation ranges and multiple open-vocabulary segmentation models as backbones. Regarding the activation ranges, we use K-Means to identify five clusters (i.e., activation ranges) in the neuron activations, as in La Rosa et al. (2023), and we associate a number with each cluster: the lower the number, the lower the activations included in that cluster. As backbones, from the extensive range of models available in the literature (Xu et al., 2023a; 2022c; Wysoczańska et al., 2024; Jiao et al., 2023; Cha et al., 2023; Liang et al., 2023a; Rao et al., 2022; Luo et al., 2023; Xu et al., 2022a; Shin et al., 2022; Zhou et al., 2022; Barsellotti et al., 2024; Zhou et al., 2023; Xu et al., 2023b; Ren et al., 2024; Lüddecke & Ecker, 2022; Ghiasi et al., 2022; Zou et al., 2024; 2023; Li et al., 2024b; Yu et al., 2023; Shen et al., 2024; Wang et al., 2023; Xu et al., 2023c), we selected four additional representative models: MasQCLIP (Xu et al., 2023d), SCAN (VitL) (Liu et al., 2024), SED (L) (Xie et al., 2024), and OpenSeed (Swin-T) (Zhang et al., 2023). These models have been selected based on the following criteria: (i) they are among the most recent ones and published in major conferences, (ii) the pre-trained models are available to the general public, and (ii) the implementation is compatible with the technical settings considered in this paper (i.e., PyTorch 1.3 (Paszke et al., 2019), Detectron2 (Wu et al., 2019), MMEngine 1.6.2 (Contributors, 2018), and MMSegmentation 0.27.0 (Contributors, 2020)), without requiring major code changes. While these models serve as examples of implementations of the framework, better explanations could potentially be obtained by using models beyond the settings tested in this paper, especially when using models trained on very large corpora (Zou et al., 2024; Li et al., 2024a). As weights, we use the pre-trained weights available in the official repositories of selected models.

In Tables 4 to 11, we report the complete results for all the open vocabulary segmentation models and the five activation ranges. Specifically, we randomly extract 50 neurons for each probed model and we generate explanations for those neurons using as a probing dataset the validation split of the following datasets:

Mapillary Vistas (Neuhold et al., 2017) (65 concepts), Cityscapes (Cordts et al., 2016) (19 concepts), Pascal VOC (Everingham et al., 2012) (20 concepts), PASCAL-Context-459 (Mottaghi et al., 2014) (459 concepts), Ade20k (in both its standard (150 concepts) and extended version (847 concepts)) (Zhou et al., 2017b), and COCO-Stuff (171 concepts)(Caesar et al., 2018). Note that we do not include OpenSeed in the Mapillary Vistas evaluation due to technical limitations[2]. As a probed model, we use the same model used in Section 5 trained on Place365, since the learned place categories are related to the concepts and segmentation masks included in these datasets[3].

With respect to the activation ranges, our framework reaches comparable results to the human baseline in datasets where they share the same concept set, in line with what we observed in Section 5. Conversely, in the case of CUB, which is the scenario our framework is targeting, Table 5 shows that our framework achieves the highest alignment scores in all the activation ranges (i.e., clusters) except for the lowest activations (Cluster 1), where scores are slightly worse. However, as noted by La Rosa et al. (2023), the lowest clusters often include fixed (uninformative) compositional explanations where the algorithm converges when no alignment is observed. In such cases, the explanations generated by different competitors differ by only one concept within these degenerate explanations (i.e., the "*human*" baseline converges on "building" while our framework converges on "person"), rendering the differences insignificant. Regarding the *Closed* approach (Bau et al., 2020), it achieves reasonable IoU scores in lower activation ranges because those ranges capture general concepts (e.g., water, sky), which are shared between CUB and COCO (Lin et al., 2014), the dataset used to train this approach. However, in the higher clusters, its compositional explanations fail to detect the alignment of more specific concepts. Indeed, the resulting explanations are associated with concepts that are too general for the given task, as noted in the previous section.

With respect to the backbone selection, we can note that no single model is able to outperform all the others in every setting, with each excelling in specific activation ranges and scores. These little differences can be attributed to the specific capabilities of recognizing more general or specific concepts of each segmentation model. For example, MasQCLIP is the only model that includes the "background" concept (by default). This difference explains the differences in the lower clusters of the Ade20K settings, typically influenced by default rules (La Rosa et al., 2023) that include this kind of concept. In the CUB Case (Table 5), as a concept set, we use the concept set identified for *Cat-Seg* (see Section D). This implies that the reported results could be further improved by refining the concept sets to better match the specific characteristics of each model. This choice is motivated by the fact that **the goal of this experiment is not to select the best open vocabulary segmentation model** but to assess the general validity of the combination between compositional explanations and research in open vocabulary semantic segmentation. Overall, we observe that while *CAT-Seg* achieves the highest performance in this dataset, all the other models except OpenSeed are able to outperform the "human" baseline, highlighting the effectiveness of the solution. In the case of *OpenSeed*, we note that *OpenSeed* and the *Closed* approach are both trained on the same dataset (i.e., COCO (Lin et al., 2014)) and share the same backbone (i.e., Swin-T (Liu et al., 2021)), suggesting that this backbone, responsible for the recognition of the mask, could be the factor influencing the lower performance of this specific implementation.

Overall, considering a wider range of datasets and activation ranges, we argue that most of the tested models achieve satisfactory results and no single model is able to outperform all the others in every setting, with each excelling in specific activation ranges and scores.

**Explanation Differences:** Given that no segmentation model consistently outperforms all the others, a natural question is how different the explanations computed using different segmentors are and, when explanations disagree, when one should be considered more reliable than the others.

To investigate this question, we measure the proportion of concepts that are unrelated (i.e., associated with low co-occurrence, as defined in Section 4) across explanations generated by different segmentors. This choice is motivated by the fact that related concepts often correspond to specializations or generalizations of the same semantic concept. In these cases, the resulting explanations can be considered almost equivalent and

---

[2]Out of Memory issues on a GTX 3090 graphic card.
[3]Note that we do not probe models trained on these datasets, as they are segmentation models specifically trained to classify the same concepts. This undermines the utility of compositional explanations.

Table 4: Avg. and Std. Dev. scores for explanations associated with a model trained on the Place365 dataset using Ade20K as a probing dataset.

| Cluster | Method | IoU | ActCov | DetAcc |
|---|---|---|---|---|
| 1 | Human (La Rosa et al., 2023) | 0.218 ± 0.016 | 0.349 ± 0.019 | 0.368 ± 0.032 |
| | Closed (Bau et al., 2020) | 0.214 ± 0.015 | 0.339 ± 0.019 | 0.368 ± 0.032 |
| | Ours$_{\text{MasQCLIP}}$ | 0.112 ± 0.010 | 0.137 ± 0.013 | 0.373 ± 0.036 |
| | Ours$_{\text{SCAN}}$ | 0.202 ± 0.014 | 0.302 ± 0.014 | 0.379 ± 0.034 |
| | Ours$_{\text{SED}}$ | 0.206 ± 0.014 | 0.313 ± 0.016 | 0.377 ± 0.033 |
| | Ours$_{\text{CAT-Seg}}$ | 0.210 ± 0.015 | 0.325 ± 0.015 | 0.374 ± 0.034 |
| | Ours$_{\text{OpenSeed}}$ | 0.226 ± 0.015 | 0.372 ± 0.021 | 0.367 ± 0.032 |
| 2 | Human (La Rosa et al., 2023) | 0.131 ± 0.020 | 0.319 ± 0.038 | 0.184 ± 0.035 |
| | Closed (Bau et al., 2020) | 0.130 ± 0.019 | 0.304 ± 0.038 | 0.187 ± 0.036 |
| | Ours$_{\text{MasQCLIP}}$ | 0.090 ± 0.014 | 0.142 ± 0.026 | 0.200 ± 0.033 |
| | Ours$_{\text{SCAN}}$ | 0.125 ± 0.021 | 0.272 ± 0.042 | 0.190 ± 0.035 |
| | Ours$_{\text{SED}}$ | 0.128 ± 0.020 | 0.285 ± 0.040 | 0.190 ± 0.034 |
| | Ours$_{\text{CAT-Seg}}$ | 0.129 ± 0.020 | 0.298 ± 0.038 | 0.188 ± 0.036 |
| | Ours$_{\text{OpenSeed}}$ | 0.136 ± 0.020 | 0.340 ± 0.046 | 0.186 ± 0.032 |
| 3 | Human (La Rosa et al., 2023) | 0.101 ± 0.029 | 0.261 ± 0.079 | 0.149 ± 0.048 |
| | Closed (Bau et al., 2020) | 0.104 ± 0.028 | 0.262 ± 0.080 | 0.157 ± 0.047 |
| | Ours$_{\text{MasQCLIP}}$ | 0.087 ± 0.023 | 0.157 ± 0.049 | 0.168 ± 0.044 |
| | Ours$_{\text{SCAN}}$ | 0.104 ± 0.030 | 0.244 ± 0.079 | 0.161 ± 0.047 |
| | Ours$_{\text{SED}}$ | 0.105 ± 0.030 | 0.256 ± 0.077 | 0.156 ± 0.046 |
| | Ours$_{\text{CAT-Seg}}$ | 0.104 ± 0.029 | 0.255 ± 0.076 | 0.156 ± 0.045 |
| | Ours$_{\text{OpenSeed}}$ | 0.108 ± 0.029 | 0.296 ± 0.089 | 0.152 ± 0.044 |
| 4 | Human (La Rosa et al., 2023) | 0.082 ± 0.030 | 0.2180 ± 0.107 | 0.140 ± 0.070 |
| | Closed (Bau et al., 2020) | 0.089 ± 0.030 | 0.223 ± 0.096 | 0.142 ± 0.057 |
| | Ours$_{\text{MasQCLIP}}$ | 0.087 ± 0.032 | 0.182 ± 0.078 | 0.154 ± 0.055 |
| | Ours$_{\text{SCAN}}$ | 0.093 ± 0.034 | 0.222 ± 0.109 | 0.154 ± 0.060 |
| | Ours$_{\text{SED}}$ | 0.091 ± 0.033 | 0.228 ± 0.114 | 0.152 ± 0.064 |
| | Ours$_{\text{CAT-Seg}}$ | 0.089 ± 0.030 | 0.231 ± 0.096 | 0.149 ± 0.072 |
| | Ours$_{\text{OpenSeed}}$ | 0.088 ± 0.032 | 0.256 ± 0.131 | 0.137 ± 0.058 |
| 5 | Human (La Rosa et al., 2023) | 0.083 ± 0.069 | 0.197 ± 0.143 | 0.166 ± 0.129 |
| | Closed (Bau et al., 2020) | 0.071 ± 0.042 | 0.211 ± 0.137 | 0.120 ± 0.079 |
| | Ours$_{\text{MasQCLIP}}$ | 0.075 ± 0.036 | 0.214 ± 0.126 | 0.118 ± 0.059 |
| | Ours$_{\text{SCAN}}$ | 0.082 ± 0.044 | 0.220 ± 0.132 | 0.139 ± 0.083 |
| | Ours$_{\text{SED}}$ | 0.081 ± 0.044 | 0.216 ± 0.134 | 0.137 ± 0.078 |
| | Ours$_{\text{CAT-Seg}}$ | 0.093 ± 0.066 | 0.214 ± 0.147 | 0.178 ± 0.120 |
| | Ours$_{\text{OpenSeed}}$ | 0.064 ± 0.038 | 0.215 ± 0.151 | 0.110 ± 0.079 |

are unlikely to create confusion for users. Conversely, unrelated concepts may lead to different explanations and potentially different interpretations of neuron behavior. Figure 6 reports the percentage of unrelated concepts obtained from pairwise comparisons between segmentors, treating one segmentor at a time as the reference. The heatmap suggests that the extent of this disagreement is limited, accounting on average for approximately one concept every five concepts (roughly one concept every two explanations) and, in the worst case, approximately one concept per explanation. This result is reassuring, as it indicates that explanations generated by different segmentors are generally closely related and differ only in a small number of concepts.

To further investigate the origin of these differences, we perform the following analysis. Let $E_A$ be the explanation generated using annotations produced by segmentation model $A$, and let $E_B$ be the explanation

Table 5: Avg. and Std. Dev. scores for explanations associated with a model trained on the CUB dataset using CUB as a probing dataset.

| Cluster | Method | IoU | ActCov | DetAcc |
|---|---|---|---|---|
| 1 | Human (La Rosa et al., 2023) | 0.248 ± 0.022 | 0.356 ± 0.019 | 0.451 ± 0.057 |
| | Closed (Bau et al., 2020) | 0.388 ± 0.040 | 0.635 ± 0.019 | 0.501 ± 0.061 |
| | Ours$_{MasQCLIP}$ | 0.306 ± 0.028 | 0.441 ± 0.022 | 0.502 ± 0.063 |
| | Ours$_{SCAN}$ | 0.439 ± 0.045 | 0.836 ± 0.020 | 0.481 ± 0.055 |
| | Ours$_{SED}$ | 0.405 ± 0.040 | 0.678 ± 0.016 | 0.503 ± 0.059 |
| | Ours$_{CAT-Seg}$ | 0.357 ± 0.034 | 0.553 ± 0.019 | 0.504 ± 0.060 |
| | Ours$_{OpenSeed}$ | 0.470 ± 0.051 | 0.929 ± 0.030 | 0.488 ± 0.059 |
| 2 | Human (La Rosa et al., 2023) | 0.130 ± 0.035 | 0.312 ± 0.059 | 0.185 ± 0.057 |
| | Closed (Bau et al., 2020) | 0.170 ± 0.032 | 0.505 ± 0.152 | 0.214 ± 0.041 |
| | Ours$_{MasQCLIP}$ | 0.161 ± 0.024 | 0.407 ± 0.076 | 0.214 ± 0.035 |
| | Ours$_{SCAN}$ | 0.174 ± 0.034 | 0.563 ± 0.188 | 0.209 ± 0.038 |
| | Ours$_{SED}$ | 0.176 ± 0.032 | 0.522 ± 0.138 | 0.215 ± 0.036 |
| | Ours$_{CAT-Seg}$ | 0.173 ± 0.028 | 0.463 ± 0.102 | 0.221 ± 0.038 |
| | Ours$_{OpenSeed}$ | 0.179 ± 0.033 | 0.602 ± 0.198 | 0.211 ± 0.036 |
| 3 | Human (La Rosa et al., 2023) | 0.085 ± 0.031 | 0.228 ± 0.088 | 0.126 ± 0.046 |
| | Closed (Bau et al., 2020) | 0.142 ± 0.030 | 0.453 ± 0.116 | 0.175 ± 0.039 |
| | Ours$_{MasQCLIP}$ | 0.136 ± 0.027 | 0.388 ± 0.074 | 0.176 ± 0.038 |
| | Ours$_{SCAN}$ | 0.144 ± 0.029 | 0.422 ± 0.101 | 0.182 ± 0.039 |
| | Ours$_{SED}$ | 0.143 ± 0.028 | 0.425 ± 0.089 | 0.180 ± 0.036 |
| | Ours$_{CAT-Seg}$ | 0.147 ± 0.030 | 0.432 ± 0.093 | 0.185 ± 0.038 |
| | Ours$_{OpenSeed}$ | 0.141 ± 0.027 | 0.463 ± 0.101 | 0.170 ± 0.034 |
| 4 | Human (La Rosa et al., 2023) | 0.063 ± 0.030 | 0.167 ± 0.101 | 0.105 ± 0.050 |
| | Closed (Bau et al., 2020) | 0.091 ± 0.027 | 0.571 ± 0.136 | 0.100 ± 0.031 |
| | Ours$_{MasQCLIP}$ | 0.098 ± 0.024 | 0.336 ± 0.105 | 0.126 ± 0.035 |
| | Ours$_{SCAN}$ | 0.101 ± 0.026 | 0.426 ± 0.139 | 0.123 ± 0.037 |
| | Ours$_{SED}$ | 0.103 ± 0.025 | 0.383 ± 0.122 | 0.129 ± 0.038 |
| | Ours$_{CAT-Seg}$ | 0.113 ± 0.027 | 0.356 ± 0.115 | 0.147 ± 0.039 |
| | Ours$_{OpenSeed}$ | 0.095 ± 0.025 | 0.413 ± 0.089 | 0.111 ± 0.031 |
| 5 | Human (La Rosa et al., 2023) | 0.052 ± 0.029 | 0.144 ± 0.124 | 0.100 ± 0.058 |
| | Closed (Bau et al., 2020) | 0.029 ± 0.014 | 0.674 ± 0.195 | 0.033 ± 0.028 |
| | Ours$_{MasQCLIP}$ | 0.059 ± 0.019 | 0.165 ± 0.067 | 0.095 ± 0.044 |
| | Ours$_{SCAN}$ | 0.060 ± 0.021 | 0.153 ± 0.080 | 0.112 ± 0.059 |
| | Ours$_{SED}$ | 0.068 ± 0.023 | 0.155 ± 0.069 | 0.125 ± 0.055 |
| | Ours$_{CAT-Seg}$ | 0.077 ± 0.024 | 0.188 ± 0.072 | 0.131 ± 0.056 |
| | Ours$_{OpenSeed}$ | 0.042 ± 0.016 | 0.170 ± 0.103 | 0.060 ± 0.039 |

generated using annotations produced by segmentation model $B$. Whenever $E_A$ and $E_B$ differ, we recompute the alignment score of $E_B$ using the annotations generated by model $A$ and compare it with the alignment score of $E_A$. The purpose of this experiment is to determine whether the disagreement arises from substantial annotation errors or from the presence of multiple explanations with very similar alignment scores and only minor differences in the segmentation masks. Figure 7 shows that the difference in alignment is very small, accounting for less than 0.01 in most cases. This result indicates that an explanation achieving high alignment on annotations generated by segmentor $A$ also tends to achieve high alignment on annotations generated by segmentor $B$. Therefore, the observed differences are more likely to arise from small variations in how individual segmentors recognize specific concepts rather than from fundamental segmentation errors. Consequently, both explanations should generally be considered valid and faithful. The existence of multiple valid explanations is related to the well-known problem of superposition (Elhage et al., 2022),

Table 6: Avg. scores for explanations associated with a model trained on the Place365 dataset using Ade20K-Extended (847 classes) as a probing dataset.

| Cluster | Method | IoU | ActCov | DetAcc |
|---------|--------|-----|--------|--------|
| 1 | Human (La Rosa et al., 2023) | 0.218 ± 0.016 | 0.350 ± 0.016 | 0.367 ± 0.034 |
| | Closed (Bau et al., 2020) | 0.215 ± 0.015 | 0.340 ± 0.019 | 0.369 ± 0.033 |
| | Ours$_{\text{MasQCLIP}}$ | 0.087 ± 0.007 | 0.100 ± 0.008 | 0.393 ± 0.033 |
| | Ours$_{\text{SCAN}}$ | 0.185 ± 0.014 | 0.263 ± 0.014 | 0.383 ± 0.038 |
| | Ours$_{\text{SED}}$ | 0.198 ± 0.014 | 0.293 ± 0.013 | 0.379 ± 0.035 |
| | Ours$_{\text{CAT-Seg}}$ | 0.203 ± 0.014 | 0.304 ± 0.013 | 0.378 ± 0.034 |
| | Ours$_{\text{OpenSeed}}$ | 0.223 ± 0.015 | 0.363 ± 0.020 | 0.368 ± 0.032 |
| 2 | Human (La Rosa et al., 2023) | 0.131 ± 0.020 | 0.320 ± 0.039 | 0.184 ± 0.035 |
| | Closed (Bau et al., 2020) | 0.130 ± 0.019 | 0.308 ± 0.039 | 0.186 ± 0.035 |
| | Ours$_{\text{MasQCLIP}}$ | 0.076 ± 0.015 | 0.107 ± 0.021 | 0.210 ± 0.043 |
| | Ours$_{\text{SCAN}}$ | 0.114 ± 0.020 | 0.222 ± 0.038 | 0.193 ± 0.042 |
| | Ours$_{\text{SED}}$ | 0.122 ± 0.020 | 0.258 ± 0.035 | 0.191 ± 0.039 |
| | Ours$_{\text{CAT-Seg}}$ | 0.124 ± 0.020 | 0.272 ± 0.034 | 0.188 ± 0.037 |
| | Ours$_{\text{OpenSeed}}$ | 0.133 ± 0.019 | 0.333 ± 0.040 | 0.184 ± 0.034 |
| 3 | Human (La Rosa et al., 2023) | 0.101 ± 0.029 | 0.263 ± 0.083 | 0.149 ± 0.046 |
| | Closed (Bau et al., 2020) | 0.104 ± 0.028 | 0.262 ± 0.079 | 0.156 ± 0.047 |
| | Ours$_{\text{MasQCLIP}}$ | 0.075 ± 0.023 | 0.118 ± 0.042 | 0.176 ± 0.049 |
| | Ours$_{\text{SCAN}}$ | 0.096 ± 0.027 | 0.203 ± 0.062 | 0.163 ± 0.047 |
| | Ours$_{\text{SED}}$ | 0.100 ± 0.029 | 0.230 ± 0.066 | 0.155 ± 0.045 |
| | Ours$_{\text{CAT-Seg}}$ | 0.100 ± 0.029 | 0.240 ± 0.073 | 0.155 ± 0.047 |
| | Ours$_{\text{OpenSeed}}$ | 0.104 ± 0.028 | 0.280 ± 0.081 | 0.151 ± 0.046 |
| 4 | Human (La Rosa et al., 2023) | 0.083 ± 0.030 | 0.219 ± 0.107 | 0.142 ± 0.073 |
| | Closed (Bau et al., 2020) | 0.089 ± 0.030 | 0.230 ± 0.098 | 0.141 ± 0.058 |
| | Ours$_{\text{MasQCLIP}}$ | 0.079 ± 0.027 | 0.141 ± 0.055 | 0.166 ± 0.057 |
| | Ours$_{\text{SCAN}}$ | 0.086 ± 0.029 | 0.192 ± 0.083 | 0.155 ± 0.072 |
| | Ours$_{\text{SED}}$ | 0.086 ± 0.031 | 0.209 ± 0.085 | 0.146 ± 0.071 |
| | Ours$_{\text{CAT-Seg}}$ | 0.086 ± 0.031 | 0.211 ± 0.091 | 0.148 ± 0.074 |
| | Ours$_{\text{OpenSeed}}$ | 0.088 ± 0.030 | 0.232 ± 0.110 | 0.145 ± 0.067 |
| 5 | Human (La Rosa et al., 2023) | 0.098 ± 0.076 | 0.196 ± 0.148 | 0.242 ± 0.171 |
| | Closed (Bau et al., 2020) | 0.071 ± 0.042 | 0.221 ± 0.145 | 0.119 ± 0.081 |
| | Ours$_{\text{MasQCLIP}}$ | 0.103 ± 0.056 | 0.186 ± 0.100 | 0.207 ± 0.092 |
| | Ours$_{\text{SCAN}}$ | 0.107 ± 0.072 | 0.195 ± 0.127 | 0.240 ± 0.140 |
| | Ours$_{\text{SED}}$ | 0.106 ± 0.071 | 0.195 ± 0.136 | 0.234 ± 0.135 |
| | Ours$_{\text{CAT-Seg}}$ | 0.103 ± 0.070 | 0.208 ± 0.145 | 0.236 ± 0.139 |
| | Ours$_{\text{OpenSeed}}$ | 0.079 ± 0.065 | 0.226 ± 0.169 | 0.152 ± 0.119 |

whereby neurons may encode multiple concepts and relationships within the same activation range. It is also influenced by the maximum explanation length imposed by compositional explanations, which select only the single highest-alignment formula rather than returning all highly aligned alternatives in order to maintain interpretability.

In scenarios where returning multiple explanations is undesirable and a single explanation must be selected from those generated by multiple segmentors, several strategies are possible. These include unifying explanations through the ontology-based process described in Section 4, selecting the explanation returned by the largest number of segmentors (majority voting), selecting the explanation associated with the segmentor that is most reliable on the concepts involved in the explanation (e.g., as measured on a validation set), or selecting the explanation associated with the highest alignment score.

Table 7: Avg. and Std. Dev. scores for explanations associated with a model trained on the Place365 dataset using Mapillary Vistas as a probing dataset.

| Cluster | Method | IoU | ActCov | DetAcc |
|---|---|---|---|---|
| 1 | Human (La Rosa et al., 2023) | 0.304 ± 0.042 | 0.652 ± 0.040 | 0.363 ± 0.052 |
| | Closed (Bau et al., 2020) | 0.310 ± 0.043 | 0.680 ± 0.045 | 0.363 ± 0.052 |
| | Ours$_{\text{MasQCLIP}}$ | 0.202 ± 0.021 | 0.308 ± 0.019 | 0.374 ± 0.056 |
| | Ours$_{\text{SCAN}}$ | 0.311 ± 0.042 | 0.691 ± 0.038 | 0.362 ± 0.051 |
| | Ours$_{\text{SED}}$ | 0.313 ± 0.042 | 0.714 ± 0.051 | 0.358 ± 0.051 |
| | Ours$_{\text{CAT-Seg}}$ | 0.313 ± 0.042 | 0.704 ± 0.038 | 0.361 ± 0.052 |
| 2 | Human (La Rosa et al., 2023) | 0.161 ± 0.035 | 0.517 ± 0.106 | 0.192 ± 0.043 |
| | Closed (Bau et al., 2020) | 0.166 ± 0.037 | 0.530 ± 0.116 | 0.197 ± 0.045 |
| | Ours$_{\text{MasQCLIP}}$ | 0.138 ± 0.028 | 0.301 ± 0.041 | 0.205 ± 0.050 |
| | Ours$_{\text{SCAN}}$ | 0.168 ± 0.037 | 0.552 ± 0.118 | 0.196 ± 0.043 |
| | Ours$_{\text{SED}}$ | 0.168 ± 0.039 | 0.569 ± 0.119 | 0.195 ± 0.046 |
| | Ours$_{\text{CAT-Seg}}$ | 0.169 ± 0.038 | 0.570 ± 0.121 | 0.195 ± 0.044 |
| 3 | Human (La Rosa et al., 2023) | 0.121 ± 0.039 | 0.409 ± 0.109 | 0.152 ± 0.055 |
| | Closed (Bau et al., 2020) | 0.124 ± 0.042 | 0.436 ± 0.126 | 0.151 ± 0.055 |
| | Ours$_{\text{MasQCLIP}}$ | 0.110 ± 0.035 | 0.276 ± 0.065 | 0.159 ± 0.055 |
| | Ours$_{\text{SCAN}}$ | 0.126 ± 0.043 | 0.460 ± 0.116 | 0.151 ± 0.054 |
| | Ours$_{\text{SED}}$ | 0.125 ± 0.043 | 0.476 ± 0.130 | 0.149 ± 0.057 |
| | Ours$_{\text{CAT-Seg}}$ | 0.126 ± 0.043 | 0.482 ± 0.125 | 0.149 ± 0.056 |
| 4 | Human (La Rosa et al., 2023) | 0.088 ± 0.042 | 0.323 ± 0.139 | 0.123 ± 0.076 |
| | Closed (Bau et al., 2020) | 0.087 ± 0.040 | 0.334 ± 0.150 | 0.117 ± 0.065 |
| | Ours$_{\text{MasQCLIP}}$ | 0.083 ± 0.037 | 0.241 ± 0.100 | 0.125 ± 0.081 |
| | Ours$_{\text{SCAN}}$ | 0.087 ± 0.041 | 0.370 ± 0.155 | 0.112 ± 0.057 |
| | Ours$_{\text{SED}}$ | 0.086 ± 0.041 | 0.371 ± 0.165 | 0.116 ± 0.072 |
| | Ours$_{\text{CAT-Seg}}$ | 0.087 ± 0.042 | 0.372 ± 0.163 | 0.116 ± 0.076 |
| 5 | Human (La Rosa et al., 2023) | 0.053 ± 0.036 | 0.266 ± 0.200 | 0.080 ± 0.068 |
| | Closed (Bau et al., 2020) | 0.050 ± 0.028 | 0.254 ± 0.207 | 0.082 ± 0.068 |
| | Ours$_{\text{MasQCLIP}}$ | 0.056 ± 0.038 | 0.187 ± 0.114 | 0.082 ± 0.072 |
| | Ours$_{\text{SCAN}}$ | 0.052 ± 0.033 | 0.264 ± 0.217 | 0.081 ± 0.057 |
| | Ours$_{\text{SED}}$ | 0.052 ± 0.035 | 0.273 ± 0.223 | 0.077 ± 0.057 |
| | Ours$_{\text{CAT-Seg}}$ | 0.052 ± 0.033 | 0.271 ± 0.226 | 0.078 ± 0.051 |

Table 8: Avg. and Std. Dev. scores for explanations associated with a model trained on the Place365 dataset using Cityscapes as a probing dataset.

| Cluster | Method | IoU | ActCov | DetAcc |
|---|---|---|---|---|
| 1 | Human (La Rosa et al., 2023) | 0.294 ± 0.038 | 0.650 ± 0.055 | 0.353 ± 0.055 |
| | Closed (Bau et al., 2020) | 0.309 ± 0.042 | 0.687 ± 0.061 | 0.363 ± 0.059 |
| | Ours$_{\text{MasQCLIP}}$ | 0.306 ± 0.045 | 0.701 ± 0.048 | 0.354 ± 0.055 |
| | Ours$_{\text{SCAN}}$ | 0.316 ± 0.044 | 0.753 ± 0.067 | 0.355 ± 0.055 |
| | Ours$_{\text{SED}}$ | 0.310 ± 0.042 | 0.724 ± 0.068 | 0.355 ± 0.057 |
| | Ours$_{\text{CAT-Seg}}$ | 0.314 ± 0.043 | 0.729 ± 0.066 | 0.359 ± 0.057 |
| | Ours$_{\text{OpenSeed}}$ | 0.309 ± 0.041 | 0.711 ± 0.069 | 0.357 ± 0.058 |
| 2 | Human (La Rosa et al., 2023) | 0.178 ± 0.044 | 0.580 ± 0.097 | 0.206 ± 0.052 |
| | Closed (Bau et al., 2020) | 0.183 ± 0.046 | 0.620 ± 0.100 | 0.208 ± 0.053 |
| | Ours$_{\text{MasQCLIP}}$ | 0.177 ± 0.048 | 0.639 ± 0.097 | 0.197 ± 0.054 |
| | Ours$_{\text{SCAN}}$ | 0.186 ± 0.046 | 0.655 ± 0.117 | 0.207 ± 0.051 |
| | Ours$_{\text{SED}}$ | 0.184 ± 0.048 | 0.655 ± 0.106 | 0.205 ± 0.054 |
| | Ours$_{\text{CAT-Seg}}$ | 0.185 ± 0.047 | 0.649 ± 0.115 | 0.207 ± 0.053 |
| | Ours$_{\text{OpenSeed}}$ | 0.183 ± 0.047 | 0.650 ± 0.103 | 0.204 ± 0.053 |
| 3 | Human (La Rosa et al., 2023) | 0.131 ± 0.045 | 0.500 ± 0.099 | 0.154 ± 0.056 |
| | Closed (Bau et al., 2020) | 0.130 ± 0.044 | 0.538 ± 0.112 | 0.149 ± 0.054 |
| | Ours$_{\text{MasQCLIP}}$ | 0.120 ± 0.042 | 0.463 ± 0.149 | 0.142 ± 0.049 |
| | Ours$_{\text{SCAN}}$ | 0.131 ± 0.044 | 0.563 ± 0.115 | 0.149 ± 0.054 |
| | Ours$_{\text{SED}}$ | 0.130 ± 0.045 | 0.571 ± 0.138 | 0.148 ± 0.055 |
| | Ours$_{\text{CAT-Seg}}$ | 0.131 ± 0.045 | 0.565 ± 0.122 | 0.149 ± 0.054 |
| | Ours$_{\text{OpenSeed}}$ | 0.130 ± 0.045 | 0.558 ± 0.130 | 0.148 ± 0.055 |
| 4 | Human (La Rosa et al., 2023) | 0.091 ± 0.047 | 0.412 ± 0.188 | 0.114 ± 0.067 |
| | Closed (Bau et al., 2020) | 0.088 ± 0.042 | 0.391 ± 0.201 | 0.109 ± 0.052 |
| | Ours$_{\text{MasQCLIP}}$ | 0.082 ± 0.036 | 0.342 ± 0.173 | 0.107 ± 0.049 |
| | Ours$_{\text{SCAN}}$ | 0.088 ± 0.043 | 0.417 ± 0.210 | 0.108 ± 0.053 |
| | Ours$_{\text{SED}}$ | 0.087 ± 0.042 | 0.447 ± 0.208 | 0.105 ± 0.051 |
| | Ours$_{\text{CAT-Seg}}$ | 0.088 ± 0.042 | 0.432 ± 0.200 | 0.106 ± 0.050 |
| | Ours$_{\text{OpenSeed}}$ | 0.086 ± 0.042 | 0.434 ± 0.211 | 0.104 ± 0.051 |
| 5 | Human (La Rosa et al., 2023) | 0.050 ± 0.038 | 0.308 ± 0.246 | 0.068 ± 0.057 |
| | Closed (Bau et al., 2020) | 0.048 ± 0.029 | 0.277 ± 0.239 | 0.068 ± 0.045 |
| | Ours$_{\text{MasQCLIP}}$ | 0.045 ± 0.028 | 0.290 ± 0.188 | 0.057 ± 0.037 |
| | Ours$_{\text{SCAN}}$ | 0.045 ± 0.031 | 0.333 ± 0.271 | 0.057 ± 0.043 |
| | Ours$_{\text{SED}}$ | 0.044 ± 0.029 | 0.352 ± 0.287 | 0.058 ± 0.044 |
| | Ours$_{\text{CAT-Seg}}$ | 0.045 ± 0.028 | 0.342 ± 0.280 | 0.060 ± 0.043 |
| | Ours$_{\text{OpenSeed}}$ | 0.043 ± 0.029 | 0.358 ± 0.283 | 0.055 ± 0.042 |

Table 9: Avg. and Std. Dev. scores for explanations associated with a model trained on the Place365 dataset using Pascal-Context with 459 labels as a probing dataset.

| Cluster | Method | IoU | ActCov | DetAcc |
|---------|--------|-----|--------|--------|
| 1 | Human (La Rosa et al., 2023) | 0.177 ± 0.011 | 0.247 ± 0.012 | 0.386 ± 0.035 |
| | Closed (Bau et al., 2020) | 0.188 ± 0.012 | 0.271 ± 0.014 | 0.383 ± 0.036 |
| | Ours$_{\text{MasQCLIP}}$ | 0.179 ± 0.012 | 0.255 ± 0.010 | 0.376 ± 0.037 |
| | Ours$_{\text{SCAN}}$ | 0.177 ± 0.012 | 0.249 ± 0.011 | 0.381 ± 0.038 |
| | Ours$_{\text{SED}}$ | 0.182 ± 0.012 | 0.259 ± 0.011 | 0.381 ± 0.038 |
| | Ours$_{\text{CAT-Seg}}$ | 0.184 ± 0.012 | 0.264 ± 0.013 | 0.380 ± 0.038 |
| | Ours$_{\text{OpenSeed}}$ | 0.193 ± 0.012 | 0.280 ± 0.014 | 0.383 ± 0.035 |
| 2 | Human (La Rosa et al., 2023) | 0.118 ± 0.011 | 0.233 ± 0.022 | 0.194 ± 0.024 |
| | Closed (Bau et al., 2020) | 0.119 ± 0.013 | 0.245 ± 0.029 | 0.190 ± 0.027 |
| | Ours$_{\text{MasQCLIP}}$ | 0.101 ± 0.013 | 0.220 ± 0.024 | 0.158 ± 0.024 |
| | Ours$_{\text{SCAN}}$ | 0.112 ± 0.013 | 0.217 ± 0.025 | 0.191 ± 0.029 |
| | Ours$_{\text{SED}}$ | 0.115 ± 0.013 | 0.228 ± 0.027 | 0.191 ± 0.028 |
| | Ours$_{\text{CAT-Seg}}$ | 0.117 ± 0.013 | 0.236 ± 0.028 | 0.192 ± 0.028 |
| | Ours$_{\text{OpenSeed}}$ | 0.121 ± 0.013 | 0.253 ± 0.031 | 0.192 ± 0.028 |
| 3 | Human (La Rosa et al., 2023) | 0.106 ± 0.018 | 0.220 ± 0.049 | 0.180 ± 0.047 |
| | Closed (Bau et al., 2020) | 0.105 ± 0.020 | 0.216 ± 0.055 | 0.182 ± 0.048 |
| | Ours$_{\text{MasQCLIP}}$ | 0.086 ± 0.019 | 0.158 ± 0.043 | 0.177 ± 0.059 |
| | Ours$_{\text{SCAN}}$ | 0.103 ± 0.019 | 0.204 ± 0.051 | 0.185 ± 0.048 |
| | Ours$_{\text{SED}}$ | 0.104 ± 0.020 | 0.212 ± 0.051 | 0.181 ± 0.046 |
| | Ours$_{\text{CAT-Seg}}$ | 0.105 ± 0.020 | 0.215 ± 0.051 | 0.181 ± 0.045 |
| | Ours$_{\text{OpenSeed}}$ | 0.106 ± 0.020 | 0.220 ± 0.055 | 0.181 ± 0.045 |
| 4 | Human (La Rosa et al., 2023) | 0.112 ± 0.055 | 0.251 ± 0.088 | 0.175 ± 0.087 |
| | Closed (Bau et al., 2020) | 0.113 ± 0.054 | 0.250 ± 0.095 | 0.177 ± 0.084 |
| | Ours$_{\text{MasQCLIP}}$ | 0.100 ± 0.051 | 0.189 ± 0.086 | 0.175 ± 0.083 |
| | Ours$_{\text{SCAN}}$ | 0.112 ± 0.055 | 0.241 ± 0.097 | 0.177 ± 0.084 |
| | Ours$_{\text{SED}}$ | 0.113 ± 0.056 | 0.246 ± 0.097 | 0.178 ± 0.088 |
| | Ours$_{\text{CAT-Seg}}$ | 0.113 ± 0.056 | 0.250 ± 0.096 | 0.177 ± 0.087 |
| | Ours$_{\text{OpenSeed}}$ | 0.113 ± 0.055 | 0.254 ± 0.096 | 0.177 ± 0.088 |
| 5 | Human (La Rosa et al., 2023) | 0.077 ± 0.055 | 0.280 ± 0.203 | 0.103 ± 0.063 |
| | Closed (Bau et al., 2020) | 0.077 ± 0.055 | 0.301 ± 0.195 | 0.102 ± 0.071 |
| | Ours$_{\text{MasQCLIP}}$ | 0.078 ± 0.052 | 0.233 ± 0.191 | 0.120 ± 0.065 |
| | Ours$_{\text{SCAN}}$ | 0.079 ± 0.055 | 0.286 ± 0.198 | 0.105 ± 0.068 |
| | Ours$_{\text{SED}}$ | 0.080 ± 0.056 | 0.279 ± 0.206 | 0.110 ± 0.069 |
| | Ours$_{\text{CAT-Seg}}$ | 0.079 ± 0.056 | 0.286 ± 0.205 | 0.109 ± 0.071 |
| | Ours$_{\text{OpenSeed}}$ | 0.078 ± 0.056 | 0.290 ± 0.209 | 0.108 ± 0.074 |

Table 10: Avg. and Std. Dev. scores for explanations associated with a model trained on the Place365 dataset using COCO-Stuff as a probing dataset.

| Cluster | Method | IoU | ActCov | DetAcc |
|---|---|---|---|---|
| 1 | Human (La Rosa et al., 2023) | 0.152 ± 0.008 | 0.205 ± 0.009 | 0.374 ± 0.032 |
| | Closed (Bau et al., 2020) | 0.187 ± 0.010 | 0.270 ± 0.009 | 0.377 ± 0.032 |
| | $\text{Ours}_{\text{MasQCLIP}}$ | 0.107 ± 0.005 | 0.129 ± 0.005 | 0.384 ± 0.032 |
| | $\text{Ours}_{\text{SCAN}}$ | 0.145 ± 0.008 | 0.191 ± 0.009 | 0.376 ± 0.031 |
| | $\text{Ours}_{\text{SED}}$ | 0.165 ± 0.009 | 0.228 ± 0.010 | 0.373 ± 0.032 |
| | $\text{Ours}_{\text{CAT-Seg}}$ | 0.164 ± 0.009 | 0.228 ± 0.009 | 0.372 ± 0.032 |
| | $\text{Ours}_{\text{OpenSeed}}$ | 0.187 ± 0.010 | 0.270 ± 0.010 | 0.378 ± 0.032 |
| 2 | Human (La Rosa et al., 2023) | 0.104 ± 0.010 | 0.189 ± 0.020 | 0.188 ± 0.021 |
| | Closed (Bau et al., 2020) | 0.117 ± 0.012 | 0.243 ± 0.029 | 0.186 ± 0.021 |
| | $\text{Ours}_{\text{MasQCLIP}}$ | 0.078 ± 0.008 | 0.119 ± 0.011 | 0.185 ± 0.027 |
| | $\text{Ours}_{\text{SCAN}}$ | 0.103 ± 0.010 | 0.185 ± 0.017 | 0.188 ± 0.021 |
| | $\text{Ours}_{\text{SED}}$ | 0.111 ± 0.011 | 0.215 ± 0.021 | 0.189 ± 0.022 |
| | $\text{Ours}_{\text{CAT-Seg}}$ | 0.111 ± 0.011 | 0.213 ± 0.022 | 0.189 ± 0.021 |
| | $\text{Ours}_{\text{OpenSeed}}$ | 0.117 ± 0.012 | 0.242 ± 0.029 | 0.186 ± 0.022 |
| 3 | Human (La Rosa et al., 2023) | 0.090 ± 0.019 | 0.182 ± 0.046 | 0.160 ± 0.045 |
| | Closed (Bau et al., 2020) | 0.098 ± 0.022 | 0.220 ± 0.058 | 0.158 ± 0.045 |
| | $\text{Ours}_{\text{MasQCLIP}}$ | 0.074 ± 0.016 | 0.119 ± 0.023 | 0.169 ± 0.045 |
| | $\text{Ours}_{\text{SCAN}}$ | 0.090 ± 0.018 | 0.180 ± 0.047 | 0.164 ± 0.047 |
| | $\text{Ours}_{\text{SED}}$ | 0.095 ± 0.020 | 0.203 ± 0.052 | 0.161 ± 0.046 |
| | $\text{Ours}_{\text{CAT-Seg}}$ | 0.095 ± 0.020 | 0.201 ± 0.052 | 0.162 ± 0.047 |
| | $\text{Ours}_{\text{OpenSeed}}$ | 0.097 ± 0.021 | 0.217 ± 0.056 | 0.159 ± 0.046 |
| 4 | Human (La Rosa et al., 2023) | 0.089 ± 0.037 | 0.185 ± 0.084 | 0.166 ± 0.070 |
| | Closed (Bau et al., 2020) | 0.094 ± 0.038 | 0.213 ± 0.087 | 0.160 ± 0.069 |
| | $\text{Ours}_{\text{MasQCLIP}}$ | 0.086 ± 0.035 | 0.150 ± 0.065 | 0.174 ± 0.065 |
| | $\text{Ours}_{\text{SCAN}}$ | 0.093 ± 0.038 | 0.188 ± 0.082 | 0.172 ± 0.071 |
| | $\text{Ours}_{\text{SED}}$ | 0.093 ± 0.038 | 0.203 ± 0.084 | 0.164 ± 0.073 |
| | $\text{Ours}_{\text{CAT-Seg}}$ | 0.093 ± 0.038 | 0.201 ± 0.085 | 0.165 ± 0.071 |
| | $\text{Ours}_{\text{OpenSeed}}$ | 0.094 ± 0.038 | 0.211 ± 0.089 | 0.161 ± 0.070 |
| 5 | Human (La Rosa et al., 2023) | 0.078 ± 0.048 | 0.197 ± 0.135 | 0.127 ± 0.071 |
| | Closed (Bau et al., 2020) | 0.078 ± 0.047 | 0.225 ± 0.148 | 0.120 ± 0.068 |
| | $\text{Ours}_{\text{MasQCLIP}}$ | 0.078 ± 0.040 | 0.200 ± 0.111 | 0.119 ± 0.058 |
| | $\text{Ours}_{\text{SCAN}}$ | 0.080 ± 0.046 | 0.220 ± 0.132 | 0.123 ± 0.068 |
| | $\text{Ours}_{\text{SED}}$ | 0.080 ± 0.047 | 0.209 ± 0.129 | 0.125 ± 0.068 |
| | $\text{Ours}_{\text{CAT-Seg}}$ | 0.080 ± 0.047 | 0.221 ± 0.141 | 0.123 ± 0.068 |
| | $\text{Ours}_{\text{OpenSeed}}$ | 0.079 ± 0.048 | 0.221 ± 0.142 | 0.122 ± 0.071 |

Table 11: Avg. scores for explanations associated with a model trained on the Place365 dataset using VOC2012 as a probing dataset.

| Cluster | Method | IoU | | ActCov | | DetAcc | |
|---------|--------|-----|-----|--------|-----|--------|-----|
| 1 | Human (La Rosa et al., 2023) | 0.077 | ± 0.007 | 0.090 | ± 0.008 | 0.362 | ± 0.040 |
| | Closed (Bau et al., 2020) | 0.193 | ± 0.014 | 0.280 | ± 0.014 | 0.385 | ± 0.039 |
| | Ours$_{\text{MasQCLIP}}$ | 0.269 | ± 0.022 | 0.499 | ± 0.015 | 0.369 | ± 0.038 |
| | Ours$_{\text{SCAN}}$ | 0.209 | ± 0.014 | 0.324 | ± 0.015 | 0.371 | ± 0.036 |
| | Ours$_{\text{SED}}$ | 0.238 | ± 0.018 | 0.407 | ± 0.015 | 0.366 | ± 0.038 |
| | Ours$_{\text{CAT-Seg}}$ | 0.189 | ± 0.011 | 0.277 | ± 0.009 | 0.377 | ± 0.036 |
| | Ours$_{\text{OpenSeed}}$ | 0.276 | ± 0.022 | 0.522 | ± 0.016 | 0.370 | ± 0.036 |
| 2 | Human (La Rosa et al., 2023) | 0.074 | ± 0.010 | 0.107 | ± 0.014 | 0.199 | ± 0.029 |
| | Closed (Bau et al., 2020) | 0.115 | ± 0.014 | 0.232 | ± 0.034 | 0.190 | ± 0.030 |
| | Ours$_{\text{MasQCLIP}}$ | 0.132 | ± 0.015 | 0.438 | ± 0.034 | 0.160 | ± 0.020 |
| | Ours$_{\text{SCAN}}$ | 0.132 | ± 0.014 | 0.296 | ± 0.038 | 0.195 | ± 0.028 |
| | Ours$_{\text{SED}}$ | 0.134 | ± 0.014 | 0.335 | ± 0.051 | 0.185 | ± 0.023 |
| | Ours$_{\text{CAT-Seg}}$ | 0.131 | ± 0.014 | 0.282 | ± 0.033 | 0.201 | ± 0.033 |
| | Ours$_{\text{OpenSeed}}$ | 0.137 | ± 0.016 | 0.479 | ± 0.042 | 0.162 | ± 0.021 |
| 3 | Human (La Rosa et al., 2023) | 0.082 | ± 0.018 | 0.134 | ± 0.031 | 0.184 | ± 0.051 |
| | Closed (Bau et al., 2020) | 0.105 | ± 0.021 | 0.201 | ± 0.051 | 0.191 | ± 0.046 |
| | Ours$_{\text{MasQCLIP}}$ | 0.107 | ± 0.020 | 0.272 | ± 0.100 | 0.166 | ± 0.047 |
| | Ours$_{\text{SCAN}}$ | 0.117 | ± 0.021 | 0.290 | ± 0.060 | 0.169 | ± 0.037 |
| | Ours$_{\text{SED}}$ | 0.115 | ± 0.021 | 0.294 | ± 0.066 | 0.164 | ± 0.037 |
| | Ours$_{\text{CAT-Seg}}$ | 0.117 | ± 0.022 | 0.305 | ± 0.051 | 0.163 | ± 0.038 |
| | Ours$_{\text{OpenSeed}}$ | 0.110 | ± 0.020 | 0.283 | ± 0.100 | 0.164 | ± 0.040 |
| 4 | Human (La Rosa et al., 2023) | 0.098 | ± 0.050 | 0.183 | ± 0.076 | 0.186 | ± 0.095 |
| | Closed (Bau et al., 2020) | 0.117 | ± 0.051 | 0.248 | ± 0.089 | 0.188 | ± 0.082 |
| | Ours$_{\text{MasQCLIP}}$ | 0.104 | ± 0.049 | 0.272 | ± 0.104 | 0.146 | ± 0.069 |
| | Ours$_{\text{SCAN}}$ | 0.103 | ± 0.042 | 0.327 | ± 0.103 | 0.136 | ± 0.062 |
| | Ours$_{\text{SED}}$ | 0.100 | ± 0.041 | 0.327 | ± 0.103 | 0.128 | ± 0.054 |
| | Ours$_{\text{CAT-Seg}}$ | 0.099 | ± 0.043 | 0.322 | ± 0.108 | 0.130 | ± 0.059 |
| | Ours$_{\text{OpenSeed}}$ | 0.101 | ± 0.044 | 0.288 | ± 0.102 | 0.138 | ± 0.065 |
| 5 | Human (La Rosa et al., 2023) | 0.077 | ± 0.062 | 0.247 | ± 0.173 | 0.106 | ± 0.078 |
| | Closed (Bau et al., 2020) | 0.084 | ± 0.053 | 0.296 | ± 0.177 | 0.114 | ± 0.068 |
| | Ours$_{\text{MasQCLIP}}$ | 0.058 | ± 0.040 | 0.322 | ± 0.212 | 0.067 | ± 0.045 |
| | Ours$_{\text{SCAN}}$ | 0.055 | ± 0.039 | 0.343 | ± 0.202 | 0.064 | ± 0.046 |
| | Ours$_{\text{SED}}$ | 0.051 | ± 0.032 | 0.370 | ± 0.210 | 0.058 | ± 0.038 |
| | Ours$_{\text{CAT-Seg}}$ | 0.051 | ± 0.030 | 0.357 | ± 0.219 | 0.058 | ± 0.034 |
| | Ours$_{\text{OpenSeed}}$ | 0.057 | ± 0.037 | 0.282 | ± 0.199 | 0.070 | ± 0.044 |

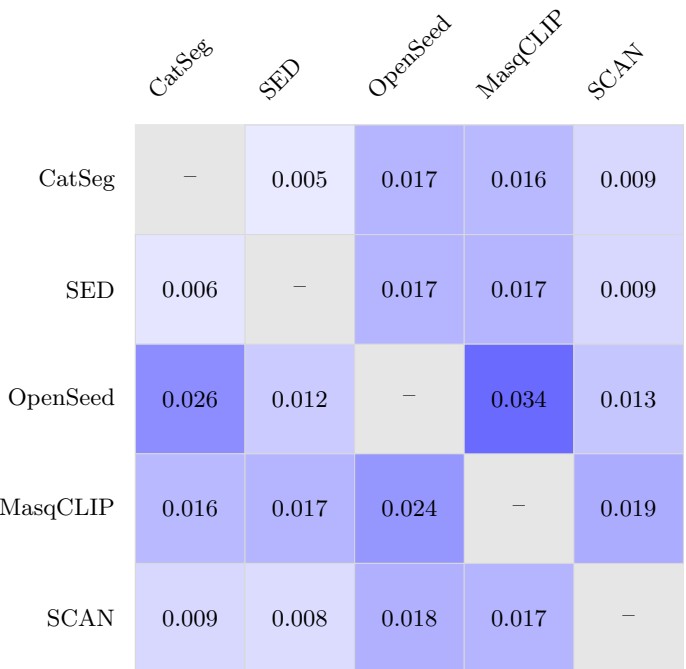

Figure 6: Percentage of unrelated concepts per explanation between explanations computed over data generated by segmentor $A$ (row) and explanation computed over data generated by segmentor $B$.

Figure 7: Average alignment difference between the best explanation computed over data generated by segmentor $A$ (row) and explanation computed over data generated by segmentor $B$, when explanations differ.

Table 12: Avg. and Std. Dev. scores for explanations associated with a DenseNet161 model trained on the Place365 dataset using Ade20K as a probing dataset.

| Cluster | Method | IoU | ActCov | DetAcc |
|---|---|---|---|---|
| 1 | Human (La Rosa et al., 2023) | 0.128 ± 0.048 | 0.296 ± 0.120 | 0.198 ± 0.072 |
|  | Closed (Bau et al., 2020) | 0.134 ± 0.049 | 0.304 ± 0.119 | 0.205 ± 0.067 |
|  | Ours | 0.133 ± 0.047 | 0.294 ± 0.119 | 0.208 ± 0.068 |
| 2 | Human (La Rosa et al., 2023) | 0.209 ± 0.041 | 0.345 ± 0.040 | 0.361 ± 0.108 |
|  | Closed (Bau et al., 2020) | 0.205 ± 0.039 | 0.334 ± 0.047 | 0.360 ± 0.100 |
|  | Ours | 0.204 ± 0.039 | 0.325 ± 0.045 | 0.369 ± 0.107 |
| 3 | Human (La Rosa et al., 2023) | 0.207 ± 0.026 | 0.345 ± 0.024 | 0.344 ± 0.061 |
|  | Closed (Bau et al., 2020) | 0.204 ± 0.026 | 0.336 ± 0.023 | 0.346 ± 0.062 |
|  | Ours | 0.201 ± 0.026 | 0.322 ± 0.025 | 0.353 ± 0.061 |
| 4 | Human (La Rosa et al., 2023) | 0.177 ± 0.058 | 0.325 ± 0.080 | 0.290 ± 0.102 |
|  | Closed (Bau et al., 2020) | 0.175 ± 0.056 | 0.320 ± 0.075 | 0.286 ± 0.099 |
|  | Ours | 0.178 ± 0.057 | 0.315 ± 0.072 | 0.299 ± 0.104 |
| 5 | Human (La Rosa et al., 2023) | 0.103 ± 0.047 | 0.274 ± 0.118 | 0.158 ± 0.086 |
|  | Closed (Bau et al., 2020) | 0.108 ± 0.048 | 0.287 ± 0.119 | 0.158 ± 0.075 |
|  | Ours | 0.108 ± 0.048 | 0.286 ± 0.116 | 0.160 ± 0.082 |

Table 13: Avg. and Std. Dev. scores for explanations associated with an AlexNet model trained on the Place365 dataset using Ade20K as a probing dataset.

| Cluster | Method | IoU | ActCov | DetAcc |
|---|---|---|---|---|
| 1 | Human (La Rosa et al., 2023) | 0.192 ± 0.024 | 0.333 ± 0.024 | 0.314 ± 0.053 |
|  | Closed (Bau et al., 2020) | 0.188 ± 0.023 | 0.322 ± 0.029 | 0.314 ± 0.050 |
|  | Ours | 0.184 ± 0.022 | 0.309 ± 0.020 | 0.317 ± 0.054 |
| 2 | Human (La Rosa et al., 2023) | 0.115 ± 0.026 | 0.300 ± 0.078 | 0.161 ± 0.035 |
|  | Closed (Bau et al., 2020) | 0.117 ± 0.025 | 0.292 ± 0.065 | 0.167 ± 0.038 |
|  | Ours | 0.117 ± 0.025 | 0.287 ± 0.075 | 0.169 ± 0.035 |
| 3 | Human (La Rosa et al., 2023) | 0.097 ± 0.028 | 0.262 ± 0.100 | 0.142 ± 0.043 |
|  | Closed (Bau et al., 2020) | 0.101 ± 0.029 | 0.277 ± 0.091 | 0.143 ± 0.042 |
|  | Ours | 0.102 ± 0.029 | 0.272 ± 0.099 | 0.145 ± 0.040 |
| 4 | Human (La Rosa et al., 2023) | 0.079 ± 0.028 | 0.233 ± 0.126 | 0.120 ± 0.039 |
|  | Closed (Bau et al., 2020) | 0.082 ± 0.028 | 0.265 ± 0.134 | 0.117 ± 0.038 |
|  | Ours | 0.082 ± 0.028 | 0.251 ± 0.127 | 0.121 ± 0.039 |
| 5 | Human (La Rosa et al., 2023) | 0.055 ± 0.028 | 0.226 ± 0.191 | 0.093 ± 0.064 |
|  | Closed (Bau et al., 2020) | 0.054 ± 0.023 | 0.254 ± 0.186 | 0.080 ± 0.049 |
|  | Ours | 0.059 ± 0.026 | 0.245 ± 0.183 | 0.094 ± 0.057 |

## A.2 Additional Probed Models

In this section, we report the results of explaining different probed models. Following Mu & Andreas (2020), we compute explanation scores for DenseNet161 (Huang et al., 2017) and AlexNet (Krizhevsky et al., 2012) pre-trained on the Place365 dataset (Zhou et al., 2017a). We report the results for our framework using the same configuration as in the main text, the "*human*" baseline, and the closed approach (Bau et al., 2020).

Specifically, we randomly extract 50 neurons for each probed model and we generate explanations for those neurons using the validation split of Ade20K (Zhou et al., 2017b) as a probing dataset. Tables 12 and 13

Table 14: Avg. and Std. Dev. scores for explanations associated with the highest activations of models pre-trained on ImageNet and using Ade20K as a probing dataset.

| Model | Method | IoU | ActCov | DetAcc |
|-------|--------|-----|--------|--------|
| ConvNext | Human (La Rosa et al., 2023) | 0.041 ± 0.019 | 0.076 ± 0.033 | 0.104 ± 0.071 |
|  | Closed (Bau et al., 2020) | 0.034 ± 0.012 | 0.085 ± 0.061 | 0.071 ± 0.039 |
|  | Ours | 0.048 ± 0.022 | 0.084 ± 0.033 | 0.111 ± 0.078 |
| EfficientVit | Human (La Rosa et al., 2023) | 0.045 ± 0.019 | 0.116 ± 0.085 | 0.095 ± 0.076 |
|  | Closed (Bau et al., 2020) | 0.036 ± 0.015 | 0.154 ± 0.111 | 0.056 ± 0.032 |
|  | Ours | 0.046 ± 0.019 | 0.133 ± 0.102 | 0.094 ± 0.08 |
| MaxVit | Human (La Rosa et al., 2023) | 0.027 ± 0.016 | 0.107 ± 0.089 | 0.043 ± 0.024 |
|  | Closed (Bau et al., 2020) | 0.024 ± 0.013 | 0.126 ± 0.094 | 0.036 ± 0.021 |
|  | Ours | 0.027 ± 0.014 | 0.124 ± 0.095 | 0.041 ± 0.023 |
| CvT | Human (La Rosa et al., 2023) | 0.015 ± 0.01 | 0.063 ± 0.072 | 0.024 ± 0.019 |
|  | Closed (Bau et al., 2020) | 0.013 ± 0.01 | 0.084 ± 0.078 | 0.02 ± 0.017 |
|  | Ours | 0.015 ± 0.01 | 0.063 ± 0.073 | 0.025 ± 0.018 |

confirm the comparable performance of the framework with respect to the competitors, making the insights independent of the probed model in use.

As a proof of concept, we also test our framework in non-traditional settings for compositional explanations in Table 14. This table includes modern networks pre-trained on ImageNet and included in the family of Transformer models (with the exception of ConvNeXt, which borrows several design choices from them). Namely, we consider as probed models: ConvNeXt (Liu et al., 2022), EfficientVit (Cai et al., 2023), MaxVit (Tu et al., 2022), and CvT (Wu et al., 2021). In this case, we report the scores computed over the highest activations (i.e., top 0.01 quantile of activations extracted from the convolutional layer in the final block or stage) rather than for all the clusters. This choice is motivated by the fact that, in these settings, five clusters appear to be too large, and the explanations converge towards default rules, similar to what has been observed in La Rosa et al. (2023). Conversely, the threshold adopted by Mu & Andreas (2020); Bau et al. (2020) (top 0.005 quantile) appears to be too small, leading to explanations associated with an exceptionally small IoU score. Through preliminary experimentation, we found that the selected threshold provides a variability of explanations and IoU scores that is more consistent with the behavior observed in established compositional-explanation settings.

From Table 14, we observe differences in performance between the baselines and our framework consistent with those observed in more established settings, representing further evidence that the performance of the framework is independent from the specific probed model. At the same time, we can note that in these cases the general scores are slightly lower than in established settings, suggesting that future research is needed in this direction to identify better setups for these families of architectures. We hypothesize this difference may be related to the necessity of finding better hyperparameters for extracting semantic activation range, to the potential weaker link between the trained data and the probed concepts, and to the fact that these architectures are designed to exploit global information, and thus they may be partially misaligned in principle with the scope of compositional explanations.

### A.3 Metrics Details and Additional Metrics

This section provides the formalization of Detection Accuracy and Activation Coverage and introduces and compares the competitors using two additional metrics: Sample Coverage and Explanation Coverage. We chose these metrics because they have been used by previous literature to study cluster-level explanations (La Rosa et al., 2023) and allow us to perform a pixel-level comparison of the different segmentation masks produced by different competitors.

We use the same notation introduced in Section 3. However, because Sample Coverage and Explanation Coverage are computed per sample, we need to introduce an additional notation. Namely, we use $\mathbb{M}^x$ to indicate the set of binarized segmentation masks associated with the sample $x$ and $\mathbb{A}^x$ to indicate the set of binarized activations associated with the sample $x$.

**Detection Accuracy** quantifies the percentage of label annotations recognized within the activation range. A high value indicates that most of the label's masks are detected by the neuron using the given activation range.

$$DetAcc(L, \mathbb{A}, \mathbb{M}) = \frac{|\mathbb{A} \cap \theta(\mathbb{M}, L)|}{|\theta(\mathbb{M}, L)|} \tag{12}$$

**Activation Coverage** measures the percentage of neuron activations within the annotated label regions. A high value indicates that the label "dominates" large parts of the activation range (i.e., there is a strong mapping).

$$ActCov(L, \mathbb{A}, \mathbb{M}) = \frac{|\mathbb{A} \cap \theta(\mathbb{M}, L)|}{|\mathbb{A}|} \tag{13}$$

**Samples Coverage** calculates the ratio of samples in the probing dataset that are captured by the explanation and where the neuron activation falls within the activation range:

$$SampleCov(L, \mathbb{A}, \mathbb{M}, \mathfrak{D}) = \frac{|\{x \in \mathfrak{D} : |\mathbb{A}^x \cap \theta(\mathbb{M}^x, L)| > 0\}|}{|\{x \in \mathfrak{D} : |\theta(\mathbb{M}^x, L)| > 0\}|} \tag{14}$$

**Explanation Coverage** calculates the ratio of samples in the probing dataset that are captured by the explanation and where the neuron activation falls within the activation range and the total number of samples where the neuron activation falls within the activation range.

$$ExplCov(L, \mathbb{A}, \mathbb{M}, \mathfrak{D}) = \frac{|\{x \in \mathfrak{D} : |\mathbb{A}^x \cap \theta(\mathbb{M}^x, L)| > 0\}|}{|\{x \in \mathfrak{D} : |\mathbb{A}^x| > 0\}|} \tag{15}$$

**Results** As shown in Table 15, the results for the additional metrics are similar to the ones reported in the main text for the other metrics. Thus, considering the large standard deviation of these metrics, the results can be considered comparable.

## B Limitations

While, as shown in the previous sections, the framework is flexible and competitive across several settings, we identified several limitations that can serve as a base for future research on both open vocabulary semantic segmentation and explainability.

**Number of Concepts.** The number of concepts that can be tested is constrained by the available memory. Ideally, we would like to evaluate every possible concept in a vocabulary (e.g., the most common 10,000 words in English). However, in practice, the output of segmentation models is a matrix $s_x \in R^{|C_i|, h, w}$, where the first dimension represents the logits (or output probabilities) of all the concepts in the given concept (sub)set. Although, as explained in Section 3, the first dimension can later be reduced by considering only the maximum value as the model's prediction, this matrix still needs to be loaded into memory, even if only temporarily. Consequently, the maximum number of concepts that can be used in explanations is limited and influenced by the available memory on the workstation and the resolution of the segmentation masks.

**Completeness of the Concept Subset** One of the limitations of the current framework is its sensitivity to the completeness of each concept subset. Since the open vocabulary segmentation model is "forced" to

Table 15: Avg. and Std. Dev Sample Coverage and Explanation Coverage for explanations associated with a model trained on the Place365 dataset using Ade20K as a probing dataset.

| Cluster | Method | SampleCov | ExplCov |
|---|---|---|---|
| 1 | Human (La Rosa et al., 2023) | 0.911 ± 0.029 | 0.873 ± 0.059 |
|   | Closed (Bau et al., 2020) | 0.904 ± 0.028 | 0.872 ± 0.078 |
|   | Ours | 0.899 ± 0.030 | 0.855 ± 0.072 |
| 2 | Human (La Rosa et al., 2023) | 0.766 ± 0.065 | 0.693 ± 0.127 |
|   | Closed (Bau et al., 2020) | 0.743 ± 0.057 | 0.690 ± 0.136 |
|   | Ours | 0.752 ± 0.072 | 0.667 ± 0.126 |
| 3 | Human (La Rosa et al., 2023) | 0.559 ± 0.103 | 0.538 ± 0.144 |
|   | Closed (Bau et al., 2020) | 0.537 ± 0.094 | 0.540 ± 0.122 |
|   | Ours | 0.549 ± 0.103 | 0.522 ± 0.128 |
| 4 | Human (La Rosa et al., 2023) | 0.380 ± 0.129 | 0.411 ± 0.186 |
|   | Closed (Bau et al., 2020) | 0.342 ± 0.112 | 0.441 ± 0.173 |
|   | Ours | 0.369 ± 0.122 | 0.417 ± 0.179 |
| 5 | Human (La Rosa et al., 2023) | 0.246 ± 0.151 | 0.285 ± 0.202 |
|   | Closed (Bau et al., 2020) | 0.174 ± 0.101 | 0.343 ± 0.211 |
|   | Ours | 0.212 ± 0.121 | 0.311 ± 0.200 |

assign at least one concept to every pixel, the concept subset must be as complete as possible to account for all the possible concepts in the input. When an input element cannot be described by using the concepts in the concept subset, that element leads to hallucinations by the segmentation model. Such hallucinations impact the explanation quality of the wrongly assigned concept, potentially triggering a cascade effect. While this issue can be mitigated by including generic concepts (i.e., "background", "thing", or "other") into the concept subset, their effectiveness depends on the training recipe used to pre-train the backbone models (e.g., whether a background class was included in the training). To address this limitation, future work could explore adaptive mechanisms to filter out unreliable masks, possibly arising from hallucinations, thereby reducing such sensitivity.

**Sensitivity to the Concept Subset** The selection of concepts within a generic concept subset can also affect both the quality of the computed explanations and the performance of the framework itself. While it is desirable to have multiple granularities across different concept subsets, including multiple granularities within a single subset could potentially cause inconsistency in explanations. For example, if both the "*animal*" and "*cat*" concepts are included in the same subset, the model is forced to choose between them when segmenting a cat, even though both could be considered correct. In these cases, the choice will depend entirely on the biases learned from the training dataset and labels used to train the segmentation model or the multi-modal model. To mitigate this issue, we recommend separating concepts with different levels of granularity into different concept subsets, ensuring that two concepts within the same subset cannot be used to describe the same element. We leave for future research the development of an algorithm that can navigate and mitigate this sensitivity.

**Dependence on Prompt Templates** One limitation associated with research in open vocabulary semantic segmentation is its reliance on prompt templates. Most of the analyzed models fine-tune the multi-modal backbone using fixed prompt templates (e.g., "*a photo of a {}*"). These prompts are typically designed for the semantic segmentation task, which focuses on objects and tangible elements (e.g., sky, tree). Once the model has been fine-tuned, the number of templates is fixed, and replacing some of them can lead to out-of-distribution issues. This lack of flexibility reduces the models' effectiveness in recognizing abstract concepts (e.g., patterns) due to the resulting unnatural descriptions and the impossibility of introducing new prompts. The only mitigation could involve additional fine-tuning of the multi-modal model for the explainability task. We call for further research in this direction to make these models more adaptable during

inference and to support greater variability in prompt templates during fine-tuning, especially to account for downstream tasks such as explainability.

**Refinements' Cascade Effect**  While this is not strictly a limitation of the framework, we want to emphasize and make the reader aware of the potential cascade effect when applying refinements to the concept set. As explained in the previous sections, users can modify the concept set after analyzing explanations to retrieve potentially improved explanations based on the refined set. However, when making such refinements, it is **important to re-generate the masks** for the subsets where the new concepts are introduced. Indeed, adding a concept to the subset changes the output size of the segmentation model and, consequently, its output distribution. Therefore, this adjustment can alter predictions, particularly the most uncertain ones, for all concepts in the concept subset, even those unrelated to the newly added concepts. At the explanation level, the empirical experiments reported in this paper did not reveal significant changes in explanations. The only difference we observe is in the selection of concepts used to exclude portions of the dataset (e.g., Cat AND NOT Car). These concepts are used by the compositional explanation algorithm to exclude edge cases of neuron behavior. In this case, multiple choices led to similar outcomes, explaining the differences. However, we expect that if the newly added concepts substantially improve the coverage of the concept subset or better align its granularity with the recognition capabilities of the backbone model, this could potentially result in more significant shifts in explanations, which should be monitored.

**Applicability**  This framework shares the same assumptions, limitations, and applicability conditions as existing methods in the compositional explanations literature. As discussed in Section 3, compositional explanations require a projection mechanism that maps neuron activations into the annotation space. To date, compositional explanations have primarily been explored in settings involving unidimensional activations and annotations (e.g., NLP (Mu & Andreas, 2020) and Deep Reinforcement Learning (Jiang et al., 2025)) and bidimensional activations and annotations (e.g., vision settings (Bau et al., 2020; Massidda & Bacciu, 2023; Makinwa et al., 2022; Mu & Andreas, 2020; La Rosa et al., 2023)). Applying the framework to vision settings where activations are not bidimensional (e.g., MLP layers) requires advances in projection mechanisms capable of mapping those activations into the annotation space. The same challenge arises for activations that are bidimensional, or can be reshaped into a bidimensional representation, but are not explicitly designed to capture spatial information. In such cases, the framework remains applicable in principle (e.g., by leveraging attention maps), but both the semantics and the scale of the resulting activations differ substantially from the traditional settings of compositional explanations. Consequently, we argue that future research should first focus on identifying appropriate projection mechanisms and experimental settings for these architectures before applying compositional explanations and, consequently, our framework.

## C  Clustered Compositional Explanations Algorithm

Let $\mathbb{A}$ be a binary activation matrix, $\mathbb{C}$ be a set of concepts, $\mathbb{M}$ be a set of binary segmentation masks, one for each concept, and $\mathfrak{L}^n$ be the set of all possible logical connections of arity at maximum $n$ between concepts in the concept set $\mathbb{C}$. These quantities are computed as described in Section 3 of the main text. The goal of compositional explanation algorithms is to find the label $L \in \mathfrak{L}^n$ whose mask maximally overlaps with the neuron binary activations $\mathbb{A}$. Formally, these algorithms find the solution for the following objective:

$$\underset{L \in \mathfrak{L}^n}{\arg\max} \, IoU(L, \mathbb{A}, \mathbb{M}) \tag{16}$$

$IoU$ is defined as:

$$IoU(L, \mathbb{A}, \mathbb{M}) = \frac{|\mathbb{A} \cap \theta(\mathbb{M}, L)|}{|\mathbb{A} \cup \theta(\mathbb{M}, L)|} \tag{17}$$

and $\theta(\mathbb{M}, L)$ is a function that returns the logical combination of the masks in $\mathbb{M}$ of the concepts involved in the label $L$.

Exhaustive search over $\mathfrak{L}^n$ is computationally infeasible in most of the settings commonly considered in literature. To address this problem, Mu & Andreas (2020) propose to use beam search in place of exhaustive

---

**Algorithm 1: Beam Search Guided by MMESH**

---

**Input:** $\mathbb{C}$, $\mathbb{M}$, $\mathbb{A}$, MMESHInfo, b, length
**Output:** BestLabel,BestIoU
Beam $\leftarrow$ empty list
UpdatedInfo $\leftarrow$ MMESHInfo
**for** $c_{k,i}$ **in** $\mathbb{C}$ **do**
  Iou $\leftarrow$ `compute_iou`$(c_{k,i}, \mathbb{M}, \mathbb{A})$
  Beam.`add`$(label = c_{k,i}, iou = Iou)$
**end**
`sort`$(Beam)$   # Sort by IoU
# Select the best b candidates
Beam $\leftarrow$ Beam[:b]
MinIoU $\leftarrow$ `find_min`$(Beam)$
**for** _2_ **to** length **do**
  SearchSpace $\leftarrow$ `expand_beam`$(Beam, \mathbb{C})$
  Estimations $\leftarrow$ `estimate_iou`$(SearchSpace, MMESHInfo)$
  `sort`$(Estimations)$
  **for** _L,_ EstIoU **in** Estimations **do**
    **if** EstIoU $<$ MinIoU **then**
      # All the other labels cannot be added to the beam
      **break**
    **end**
    Iou $\leftarrow$ `compute_iou`$(L, \mathbb{M}, \mathbb{A})$
    Beam.`add`$((label=L, iou=Iou))$
  **end**
  `sort`$(Beam)$
  # Select the best b candidates
  Beam $\leftarrow$ Beam[:b]
  # Compute and update info
  MinIoU $\leftarrow$ `find_min`$(Beam)$
  MMESHInfo $\leftarrow$ `update_info`$(MMESHInfo, Beam)$
**end**
BestLabel, BestIoU $\leftarrow$ `max`$(Beam)$
**return** _BestLabel, BestIoU_

---

search. This algorithm has been extended by La Rosa et al. (2023) to speed up the computation of explanation using a beam search guided by the *Min-Max Extension per Sample Heuristic* (MMESH).

While we refer the reader to Mu & Andreas (2020) and La Rosa et al. (2023) for full details of the algorithm and its procedures, we briefly outline its main steps and components below.

The pseudocode is shown in Algorithm 1. At each step $i$, the algorithm maintains a beam of $b$ candidate explanations, selected based on the highest IoU scores from the previous step. From this beam, it generates a search space by combining the beam labels with the concepts in the concept set $\mathbb{C}$. The combinations are based on the bitwise propositional logic operators AND, OR, and AND NOT. For each candidate in this search space, the algorithm estimates the IoU using precomputed heuristic information. The candidates are then sorted based on these estimated scores. At this point, the algorithm computes the IoU for the candidates associated with an estimate IoU greater than the current beam minimum, and the $b$ candidates with the highest IoU are retained as the beam for the next step $i + 1$. This process is repeated until the maximum allowed explanation length is reached. Finally, the algorithm returns the explanation that achieved the highest IoU across all steps.

**Estimating IoU** For each sample and each concept, MMESH computes both the bounding boxes and the inscribed rectangles within the concept regions. This geometric information is then combined with concept sizes to estimate the IoU of a given label $L$.

In formulas:

$$\widehat{IoU}(L, \mathbb{A}, \mathbb{M}, \mathfrak{D}) = \frac{\widehat{I}}{\widehat{U}} = \frac{\sum_{x \in \mathfrak{D}} \widehat{I^x}}{\sum_{x \in \mathfrak{D}} \widehat{U^x}} =$$

$$= \frac{\widehat{I_x}}{\sum_{x \in \mathfrak{D}} |\mathbb{A}| + \sum_{x \in \mathfrak{D}} |\widehat{\theta(\mathbb{M}^x, L)}| - \widehat{I_x}}$$

(18)

The specific computation of the estimate intersection $\widehat{I^x}$ and the estimated label mask $\widehat{\theta(\mathbb{M}^x, L)}$ depends on the logical operator connecting the left side $(L_\leftarrow)$ and right side $(L_\rightarrow)$ of the label. In all cases, $\widehat{I^x}$ is an overestimation of the actual intersection and $\widehat{\theta(\mathbb{M}^x, L)}$ is an underestimation of the actual label mask. These conservative estimations ensure that the algorithm finds the optimal solution within the beam. Specifically:

**OR**

$$\widehat{I^x} = min(|IMS(x, L_\leftarrow)| + |IMS(x, L_\rightarrow)|, |M(x)|)$$

(19)

$$\widehat{\theta(\mathbb{M}^x, L)} = max(|\theta(M^x, L_\leftarrow)|,$$
$$|\theta(M^x, L_\rightarrow)|,$$
$$\theta(\widehat{M^x, L_\leftarrow \cup L_\rightarrow}))$$

(20)

**AND**

$$\widehat{I^x} = min(|IMS(x, L_\leftarrow)|, |IMS(x, L_\rightarrow)|)$$

(21)

$$\widehat{\theta(\mathbb{M}^x, L)} = max(MinOver(L), I_x)$$

(22)

**AND NOT**

$$\widehat{I^x} = min(|IMS(x, L_\leftarrow)|, |\mathbb{M}^x| - |IMS(x, L_\rightarrow)|)$$

(23)

$$\widehat{\theta(\mathbb{M}^x, L)} = max(|\theta(\mathbb{A}^x, L_\leftarrow)| - MaxOver(L), I_x)$$

(24)

where: $\mathbb{M}^x$ and $\mathbb{A}^x$ are defined as in Section A.3, $IMS(x, L)$ denotes the intersection size between the label mask $\theta(M^x, L)$ and the neuron binary activation $\mathbb{A}^x$ computed a generic activation range $(\tau_1, \tau_2)$; $MaxOver(L)$ is a function that returns the maximum possible overlap between the bounding boxes associated with the left and right sides of $L$ in the sample $x$; and $MinOver(L)$ is a function that returns the minimum possible overlap between the inscribed rectangles sassociated with the left and right sides of $L$ in the sample $x$.

For a complete derivation of these estimations and proofs, we refer the reader to La Rosa et al. (2023).

# D  Concept Set for CUB

This section describes the concept set used for the experiments on the CUB dataset in the main text and discusses alternatives and challenges in the selection process for concept sets.

The concept set was chosen based on the availability of a list of relevant concepts for the task, specifically the categories used in the dataset's annotations. Note that we do not use the annotations themselves; the only relevant information is the list of concepts. This list includes bird species, as well as combinations of colors, shapes, and patterns associated with bird parts. After iterative refinements, the resulting concept set is divided into the following subsets:

1. Bird species (e.g., *black footed albatross*)

2. Element colors (e.g., bird colors like *blue bird* and background colors)

3. Bird shapes (e.g., *long-legged bird*)

4. Parts (e.g., *bird's wing*)

5. Colored parts (e.g., *blue bird's wing*)

6. Part shape (e.g., *curved bird's bill*)

7. Part patterns (e.g., *solid bird's breast*)

These subsets are further divided into three levels of granularity: the first includes bird species, bird shapes, and colors; the second includes parts; and the third includes all remaining subsets. We also include the set of concepts annotated in the Ade20k dataset as an additional subset. This decision allows the detection of neurons that capture individual background elements (e.g., water), potentially exploited by biases in the network, as well as neurons that generally recognize birds (recognized as "animals") without specialization. To mitigate hallucinations, we also added the generic concepts "*background*" and "*other*" to each subset to provide the segmentation model with default choices. Masks generated for these generic concepts are excluded from the explanation generation. The full list of concepts will be released as supplemental material and included in the official repository upon acceptance.

It is worth noting that the specific concept set obtained after iterations of our refinements is not, in general, the optimal one and potentially better sets could be found for specific implementations of the framework. Other than identifying the limitations discussed in Section B, throughout the refinement process, we also observed a **relation between the specificity of the concepts and the completeness of the concept subset**. In this context, we noted that greater specificity in the concept subset helps the segmentation model to reduce hallucinations when the concept subset is either highly specific or weakly complete (i.e., the set is completed by the concepts "background" and "other" whose effectiveness depends on the specific backbone model). For example, adding the middle term "*bird's*" to the concept subset of parts empirically improved segmentation masks. A similar effect could be achieved by merging the Ade20k set with single-granularity concept subsets. However, sharing concepts across multiple subsets causes inconsistencies in mask generation (i.e., one can have different masks for the same concept across two different subsets) and, consequently, in the explanation process. We leave the development of a framework capable of addressing and managing repeated concepts across multiple concept subsets as a direction for future work.

## E  Leveraging WordNet to Analyze the Misalignment

This section describes the specific design choices related to step 1 and step 3 when using WordNet for studying the misalignment.

**Step 1: Mapping the Concept Set to Nodes in WordNet**  We utilize information from Ade20k to implement it in a semi-automatic manner. Specifically, each class in the dataset is associated with a list of synonyms retrieved from WordNet. We leverage this list to locate the corresponding node in the WordNet graph. Given a concept and its list of synonyms, we select the node whose lemmas have the maximum overlap with the list of synonyms. For concepts without available synonyms, we extract the most common node in WordNet associated with that concept (i.e., the first result returned by a WordNet query). Finally, we manually inspect the generated mappings and refine the associations for the following concepts: *water*, *cushion*, *van*, *plate*, and *radiator*.

**Step 3: Selection of Excluded High-Level Nodes.**  The purpose of the ancestor-identification step is to determine whether differences between explanations arise from variations in concept granularity. However, because every pair of concepts in a semantic hierarchy eventually shares a common ancestor, not every ancestor provides meaningful information about explanation differences. If ancestors that are overly abstract

Table 16: Number of related and unrelated concepts merged when removing high-level nodes in WordNet. Concepts are considered related if they co-occur in the same samples activated by a given explanation.

| Removed Nodes | Merged | |
|---|---|---|
| | Related | Unrelated |
| None | 0.27 | 0.73 |
| Abstract | 0.28 | 0.72 |
| Abstract + General | 0.51 | 0.49 |
| Abstract + General + Task | 0.55 | 0.44 |

Table 17: Distribution of disagreement categories between explanations computed using human-annotated data and CAT-Seg-generated annotations, before and after removing high-level ontology nodes and generalizing the corresponding concept sets.

| Nodes Removed? | Identical | Hyper-Related | Highly Related | Low Related |
|---|---|---|---|---|
| No | 0.53 | 0.10 | 0.18 | 0.19 |
| Yes | 0.56 | 0.08 | 0.16 | 0.19 |

or excessively broad are retained, semantically unrelated concepts may incorrectly appear as different granularities of the same concept. To avoid this issue, we exclude a subset of high-level nodes from the analysis. Specifically, we divide the excluded nodes into three categories.

**Abstract nodes.** The first category includes abstract concepts that are either difficult to represent visually or irrelevant to the computer vision domain. Examples include concepts such as *entity*, *abstraction*, *substance*, *relation*, and *physical entity*. These nodes are located near the root of the ontology and provide little semantic information regarding the visual content of an image. Therefore, treating two concepts as different granularities of such ancestors would produce misleading results. For example, without excluding these nodes, concepts such as *airplane* and *lake* could be considered different granularities of the common ancestor *physical entity*, despite representing semantically unrelated visual concepts.

**General-purpose nodes.** The second category includes concepts that are visually representable but remain excessively broad for most computer vision analyses. Examples include *object*, *device*, *natural object*, *structure*, and *surface*. While these concepts are more informative than the abstract nodes above, they are still too coarse to provide insight into explanation differences. For example, retaining the ancestor *object* would cause concepts such as *stairs* and *flowerpot* to be treated as different granularities of the same concept, which provides little information for understanding explanation differences.

**Task-specific nodes.** The final category contains concepts that are sufficiently specific to be visually meaningful but remain too coarse for the particular task being analyzed. Unlike the previous categories, these nodes are task-dependent and should be selected through dataset analysis and domain knowledge. In our experiments, we analyze models trained on Places365, whose objective is scene recognition. In this context, concepts such as *piece of furniture*, *container*, *equipment*, and *transport* are often too broad to support analysis. For example, treating *toilet* and *table* as different granularities of *piece of furniture* would hide distinctions that are informative for discriminating between scene categories such as bathrooms and dining rooms. Similarly, treating *tank* and *bathtub* as different granularities of *container* would excessively broaden the notion of granularity and reduce the interpretability of the resulting analysis.

The identification of nodes to remove is ontology- and task-dependent and should primarily be guided by the analysis itself. However, as additional support, users may leverage the percentage of potentially unrelated and related concepts that are merged over the total number of merged concepts when specific nodes are included or excluded from the process. These quantities can be computed by measuring how often the two concepts co-occur in the samples selected by the corresponding explanations. Table 16 shows that the number of unrelated concept pairs decreases substantially when the nodes listed above are removed, while

the number of related concept pairs increases. These quantities can therefore be used as a robustness check. However, they should not be interpreted as metrics to optimize, since the purpose of the process is to support analysis rather than maximize a numerical score. Moreover, concepts may still be semantically related even if they do not co-occur in the specific samples selected by an explanation, either because of dataset-specific factors (e.g., media-player and cd-player are mutually exclusive in the dataset) or because of the restricted subset induced by the preceding concepts in the explanation. Overall, we do not believe that there exists a universally correct cutoff for excluding high-level nodes. Instead, the choice should be guided by the specific analysis, the ontology being used, and the goals of the user.

Finally, Table 17 shows that the overall distribution of disagreement categories remains largely unchanged before and after removing high-level nodes and generalizing the corresponding concept sets. Specifically, the proportion of low-related disagreements remains identical, while only minor variations are observed in the remaining categories. This result suggests that the conclusions of the disagreement analysis are independent of the specific choice of high-level nodes removed from the ontology, further proving that the exclusion process aims to improve the characterization and interpretation of explanation differences rather than altering the overall findings.

An important note is that excluding these nodes does not imply that relationships between the corresponding concepts are ignored. Concepts that are not unified through the ontology may still be identified as *hyper-related* or *highly related* through the co-occurrence analysis described in Section 4. Therefore, removing high-level nodes only affects the granularity analysis and does not prevent the identification of other forms of semantic relatedness.

Below, we provide the following complete list of removed nodes:

- **Abstract nodes:** [*"attribute"*, *"form, shape"*, *"part, portion, component, constituent, component part"*, *"relation"*, *"artefact, artifact"*, *"physical entity"*, *"unit, whole"*, *"language unit, linguistic unit"*, *"being, organism"*, *"abstraction, abstract entity"*, *"measure, amount, quantity"*, *"cause, causal agency, causal agent"*, *"animate thing, living thing"*, *"creation"*, *"entity"*, *"grouping, group"*, *"matter"*, *"instrumentality, instrumentation, means"*, *"substance"*]

- **General-purpose nodes:** [*"natural object"*, *"object, physical object"*, *"covering"*, *"surface"*, *"good, trade good, commodity"*, *"durable goods, durables, consumer durables"*, *"structure, construction"*, *"consumer goods"*, *"impediment, impedimenta, obstructer, obstruction, obstructor"*, *"tracheophyte, vascular plant"*, *"device"*]

- **Task-specific node:** [*"equipment"*, *"container"*, *"piece of furniture, article of furniture, furniture"*, *"furnishing"*, *"barrier"*, *"art, fine art"*, *"transport, conveyance"*, *"vessel"*, *"craft"*, *"way"*, *"path"*]

## F   Isolating Concept's Impact on Explanations

This section describes the procedure to isolate and evaluate the effect of a concept in both an explanation and the corresponding neuron's activations. Specifically, given an explanation of length $n$ and a concept $c_i$ included in the explanation, we compute: (1) the samples where the full explanation holds, (2) the samples where $c_i$ is present, and (3) the samples where the neuron is active within the considered activation range. Then, we compute the intersection of these three sets and we randomly visualize $m$ samples, highlighting the masks associated with $c_i$. These visualized samples represent instances where the concept is present, the neuron is active, and the concept actively contributes to the explanation.

To analyze the unexplained portion of the neuron's behavior, we consider sub-explanations $SE$ of the original explanation, where $SE$ has a length $s < n$. We then extract the samples where the neuron is active, but the sub-explanation $SE$ does not hold. This set represents the portion of the neuron's activations not explained by the sub-explanation. In our case, we use as a sub-explanation the literals shared between two different approaches. As in the previous case, we randomly visualize $m$ samples from this set, highlighting the masks produced by binarizing the activations within the specified activation range. These two visualizations are compared against each other to identify a potential misalignment between parts of the explanation and the

Table 18: Std. Dev for Alignment, Precision, and Relevance scores attributed by 100 participants to explanations computed by all the competitors. The superscript[*] indicates that the results are computed on a different probing dataset.

| Scores | Align | Prec | Relev |
|---|---|---|---|
| Places365 Probed Model | | | |
| Human (La Rosa et al., 2023) | 1.12 | 1.22 | 1.22 |
| Closed (Bau et al., 2020) | 1.42 | 1.41 | 1.24 |
| Our | 1.05 | 1.27 | 1.30 |
| CUB Probed Model | | | |
| Human (La Rosa et al., 2023) | 1.31[*] | 1.39[*] | 0.72[*] |
| Closed (Bau et al., 2020) | 1.22 | 1.47 | 1.09 |
| Our | 1.37 | 1.40 | 0.92 |

neuron's behavior. To ease the analysis and comparison of the visualizations, we prioritize samples associated with larger masks when selecting the samples for visualization.

## G  User Study Details

As described in the main paper, we conducted a user study to qualitatively assess the performance of our framework. Designing such a study presents several challenges. Indeed, neuron-level explanations are intended for researchers or developers involved in building or analyzing the model. Therefore, the complexity of the logical formulas and the need for a deep understanding of the model's training task represent critical factors and current limitations of this type of explanation. These characteristics constrain the pool of suitable participants and necessitate careful consideration when designing the study instructions.

**Setup**   To recruit participants, we used the Prolific platform[4]. Eligibility criteria required participants to be AI taskers (i.e., a special group of Prolific participants with proven skills in completing AI evaluation and training tasks[5]). Additionally, participants were required to have completed over 100 prior submissions with an approval rate above 90%, and not be affected by color vision deficiency. The former requirements are common in the platform and ensure familiarity with the platform itself. The latter requirement was necessary due to the use of color-based concepts in the explanations and the importance of color features for distinguishing bird species in the CUB dataset. The survey is hosted and has been created using the Qualtrics platform[6].

The median completion time for the survey was ∼30 minutes and all participants were compensated at a rate above the minimum wage in the country of the data collector (full details will be disclosed upon acceptance to preserve anonymity). We recruited a total of 100 participants. Three responses were excluded from the analysis due to their completion times (less than 10 minutes), which were significantly shorter than the median and indicated potential low-quality evaluation.

The full question pool consisted of 120 questions (i.e., 60 questions per model, 20 questions per score and 20 per method). Each model question refers to a different neuron, thus we consider explanations associated with 60 different neurons. Neurons have not been cherry-picked and they correspond to the neurons at the indices 0-60 for their respective models. All the explanations are associated with the highest cluster and competitors do not share neurons. Each participant was presented with a randomized subset of 30 questions, comprising 10 questions for each score. Due to this randomization, the 10 questions assigned per score could include questions related to one, two, or all competitor methods (see the "Alternative Design Choices" paragraph below for a discussion of this design).

---

[4]https://www.prolific.com/
[5]https://participant-help.prolific.com/en/article/5baf0c
[6]https://www.qualtrics.com/

**Scores** Participants were asked to rate how many concepts in the explanations generated by each method were aligned, precise, and relevant on a scale from 1 (none) to 5 (all). Given a randomly sampled set of activation masks produced by a neuron within a specific activation range, a concept is considered aligned if it appears in at least a subset of the activated masks; precise if its level of granularity matches that of the concepts included in the activation masks; and relevant if it is perceived as discriminative for the given task. Note that, since the visualization is extracted **independently** from the specific explanation, masks can be noisy and include more or different concepts from the ones included in the explanations due to superposition (Elhage et al., 2022; O'Mahony et al., 2023). The definitions of the scores given to the participants are the following:

- The **alignment score** measures the alignment between the label and what is shown in the unmasked regions of a collection of images. A high alignment score means that most (or all) of the concepts included in the label are also present in the unmasked regions of the images. A low alignment score indicates that few (or none) of the label's concepts appear in the unmasked regions of the images.

- The **precision score** measures the difference between the granularity of concepts included in a label and the granularity of the same concepts visualized in the unmasked regions of a set of images. A high precision score indicates that the label accurately reflects the level of detail (granularity) shown in the unmasked regions of the images. A low precision score means that the label is either more general or more specific compared to what is shown.

- The **relevance score** is used to rate how relevant a concept is for the task (i.e., how informative it is about what the model has learned). A concept is highly relevant if it is discriminative for the task (i.e., it provides useful information to distinguish between different classes or categories of objects). It is correlated to the task if it may frequently appear in the data related to the task but does not help differentiate between classes. A concept is considered low in relevance if it is neither discriminative nor highly correlated.

**Results** The average scores are reported in the main paper, while the standard deviations are provided in Table 18. As discussed in the main paper, the user study confirms that our framework performs consistently across both datasets. In this section, we provide a more detailed analysis of these results.

We begin with the model trained on the Places365 dataset. As expected, the explanations generated by both the "*human*" baseline and our framework achieve comparable scores, with no statistically significant difference between them. This similarity is reasonable given that both methods use ADE20K as a concept set, which contains concepts that are known to closely align with the semantic space learned by the Places365 model (Bau et al., 2017) and the differences in concept granularity affect only a small subset of the explanations, as discussed in Section 5.3. In contrast, the Closed approach (Bau et al., 2020) receives the lowest granularity score. While some concepts are shared between COCO (i.e., the dataset used to train the segmentation model underlying the Closed approach) and Places365, many relevant concepts are either missing or represented at a different level of granularity, resulting in being considered either too broad or too specific. The statistical significance of this difference is supported by P-values $< 0.001$, obtained using a two-tailed t-test comparing the Closed approach's precision scores to those of both the "*human*" baseline and our framework.

For the model trained on the CUB dataset, the "*human*" baseline probes the model using the Ade20K dataset. While its precision score remains comparable to that observed when applied to the Places365 model, its alignment score significantly drops (P-value $< 0.01$), likely due to noise introduced by the model's hallucinations over this dataset. We hypothesize that this misalignment could become even worse when the probing dataset differs substantially in terms of visual features from those used to pre-train and fine-tune the probed model. Moreover, this approach achieves the lowest score in terms of relevance (P-value $< 0.01$, two-tailed t-test, compared to our framework's score for the same model). This drop is related to the fact that, in the vast majority of explanations, the selected concepts are not semantically related to the task of bird species recognition and thus are scored low by users.

Regarding the Closed approach, it received unexpectedly similar precision scores to our framework. This result is due to the inclusion of the concept "bird" in every explanation, which aligns with the fact that all

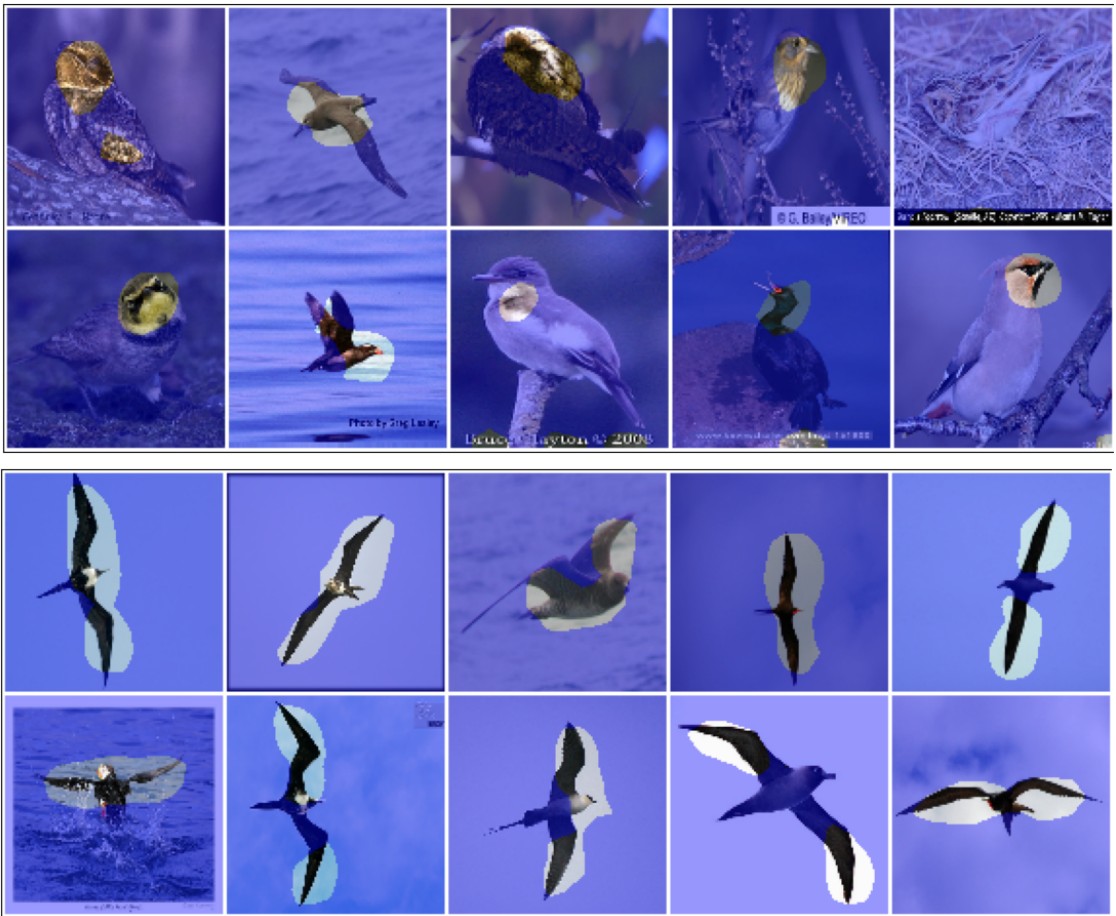

Figure 8: Examples of highly scored "bird" explanations for precision despite most of the images refer to bird parts (head (top) and wings (bottom)). Participants were asked to evaluate only the **unmasked regions** of the images.

images in the CUB dataset show birds. Since this behavior represents a degenerate case, where a generic concept is trivially included in all explanations, it should ideally be penalized in a meaningful evaluation. After analyzing the participants' responses, we hypothesize two main reasons for this outcome. First, at the individual instance level, it is difficult for inexperienced users to penalize the use of a concept like "bird" in a bird dataset, even when activation masks highlight only parts of the bird (Figure 8). This is especially true when, due to the study design and the randomness of the sampling process, users are never exposed to explanations associated with a more fine-grained granularity (i.e., they receive questions related to only the competitors' approaches). Second, Closed explanations often contain only one concept ("bird" in this case). This creates a perceived scenario in which users must decide between two extremes: either the explanation perfectly fits or does not fit at all with the granularity shown within the images. In such cases, participants may be reluctant to assign the lowest score to a concept that is slightly more general than the visualization.

In conclusion, our framework represents the preferred approach overall. Indeed, when applied to explain the CUB model, its explanations are ranked as the most relevant explanations (P-value < 0.001 using a two-tailed t-test with respect to both the baselines) by a significant margin while also achieving high scores in both alignment and precision and avoiding the degenerate behaviors observed in the other two approaches. When applied to the Places365 model, it is ranked comparably to the best approach (human) in all the scores.

**Alternative Design Choices and Limitations of User Studies**   The instructions and the user study design we used in this paper are the product of several iterations aimed at reducing the bias and improving the evaluation quality. Specifically, the resulting design is the one that penalizes the competitors the least. Below, we briefly discuss some alternative designs discarded because the resulting user study would have been too hard to understand for an inexperienced user, would have biased the evaluation, or would have penalized a competitor too much.

- **Let the participants rank different explanations for the same neuron.** One of the first designs we considered was to ask users to rank explanations produced by different methods for the same neurons. This approach would have allowed us to directly identify which explanation is preferred on average. However, this design would have unfairly penalized the "*human*" baseline in the questions related to the CUB model. In that case, the "*human*" baseline generates explanations based on a different dataset (Ade20K). As a result, if we had shown random activation masks from the CUB dataset (on which the model is trained), most, if not all, of the explanations from the human-based approach would appear misaligned, imprecise, and irrelevant, as they were computed using different concepts and a different dataset. We considered the alternative of showing two sets of images per question, one for the dataset used to generate the explanations and one for the dataset used to train the model, but this would likely have confused participants, as the two sets of images would refer to entirely different concepts and the survey setup would have been different between the CUB and Place365 models. To resolve this issue, we adopted a design in which participants evaluate each explanation independently, without seeing competing explanations. While this approach prevents us from directly extracting rankings, we can still extract insights through indirect comparisons. More importantly, this design keeps the structure of the survey consistent and easier for participants to follow and understand.

- **Let the participants score the grade of alignment/precision/relevance instead of the number of concepts.** An alternative design is to ask participants to assign a numeric score to each explanation, reflecting their perception of its overall alignment, precision, or relevance. While this setup could be more suitable for expert participants (see discussion below regarding participant pools), we believe it is not appropriate for non-researcher participants. One of the main challenges with this design is achieving a consistent interpretation of the scoring scale across participants. Although the instructions could include example ratings, for non-experts the required level of detail would likely be so extensive that it could bias the evaluation process and compromise the statistical significance of the results. Instead, we ask participants to rank the number of concepts they perceive as aligned, precise, or relevant in each explanation. This approach simplifies the task for non-expert users and avoids the need for detailed examples or guidelines that might influence their judgments. The trade-off of this design, however, is that methods producing shorter explanations, particularly those with only one concept (i.e., the Closed approach applied to the CUB model), may gain an unintended advantage. In such cases, participants are often forced to choose between the two extremes of the ranking scale (either all or none of the concepts are aligned, precise, or relevant) and we observed a tendency to favor the positive extreme in these situations, as we discussed in the previous paragraph.

- **Different level of details for instructions.** We iterated several times on the level of detail provided in the instructions and tested them with different types of users. While there is no one-size-fits-all solution, the current version of the instructions is perceived differently by different users. Based on early feedback, we found that some researchers working in the same area as this paper might consider the instructions overly detailed or guided. However, given the very limited number of experts worldwide in this specific field, the likelihood of such users being recruited through a crowd-sourcing platform is extremely low and can be considered negligible. In contrast, most participants are individuals with some familiarity with the AI domain, but who likely lack deep knowledge of explainability or the specific tasks discussed in this work. According to some participants' feedback, they would have preferred even more detailed instructions and a more guided process, as they often struggled to evaluate the explanations due to several challenges (e.g., image and mask noise and resolution, edge cases). Regardless, we intentionally chose not to provide additional guidance to

avoid introducing bias into the evaluation process. We believe that the "uncertainty" experienced by some users is an intentional and even desirable aspect, as there are no definitive right or wrong answers (i.e., ground truth) in the context of these types of explanations.

- **Different participants pool.** Given the expertise required to understand logical formulas and the deep familiarity with the underlying tasks needed to evaluate such explanations, one possible option would have been to select participants for the survey exclusively based on these two criteria. However, this approach would have resulted in a very limited participant pool, making it difficult to obtain statistically significant results. Moreover, identifying and recruiting such participants would have required considerable time and effort, effectively ruling out the use of crowdsourcing platforms. For the same reasons, enlarging the participant pool would have meant reducing the quality of the evaluations, as many potential participants might lack knowledge of what constitutes an AI task or even a basic understanding of AI itself. This would increase both the time required to comprehend the instructions and the survey, and introduce noise into the user study, making it more challenging to extract meaningful insights.

In conclusion, given the challenges described in this section regarding the design of user studies for this type of explanation, we argue that quantitative metrics, such as those used in Section 5, should remain the main tool for evaluating these methods. However, user studies can still provide insights into aspects that are difficult to capture quantitatively (e.g., relevance). As we have discussed, designing unbiased and fair surveys for these neuron explanations, without compromising the evaluation quality or statistical significance, presents several challenges. We therefore call for further research to lower the expertise needed to interpret logical explanations and to address the need for deep domain knowledge of the training dataset and tasks to evaluate them. These limitations currently restrict the usefulness of these explanations to researchers or developers who are directly involved in training the models to be explained.

## H   Broader Impact Statement

The opacity of the learning process in deep neural networks remains a major barrier to their adoption in domains where understanding the rationale behind model decisions is essential for trust and accountability. In this paper, we address one of the limitations highlighted in the broader impact statement of Mu & Andreas (2020), namely the reliance on annotated datasets, which *"may be expensive to collect and may be biased in the kinds of features they contain (or omit)"* (Mu & Andreas, 2020). We argue that the explanations generated by our framework can positively contribute to the broader impact of explainability methods by expanding the range of use cases and potential users.

Although the contributions of this work are experimental and not deployed in downstream applications, we recognize potential sources of negative societal impact if the explanation process is not properly verified or is maliciously manipulated. Specifically, incorporating pre-trained open-vocabulary segmentation models into the explanation pipeline may introduce biases embedded in the segmentation process. However, detecting and mitigating such bias is as challenging in model-generated segmentations as it is in human-annotated datasets.

A more concrete vulnerability lies in the segmentation masks themselves: an adversarial actor could subtly alter the output of the segmentation model in ways that are not immediately noticeable to users but significantly distort the resulting explanations. Furthermore, as discussed in Section G, this work does not address the challenge related to the technical expertise required to implement and interpret these explanations. Both these limitations can be mitigated in future research exploring adversarial settings and improving the usability of compositional explanations.

## I   Impact of Labeling Errors and Mask Noise

In this section, we analyze the robustness of compositional explanations to errors in the concept annotations and segmentation masks. Our goal is to understand which properties an open-vocabulary segmentation model should exhibit for the resulting explanations to remain reliable. In particular, we study how often

| Label Errors | Concepts Relation | | | IoU Diff | |
|---|---|---|---|---|---|
| | Same | Related | Unrelated | Tot | wrt Error Free |
| Errors only in Unselected Concepts | | | | | |
| 0.01 | 0.993 | 0.0 | 0.007 | $\leq 1e^{-3}$ | 0.002 |
| 0.05 | 0.993 | 0.007 | 0.0 | $\leq 1e^{-3}$ | 0.000 |
| 0.1 | 0.993 | 0.0 | 0.007 | $\leq 1e^{-3}$ | 0.002 |
| 0.2 | 0.993 | 0.007 | 0.0 | $\leq 1e^{-3}$ | 0.000 |
| 0.3 | 0.993 | 0.007 | 0.0 | $\leq 1e^{-3}$ | 0.000 |
| 0.5 | 0.993 | 0.0 | 0.0 | $\leq 1e^{-3}$ | 0.000 |
| Random Errors | | | | | |
| 0.01 | 0.917 | 0.042 | 0.041 | 0.001 | 0.001 |
| 0.05 | 0.786 | 0.118 | 0.097 | 0.009 | 0.003 |
| 0.1 | 0.669 | 0.158 | 0.172 | 0.015 | 0.005 |
| 0.2 | 0.593 | 0.173 | 0.234 | 0.024 | 0.007 |
| 0.3 | 0.510 | 0.179 | 0.310 | 0.032 | 0.012 |
| 0.5 | 0.407 | 0.180 | 0.414 | 0.043 | 0.014 |

Table 19: Variation in explanation both at concept and IoU level when errors are introduced in the labeling process. The IoU Difference is computed against the baseline Avg. IoU of 0.083

explanations change under increasing levels of noise, the nature of these changes, and their impact on the explanation quality measured through IoU.

We distinguish between two types of perturbations: errors in the labels assigned to segmentation masks and errors affecting the masks themselves. For each experiment, we randomly sample 50 neurons and evaluate the explanations generated under different noise levels (i.e., $[0.01, 0.05, 0.1, 0.2, 0.3, 0.5]$) for a ResNet model trained on Place365 and using Ade20k as a probing dataset.

### I.1 Labeling Errors

We first analyze the effect of incorrect concept labels. We consider two settings: ones where label errors are introduced only in concepts that are not selected by the explanation associated with any of the selected neurons, and ones where errors are introduced uniformly at random across all concepts. Table 19 shows that when label errors are restricted to concepts that are not part of the selected explanations, the explanations are extremely stable. Across all noise levels, only a single concept changes, corresponding to less than 1% of the total concepts. This result suggests that annotation errors affecting unrelated concepts have virtually no impact on the generated explanations. Moreover, the only explanation that changes corresponds to an alternative explanation with nearly identical IoU even before the introduction of errors, indicating that both explanations were already among the highest-ranked candidates. Consequently, the resulting explanation remains faithful to the neuron behavior despite the noise introduced in the labeling process.

When label errors are introduced at random, they can affect concepts involved in the explanations, making changes more likely. In this case, we observe a different trend. Explanations are less stable, and the percentage of unchanged explanations decreases progressively from 91.7% at 1% label noise to 40.7% at 50% label noise. In this case, two observations are noteworthy. First, the IoU of the best explanation decreases as the noise level increases, indicating that annotation errors reduce the measured alignment between concepts and neuron activations. Second, although explanations change more frequently, the IoU difference with respect to the original error-free explanation remains small, increasing from 0.002 at 1% noise to only 0.015 at 50% noise. This suggests that the newly selected explanations are often alternatives that were already highly ranked in the error-free setting. In other words, the optimization process continues to identify explanations that remain faithful to the underlying neuron behavior, while the quality of the measured alignment deteriorates because of the errors present in the annotations.

| Mask Errors | Concepts Relation | | | IoU Diff | |
|---|---|---|---|---|---|
| | Same | Related | Unrelated | Tot | wrt Error Free |
| | | | Boundary Errors | | |
| 0.01 | 0.945 | 0.055 | 0.0 | $\leq 1e^{-3}$ | $\leq 1e^{-3}$ |
| 0.05 | 0.917 | 0.077 | 0.007 | $\leq 1e^{-3}$ | $\leq 1e^{-3}$ |
| 0.1 | 0.903 | 0.091 | 0.007 | $\leq 1e^{-3}$ | $\leq 1e^{-3}$ |
| 0.2 | 0.848 | 0.125 | 0.028 | $\leq 1e^{-3}$ | $\leq 1e^{-3}$ |
| 0.3 | 0.869 | 0.097 | 0.034 | $\leq 1e^{-3}$ | 0.001 |
| 0.5 | 0.697 | 0.180 | 0.124 | $\leq 1e^{-3}$ | 0.002 |
| | | | Random Errors | | |
| 0.01 | 0.041 | 0.959 | 0.0 | 0.045 | 0.012 |
| 0.05 | 0.0 | 1.0 | 0.0 | 0.063 | 0.021 |
| 0.1 | 0.0 | 1.0 | 0.0 | 0.068 | 0.025 |
| 0.2 | 0.0 | 1.0 | 0.0 | 0.072 | 0.036 |
| 0.3 | 0.0 | 1.0 | 0.0 | 0.074 | 0.042 |
| 0.5 | 0.007 | 0.993 | 0.0 | 0.076 | 0.081 |

Table 20: Variation in explanation both at concept and IoU level when errors are introduced in the mask generation process. The IoU Difference is computed against the baseline Avg. IoU of 0.083

Overall, these results indicate that compositional explanations are robust to low and moderate levels of label annotation noise. Even when explanations change, the alternative explanations typically remain close to the original best explanation, resulting in only minor differences in alignment quality. This observation is encouraging in the context of open-vocabulary segmentation models. As shown in Section 5, the alignment scores obtained using these models are of a similar magnitude in IoU, suggesting that current segmentation technology is already sufficiently reliable for compositional explanation generation.

## I.2 Mask Noise

In this section, we analyze the effect of errors in the mask generation process. Specifically, we consider two types of perturbations: boundary errors, obtained by randomly dilating or eroding the masks, and random artifacts, obtained by randomly modifying pixels within the masks. The former simulates the realistic scenario in which segmentation masks are imprecise near object boundaries, whereas the latter represents a more extreme setting that is closer to an adversarial perturbation. Indeed, modern segmentation models typically enforce spatial consistency and connected components, making purely random artifacts uncommon in practice.

Table 20 shows that compositional explanations are highly robust to boundary errors. Even at high noise levels (30%), more than 84% of explanations remain unchanged, while only 2.8% become unrelated to the original explanation. Furthermore, the IoU difference is negligible across all perturbation levels, both when considering the noisy masks and when comparing the selected explanations in the original error-free setting. These results indicate that boundary perturbations, which affect the contour of a concept while preserving its semantic identity, largely preserve the relative ranking of explanations. This causes the algorithm to either recover the same explanation or an alternative explanation that was already highly ranked in the error-free scenario.

Conversely, when random artifacts are introduced into the masks, the impact on the explanation quality is larger. The IoU difference increases with the amount of noise, reaching values comparable to the average alignment score itself. At 50% noise, the difference with respect to the original error-free explanation reaches 0.081, while the average IoU of the explanations in the error-free setting is only 0.083. This indicates that explanation selected under noisy conditions would not have been considered among the highest-ranked candidates in the original setting. Therefore, unlike the previous experiments, the optimization process is distorted and it is no longer selecting a nearby high-ranking alternative explanation.

To better understand this phenomenon, we analyzed the relationship between the original explanations and the explanations obtained after introducing random mask artifacts. Interestingly, nearly all related explanations correspond to specialization relationships (Table 20). In these cases, one of the concepts appearing in the original explanation is replaced by a more specific concept. For example, an explanation such as $(Cat \lor Dog \lor Mouse)$ may become $(Cat \land White \land \neg Table)$. This observation suggests that high amount of random mask artifacts bias the search process toward specialized concepts. As the noise level increases, the compositional explanation often collapses toward a refinement of one of the original concepts, producing explanations that resemble the single-concept explanations identified by Network Dissection (Bau et al., 2020). While these explanations remain faithful, since they still involve concepts appearing in the original explanation, they lose some of the advantages of compositional explanations by providing a less complete characterization of the neuron behavior.

Overall, these results indicate that compositional explanations are highly robust to realistic mask imperfections, such as boundary inaccuracies, but are more vulnerable to adversarial-like random artifacts. Future work could investigate mechanisms for detecting and filtering unreliable masks before explanation generation, as well as robustness-aware objective functions that explicitly account for annotation uncertainty.

### I.3 Summary and Other Error Scenarios

Overall, we observe a reassuring robustness of compositional explanations against low and moderate levels of noise in both the labeling process and the mask generation process. A recurring pattern across all experiments is that noise affects the absolute IoU values more than the relative ranking and faithfulness of candidate explanations. As a result, the explanation selected in the noisy setting is frequently identical, closely related, or similarly ranked to, the explanation selected in the error-free setting. This behavior is related to the nature of compositional explanations. These explanations quantify the spatial alignment between neuron activations and semantic concepts. When noise is introduced into either quantity, the measured alignment naturally decreases. However, because the noise is typically distributed approximately uniformly across the state space, it tends to affect all candidate explanations similarly. Consequently, the optimization process often preserves the relative ordering of explanations even when the absolute scores are degraded. Therefore, if a neuron exhibits a strong and unique alignment with a particular combination of concepts, the optimization process will generally continue to converge to the same explanation or a closely related one, albeit with a progressively lower IoU score. If the amount of noise becomes sufficiently large to eliminate the measured alignment entirely, the neuron will naturally be classified as non-interpretable according to the same criteria used throughout this work. The settings discussed above assume that the neuron activations remain unchanged with respect to those analyzed in this paper and that only the annotations are affected by noise. Below, we discuss two additional scenarios: when neuron activations are not meaningfully aligned with any concept, and when both the neuron and the segmentation model converge toward the same representation, but such representation is misaligned with the human definition of the concept.

**Misaligned neurons.** When a neuron is not strongly aligned with any combination of concepts, accidental alignment between its activation and noisy segmentation masks across a large dataset is highly unlikely. In these cases, annotation noise is not expected to create meaningful explanations. Moreover, previous work (La Rosa et al., 2023) provides mechanisms for detecting and handling such neurons, for example by comparing the resulting explanations against those obtained from randomly initialized networks. Therefore, explanations that fail these validation procedures should be discarded irrespective of whether the annotations are generated by segmentation models or humans.

**Shared Model Biases.** A final scenario worth considering is when both the neuron under analysis and the segmentation model represent a concept in a similar way, but in a way that differs from the human interpretation of the same concept. For example, both the neuron and the segmentation model may recognize the concept of "sky" as only a subset of the pixels that a human would identify as sky. In this case, the explanation may be associated with a high alignment score while still being semantically misaligned with human expectations. Unlike the previous forms of noise, there are currently no safeguards against this type of behavior. However, these cases are relatively easy to identify through visual inspection, since they

necessarily involve concepts that appear in the explanation itself. Visualizing the overlap between neuron activations and segmentation masks is typically sufficient to reveal such discrepancies.

It is also important to note that, in this last scenario, the explanation algorithm is still working as intended. The purpose of compositional explanations is to identify patterns aligned with neuron activations, and the algorithm successfully recovers those patterns. What may be incorrect is not the alignment itself, but rather the semantic label associated with it. Interestingly, this scenario may itself represent a valuable scientific finding. If two models trained on different tasks and datasets independently converge toward the same representation, yet that representation differs from the human one, this would suggest the emergence of a shared abstraction that is not grounded in human semantics. While we did not observe clear evidence of this behavior in our experiments, we believe it represents an interesting direction for future research. More generally, understanding when models converge toward representations that differ from human ones, and whether such representations are beneficial or detrimental for downstream tasks, could provide useful insights into the nature of learned abstractions in deep neural networks.

## J  Reproducibility

To ensure full reproducibility, we will release the complete codebase and all scripts required to reproduce the results presented in this paper upon acceptance. In the meantime, this section serves as a brief summary and documentation of the experimental setup used by our framework, along with the resources required.

### J.1  Dataset, Models, and Explanations

In this section, we provide the repository, dataset, explanations, and model information, versions, their corresponding licenses, download links, and a brief description of the modifications required to ensure compatibility with our framework.

**Datasets**

- Mapillary Vistas (Neuhold et al., 2017) v. 1.2
  - Accessible at: `https://www.mapillary.com/dataset/vistas`
  - License: CC BY-NC-SA and subject to Mapillary Terms of Use[7]

- Cityscapes (Cordts et al., 2016)
  - Accessible at: `https://www.cityscapes-dataset.com/`
  - License: MIT license and custom terms of use[8]

- Pascal VOC (Everingham et al., 2012)
  - Accessible at: `http://host.robots.ox.ac.uk/pascal/VOC/`
  - License: flickr terms of use[9]

- PASCAL-Context-459 (Mottaghi et al., 2014)
  - Accessible at: `https://cs.stanford.edu/~roozbeh/pascal-context/`
  - License: flickr terms of use[9]

- Ade20k (Zhou et al., 2017b)
  - Accessible at: `https://ade20k.csail.mit.edu/`
  - License: MIT

- COCO-Stuff (Caesar et al., 2018)

---

[7]https://www.mapillary.com/terms
[8]https://www.cityscapes-dataset.com/license/
[9]https://www.flickr.com/help/terms

- Accessible at: `https://cocodataset.org/`
- License: CC-BY 4.0 and Flickr terms of use[9]

- CUB (Wah et al., 2011)

  - Accessible at: `https://www.vision.caltech.edu/datasets/cub_200_2011/`
  - License: CCO

To make the datasets compatible with Detectron2 (Wu et al., 2019), we follow the instructions reported in the following repositories:

- `https://github.com/cvlab-kaist/CAT-Seg/tree/main` for Ade20k (150 classes and its extended version), Pascal VOC, Pascal-Context, and COCO-Stuff

- `https://github.com/facebookresearch/MaskFormer/tree/main` for Cityscapes and Mapillary Vistas

**Models**

- CAT-Seg (Cho et al., 2024)

  - Accessible at: `https://github.com/cvlab-kaist/CAT-Seg`
  - License: MIT
  - Version: Large (L)

- MasQCLIP (Xu et al., 2023d)

  - Accessible at: `https://github.com/mlpc-ucsd/MasQCLIP`
  - License: CC BY-NC 4.0
  - Version: Cross-Dataset

- SCAN (Liu et al., 2024)

  - Accessible at: `https://github.com/yongliu20/SCAN`
  - License: CC BY-NC 4.0
  - Version: SCAN-VitL

- SED (Xie et al., 2024)

  - Accessible at: `https://github.com/xb534/SED`
  - License: Apache 2.0
  - Version: SED (L)

- OpenSeed (Zhang et al., 2023)

  - Accessible at: `https://github.com/IDEA-Research/OpenSeeD`
  - License: Apache 2.0
  - Version: COCO o365 SwinT

- Mask2former (Zhang et al., 2023)

  - Accessible at: `https://github.com/facebookresearch/Mask2Former`
  - License: CC BY-NC 4.0
  - Version: COCO Panoptic SwinT

We slightly modified the implementation of all these models to provide a unified interface compatible with the capabilities of our framework. Importantly, these modifications do not affect the pre-trained weights and do not require retraining the segmentation models. Specifically, we extended the models with an interface that allows arbitrary concepts to be added, removed, or specified on the fly. This replaces the default interface, which relies on dataset-specific classes supported by Detectron2. When necessary, we preserve model-specific dataset customizations by loading concepts from JSON files provided by the original authors. For all the models, we use the default parameters suggested and tested by the original authors.

**Explanations**   Our framework generates explanations through a heuristic search guided by the MMESH heuristic (La Rosa et al., 2023). The implementation of the heuristic is based on the one provided by the original authors, available at `https://github.com/KRLGroup/Clustered-Compositional-Explanations`, while the search procedure is inspired by the compositional explanation repository at `https://github.com/jayelm/compexp`. In our experiments, we fix the number of clusters to 5 and set the explanation length to 3, following the setup proposed in Mu & Andreas (2020). The beam branching factor is also set to 5. For building the logical forms of explanations, we employ the AND, OR, and AND NOT operators, as specified in Mu & Andreas (2020).

**Repository**   The technical settings considered in this paper and the ones needed **to replicate its results** include the following libraries: PyTorch 1.3 (Paszke et al., 2019), Detectron2 (Wu et al., 2019), MMEngine 1.6.2 (Contributors, 2018), and MMSegmentation 0.27.0 (Contributors, 2020)). Note, however, that our framework generally supports custom datasets and models. The core implementation is compatible with any PyTorch version > 1.3 and does not rely on functionalities specific to MMEngine or MMSegmentation. The only **general requirement** is that the open vocabulary segmentation model must be adapted to the common interface expected by our framework for parsing datasets and that the dataset loading function is made compatible with our implementation. A complete guide on how to integrate custom models and datasets will be provided in the repository upon acceptance.

## K   Visual Comparison between Closed and Open Vocabulary Explanations

This section includes a visual comparison of the explanations generated by the *Closed* approach and our framework on the CUB dataset. Specifically, we show the explanations generated for the first 20 neurons of the CUB model described in Section 5 for the highest cluster (Figures 9 to 15).

As noted in the main text, we can observe that the *Closed* approach (Bau et al., 2020) fails to recognize the specific concepts captured by the activation range and its explanations are comparable only when the neuron focuses on background elements or general concepts (e.g., water, sky), thus highlighting the lack of flexibility of this approach.

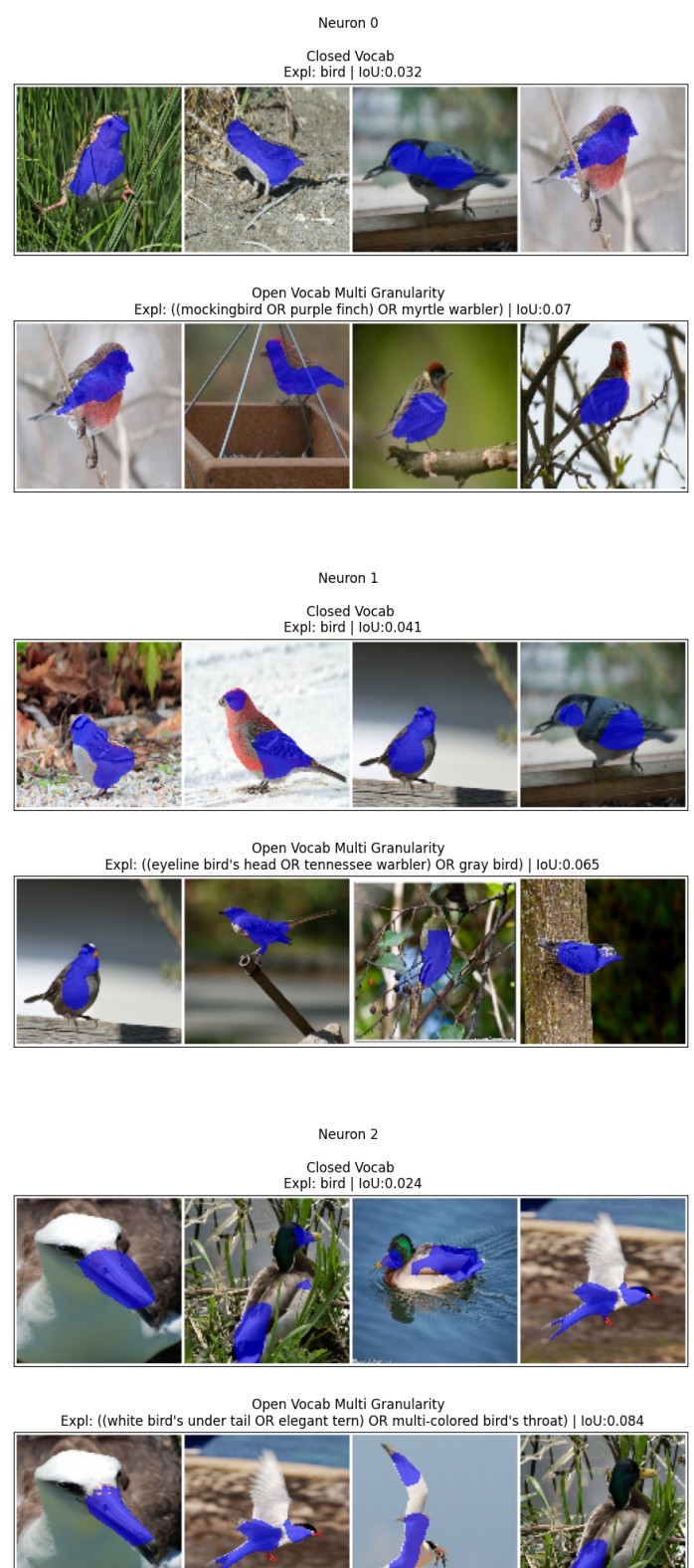

Figure 9: Explanations associated with Cluster 5 of neurons from 0 to 2 by the *Closed* approach (Bau et al., 2020) and our framework. In blue are areas of neuron activation within the considered range.

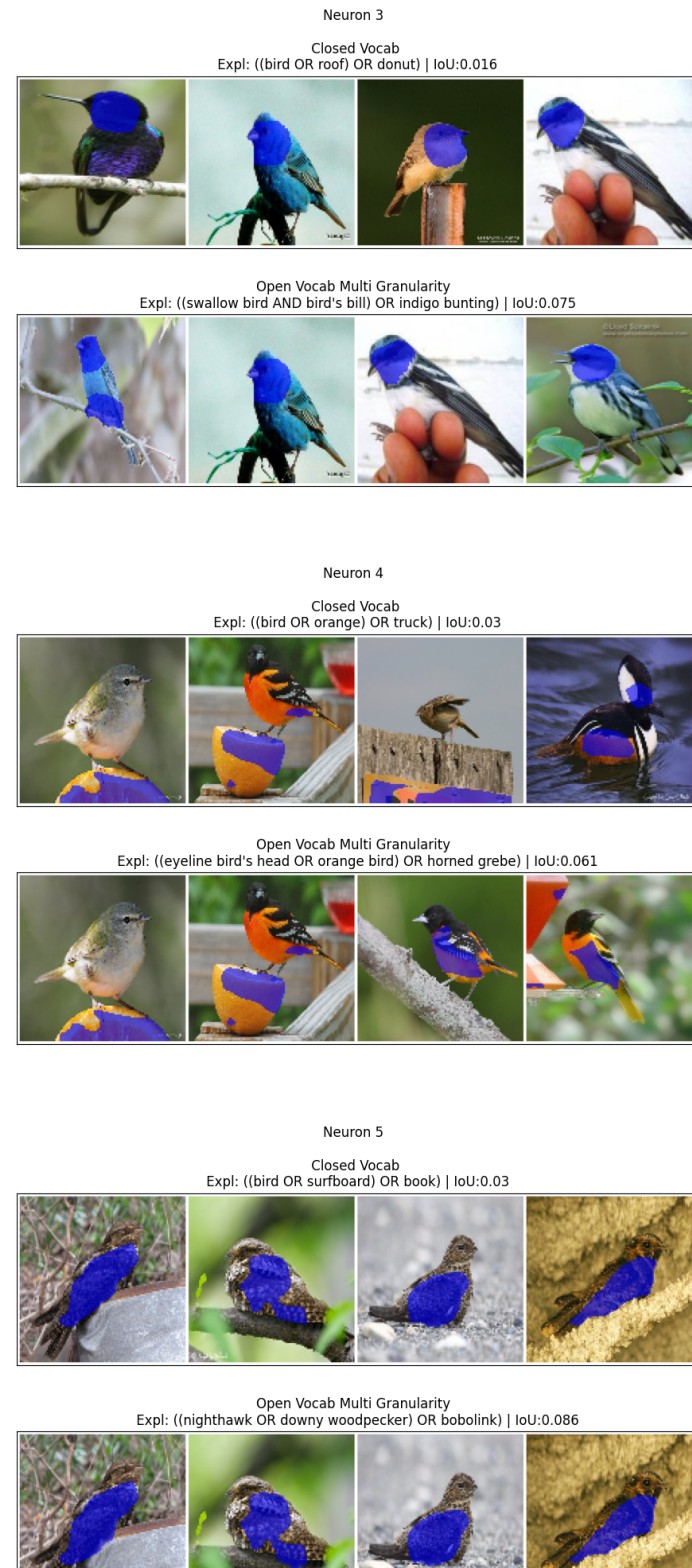

Figure 10: Explanations associated with Cluster 5 of neurons from 3 to 5 by the *Closed* approach (Bau et al., 2020) and our framework. In blue are areas of neuron activation within the considered range.

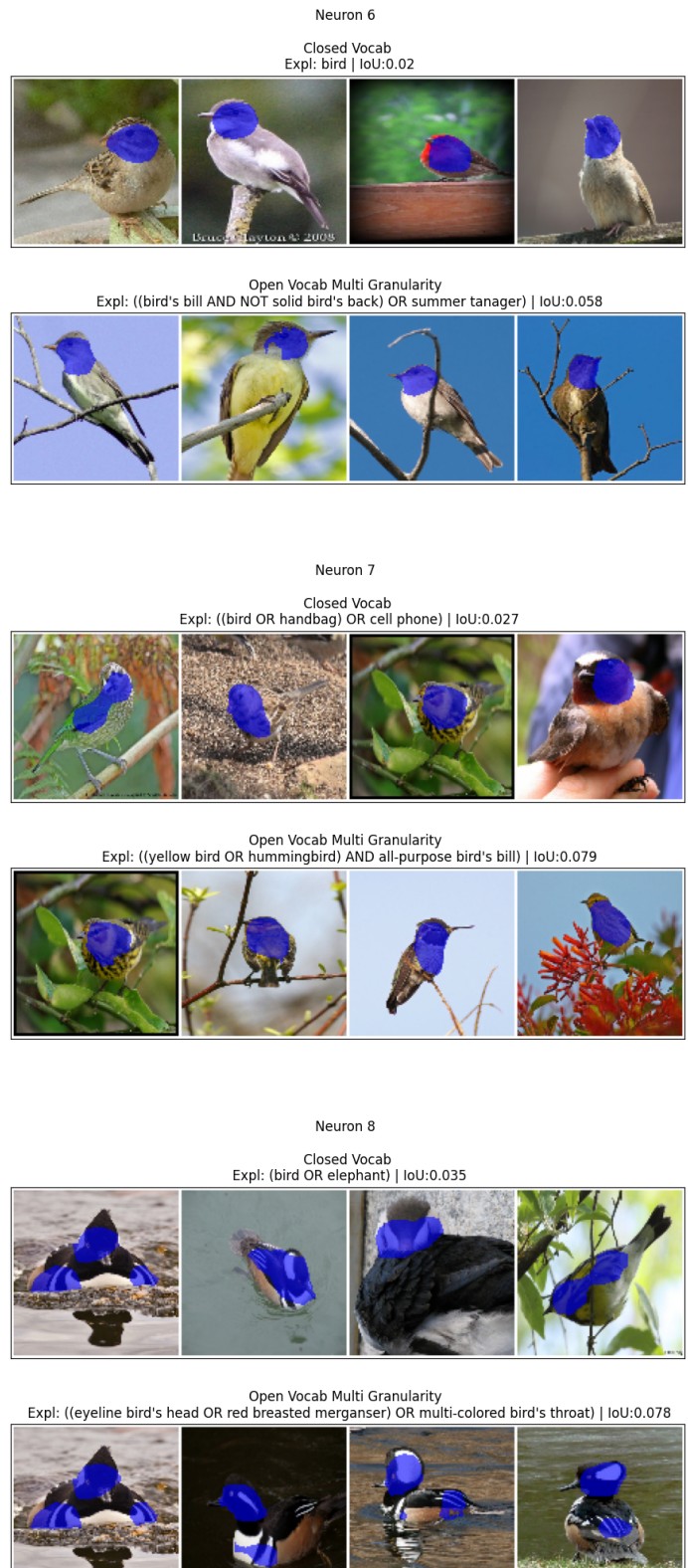

Figure 11: Explanations associated with Cluster 5 of neurons from 6 to 8 by the *Closed* approach (Bau et al., 2020) and our framework. In blue are areas of neuron activation within the considered range.

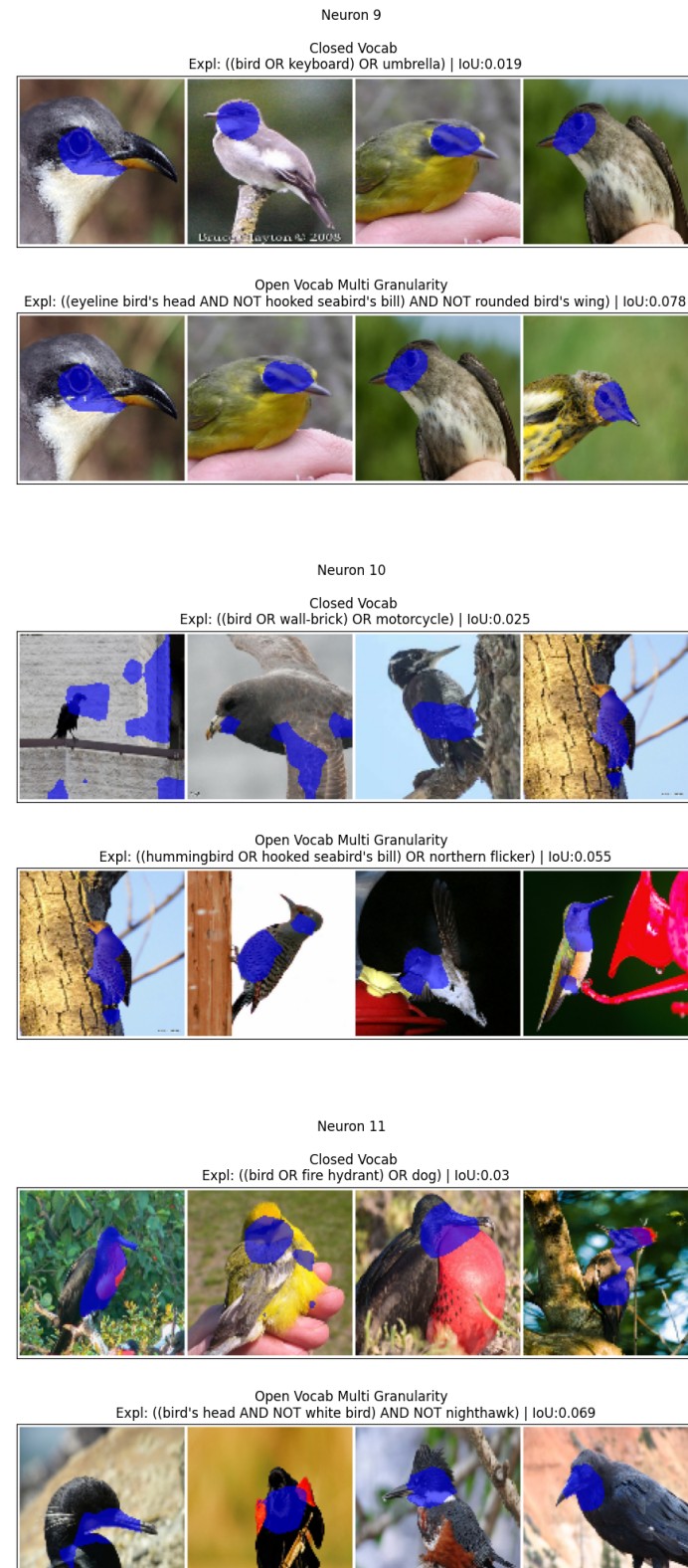

Figure 12: Explanations associated with Cluster 5 of neurons from 9 to 11 by the *Closed* approach (Bau et al., 2020) and our framework. In blue are areas of neuron activation within the considered range.

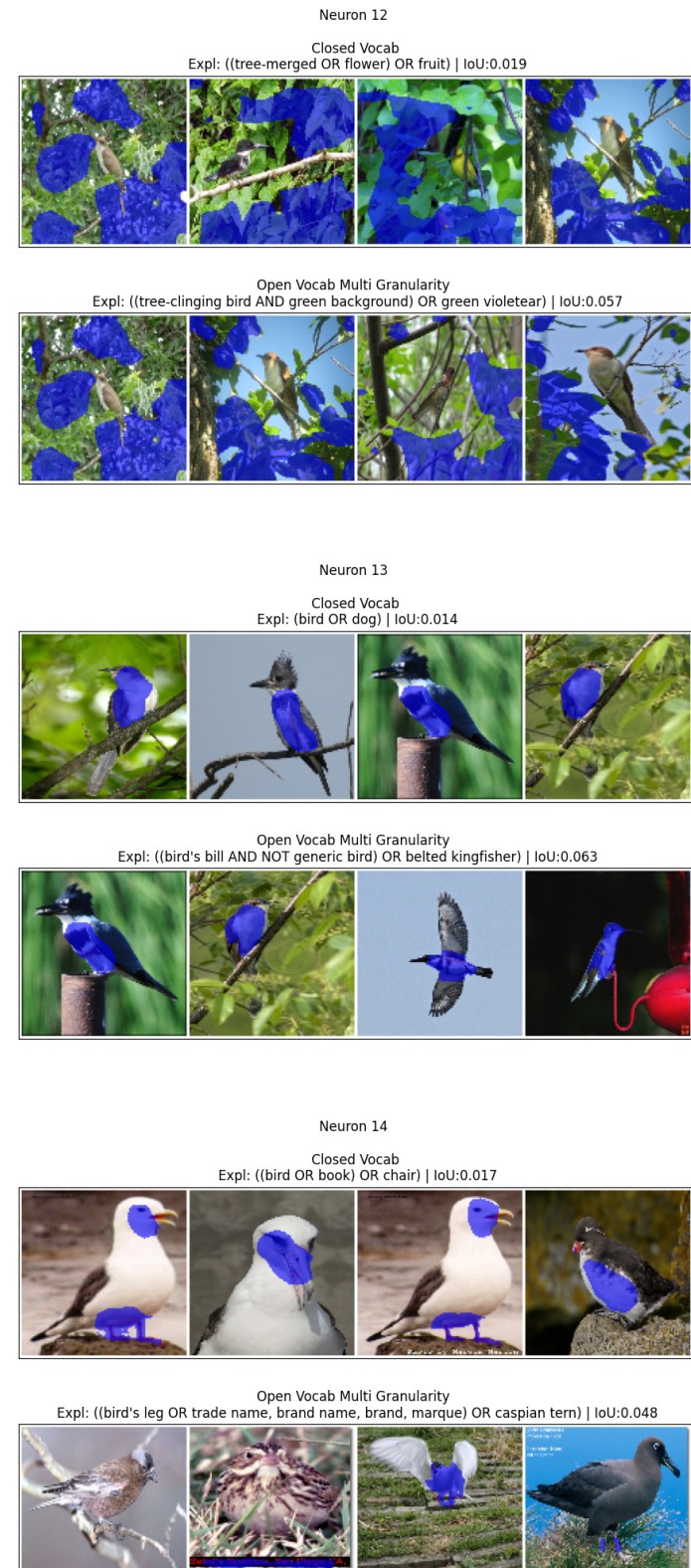

Figure 13: Explanations associated with Cluster 5 of neurons from 12 to 14 by the *Closed* approach (Bau et al., 2020) and our framework. In blue are areas of neuron activation within the considered range.

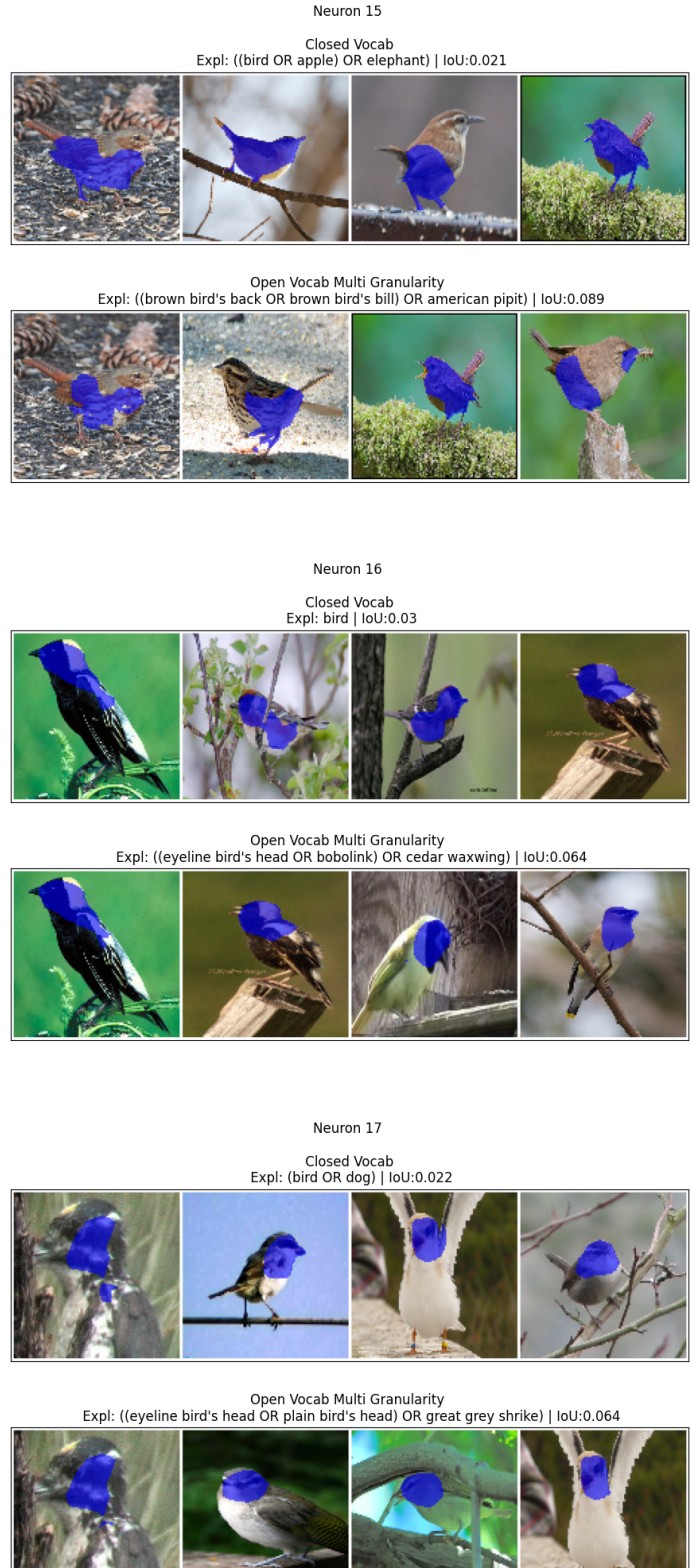

Figure 14: Explanations associated with Cluster 5 of neurons from 15 to 17 by the *Closed* approach (Bau et al., 2020) and our framework. In blue are areas of neuron activation within the considered range.

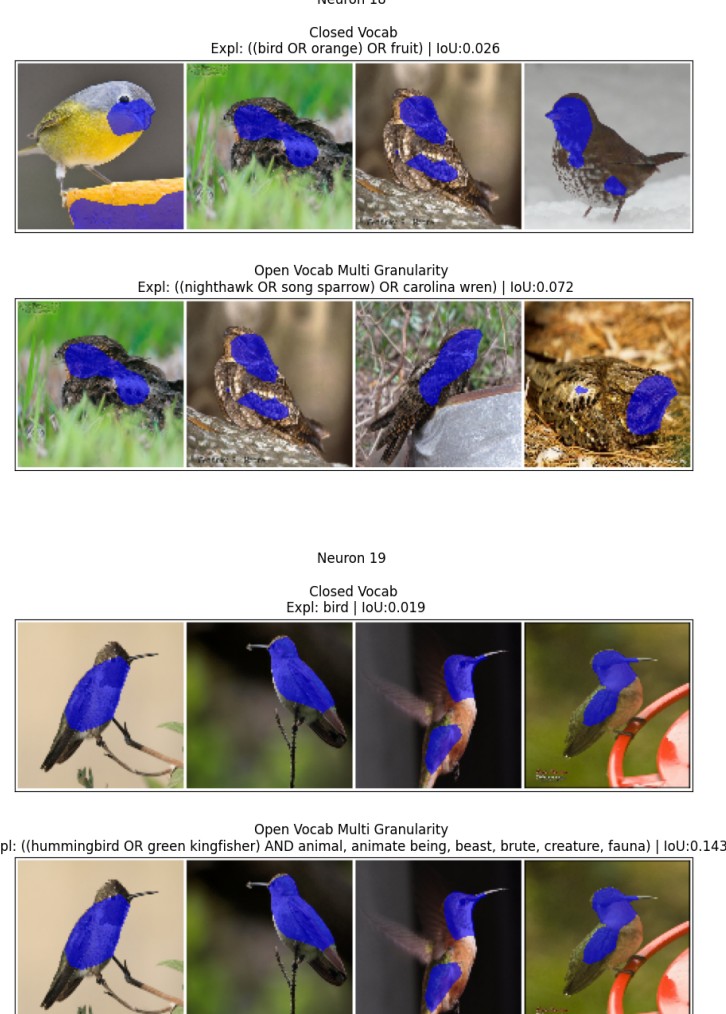

Figure 15: Explanations associated with Cluster 5 of neurons from 18 to 19 by the *Closed* approach (Bau et al., 2020) and our framework. In blue are areas of neuron activation within the considered range.

