# OpenReview forum: "Open Vocabulary Compositional Explanations for Neuron Alignment"
_TMLR — Under review for TMLR_

### Review · Reviewer_aed7 · 2026-05-19

**Summary Of Contributions:**

This paper tackles the heavy reliance on human-annotated datasets in generating compositional explanations for neuron alignment. The authors propose an open-vocabulary framework for the vision domain, dynamically generating segmentation masks for arbitrary, user-defined concepts using off-the-shelf open-vocabulary segmentation models. These masks are then fed into a beam search and heuristic algorithm to extract logical compositional explanations. Additionally, a knowledge graph-based process is introduced to analyze misalignments between model-generated and human-annotated explanations.

Strengths:

The primary strength lies in its practical engineering value，it bypasses the constraints of fixed concept sets and expensive manual annotations, enabling iterative, multi-granular explanation refinement. The extensive benchmarking across various open-vocabulary segmentation backbones in the appendix is also commendable.

Key Weaknesses:

1. The methodological contribution feels like a straightforward pipeline integration of existing open-vocabulary segmenters with established compositional search algorithms.

2. The experimental testbed is restricted to legacy architectures (ResNet18/50), ignoring modern vision models.

3. The inherent hallucinations and semantic biases of the explainer (the segmentation model) itself are not adequately disentangled from the actual behavior of the probed neurons.

4. The manuscript completely lacks a framework or pipeline diagram. Given the multi-step nature of the proposed method, relying solely on text makes the methodology unnecessarily difficult for readers to follow and grasp.

**Audience:**

Yes

**Audience Explanation:**

Researchers in mechanistic interpretability will find the pipeline helpful.

**Claims And Evidence:**

No

**Claims Explanation:**

The authors claim that their framework provides flexible and accurate neuron explanations for vision models. However, the supporting evidence has two critical methodological gaps.

First, the qualitative and quantitative probing experiments are entirely based on heavily outdated Convolutional Neural Networks, specifically ResNet18 and ResNet50. In an era where Vision Transformers (ViTs) and multi-modal alignment architectures dominate computer vision, the absence of experiments on modern architectures leaves the claim of universal applicability unproven.

Second, adopting off-the-shelf open-vocabulary segmentation models as mask generators inevitably conflates the segmentation model's own hallucinations and semantic biases with the actual behavior of the probed neurons. While the authors discuss misalignment qualitatively, they lack a quantitative assessment of this error propagation. The current experiments cannot definitively prove whether a high alignment score stems from the neuron genuinely capturing the concept, or merely from the segmentation model and the probed model undergoing similar feature collapse on the same data domain.

**Requested Changes:**

1. Evaluate the framework on at least one modern architecture (e.g., ViT). We need to see if this local pixel-based search remains effective in feature spaces with global attention.
2. Add a quantitative ablation showing how segmentation artifacts or hallucinations affect the final logical formulas.
3. Add a figure illustrating the full system pipeline. The text-only explanation of the multi-step process is difficult to follow.

---

> ### Author Response · Authors · 2026-06-15
> **Rebuttal Part 1**
>
> We thank the reviewer for the time spent reading the paper and providing useful feedback. We implemented the requested changes in the new revision. In this and the next replies, the reviewer can find the explanation and points to each of the requested changes.
>
> > Evaluate the framework on at least one modern architecture (e.g., ViT). We need to see if this local pixel-based search remains effective in feature spaces with global attention
>
> To address the reviewer’s request, **we added experiments on one modern convolutional architecture (ConvNeXt) and three transformer-based architectures (MaxViT, EfficientViT, and CvT) in Section A.2 (Table 14).** Across all four architectures, the relative behavior of the proposed framework and the baselines remains consistent with the results observed in the main experiments.
>
> More generally, we would like to clarify the scope of the paper. Our goal is not to establish “the universal applicability” of compositional explanations across all architectures, but rather to broaden the applicability of compositional explanations to settings where human annotations are unavailable and to increase the flexibility of the annotation process. For this reason, the main experiments focus on settings that are already well established in the compositional explanation literature, allowing us to isolate and evaluate the impact of the proposed modifications to the annotation pipeline.
>
> We agree that evaluating more recent architectures is important. However, unlike the classical convolutional settings, transformer-based architectures do not yet have well-established experimental protocols within the network dissection or compositional explanation literature. As a result, the experiments reported in Section A.2 should be viewed as proof-of-concept evaluations. For example, activation distributions in these architectures differ substantially from those typically observed in convolutional networks. Consequently, activation-range extraction strategies used in prior compositional explanation work, such as fixed quantiles from Network Dissection or clustering-based approaches from Clustered Compositional Explanations, are less effective without additional tuning. Similarly, the relationship between the pretrained task and the probing dataset can differ from the settings traditionally studied in the literature. These factors lead to lower alignment scores, although several of the evaluated units still exceed the interpretability threshold (0.04) adopted in the Network Dissection literature.
>
> We believe these results are encouraging and suggest that compositional explanations can be extended to modern architectures. At the same time, identifying the most appropriate settings, activation-range extraction procedures, and hyperparameters for transformer-based models remains an open research problem. For this reason, while we added transformer experiments as requested and observed conclusions consistent with those reported for convolutional architectures, we consider the identification of optimal settings for transformer-based models to be outside the scope of the current paper and we believe it represents an interesting direction for future work.
>
> > Add a figure illustrating the full system pipeline. The text-only explanation of the multi-step process is difficult to follow.
>
> We thank the reviewer for the suggestion. **We added the figure (Figure 1 in the main text) illustrating the full pipeline and one additional (Figure 2) to describe the process in Section 4**. We also **added some examples** in the text to facilitate understanding.

---

> ### Author Response · Authors · 2026-06-15
> **Rebuttal Part 2**
>
> > Add a quantitative ablation showing how segmentation artifacts or hallucinations affect the final logical formulas.
>
> We thank the reviewer for this suggestion. We **added a new section in the appendix (Section I) entirely dedicated to artifacts and hallucinations**, where we quantitatively analyze the robustness of compositional explanations under different sources of annotation noise.
>
> Specifically, we study both errors in the concept labels and errors in the masks themselves. For the latter, we distinguish between realistic boundary perturbations (obtained through mask dilation and erosion) and more extreme random artifacts. For each setting, we measure how often explanations change, whether the new explanations remain related to the original ones, and how much the corresponding IoU differs from the error-free case. Our results show that compositional explanations are highly robust to realistic annotation errors. When label errors affect concepts that are not selected by the explanation, less than 1% of explanations change across all noise levels. Even when labels are perturbed uniformly at random, explanations often remain identical, closely related to those obtained from the error-free annotations, or extracted from similarly highly ranked explanations in the error-free annotations. Similarly, boundary perturbations have a limited impact on the resulting explanations, with most explanations remaining unchanged and only negligible differences in IoU. Overall, these results suggest that annotation errors have more effect on the absolute alignment score than on the ranking of explanations. Consequently, the optimization process typically converges to the same explanation, or to an alternative explanation that was already highly ranked in the error-free setting.
>
> We also considered a more extreme setting involving random artifacts in the masks. In this case, the impact is larger and the optimization process may converge to different explanations. Interestingly, these explanations are typically specialization variants of the original ones rather than completely unrelated explanations. We believe this setting is closer to an adversarial perturbation than to the behavior of modern segmentation models. Nevertheless, it provides insight into the failure modes of the framework and could be useful as preliminary experiments for future work.
>
> Regarding the specific scenario mentioned by the reviewer, namely that the segmentation model and the probed network may converge toward the same representation while both being misaligned with the human interpretation of a concept, we now discuss this possibility explicitly at the end of the new section. We agree that this scenario cannot be completely ruled out by the current framework. However, in this case, the explanation algorithm is still correctly identifying a pattern that is shared by the neuron and the segmentation model. What may be incorrect is not the alignment itself, but rather the semantic label associated with it. But, because such cases necessarily involve concepts appearing in the explanation (due to the experiments included in the new section), they can be identified through direct visualization of the overlap between neuron activations and segmentation masks. In this regard,  we would like to emphasize that the goal of compositional explanations is to characterize the patterns represented by neurons. If both the segmentation model and the probed model independently converge toward the same representation, even when that representation differs from the human one, the explanation remains faithful to the underlying neuron behavior. Understanding if and when these shared non-human representations emerge, and whether they correspond to useful abstractions or model biases, is an interesting direction for future research, and **we explicitly acknowledge and mention it in the Section I**.

---

### Review · Reviewer_hmev · 2026-05-27

**Summary Of Contributions:**

This paper addresses an important practical limitation of compositional explanations for neurons in vision models: these methods usually depend on densely annotated probing datasets, where each concept must be manually localized in the image. This limits their use to a small number of datasets and to a fixed set of predefined concepts.

The authors propose a simple and useful framework that replaces manual annotations with masks generated by open vocabulary semantic segmentation models. The user provides one or more sets of concepts, possibly at different levels of granularity, the segmentation model produces masks for these concepts, and then a standard compositional explanation method is applied on top of these masks. The paper also proposes a knowledge-graph-based procedure, using WordNet, to compare explanations produced by different approaches and to understand whether disagreements come from granularity differences, hallucinations, or missing concepts.

The experiments cover several probing datasets, including Ade20K, Mapillary Vistas, Cityscapes, Pascal-Context, COCO-Stuff, VOC, and CUB. The evaluation uses standard pixel-level metrics such as IoU, Activation Coverage, and Detection Accuracy, together with a user study. The paper also includes application examples showing how the framework can be used to inspect neurons at different granularities and to improve explanations by refining the concept set.

**Audience:**

Yes

**Audience Explanation:**

See my detailed review above.

Moreover, I like that the paper is also fairly honest about limitations, and I think that is very transparent. It acknowledges that the framework inherits errors and biases from the segmentation model, that the concept set matters, that mixing granularities can create inconsistencies, and that very large concept sets may create memory and computational issues. The commitment to releasing code and the detailed experimental setup also help reproducibility.

**Broader Impact Concerns:**

None.

**Claims And Evidence:**

Yes

**Claims Explanation:**

The paper is well motivated. The dependence on human-annotated concept datasets is a real bottleneck for compositional explanations, and the proposed use of open vocabulary segmentation is a natural and practically useful way to relax this requirement. The framework is also conceptually clean: it does not replace the compositional explanation method itself, but instead changes how the concept masks are obtained.

In my opinion, the strongest part of the paper is the CUB experiment. This is the setting where the framework is most needed: a fine-grained classification task where suitable human concept masks are not readily available. In this case, the human-annotation baseline has to rely on a different annotated dataset, which leads to explanations involving irrelevant concepts such as "pool table" or "car". The closed-vocabulary baseline tends to produce overly general explanations such as "bird" or "animal". In contrast, the proposed method can recover more task-relevant concepts, such as bird species, parts, and colors. This makes the practical value of the method clear.

The multi-granularity aspect is also valuable. The ability to ask whether a neuron aligns with object-level concepts, part-level concepts, colors, or more specific task-related concepts gives the user more control over the interpretation process. This is a meaningful improvement over methods tied to a fixed annotation vocabulary.

The knowledge-graph analysis is a useful addition. It gives the authors a way to go beyond average scores and ask why two explanation methods disagree. This is important because two labels can differ while still pointing to related visual evidence, for example when one method labels a region as "cdplayer" and another as "videoplayer". The paper’s attempt to separate granularity issues from hallucinations and missing concepts is helpful and could be useful for future work on explanation comparison.

**Requested Changes:**

My main concern is that the quantitative gains on datasets with human annotations are somewhat overstated. In Table 1, the proposed method is often comparable to the human-annotation baseline, but the differences are usually small, and in some cases the method performs worse, for example on VOC 2012. I think the fairer conclusion is that the framework can replace human annotations without a major loss in quality, rather than that it clearly improves over human annotations.

A second concern is that the method depends strongly on the open vocabulary segmentation model and on the user-specified concept set. The paper acknowledges this, but the experiments do not fully show how stable the explanations are when changing the segmentation backbone or when using a simpler concept vocabulary. This matters because, in practice, if two segmentation models or two concept sets produce different explanations for the same neuron, users may not know which explanation to trust.

The user study is useful, but some scores are hard to interpret. For example, a generic explanation such as "bird" can receive a good precision score on CUB even though it is not very informative for a fine-grained bird classifier. This suggests that the relevance score may be more meaningful than the alignment or precision scores, and the paper should clarify which parts of the user study are most important for its conclusions.

Finally, some parts of the analysis would benefit from more robustness checks. In particular, the WordNet-based disagreement analysis in Section 5.3 depends on choices such as co-occurrence thresholds and which high-level nodes are excluded. A short sensitivity analysis would make these claims more convincing. There are also minor presentation issues, such as inconsistent equation references and a Figure 2 caption that could more clearly explain the different granularity levels.

---

> ### Author Response · Authors · 2026-06-15
> **Rebuttal Part 1**
>
> We thank the reviewer for appreciating our paper, the time spent reading the paper, and providing useful feedback. We implemented the requested changes in the new revision. In this reply and the next ones, the reviewer can find the explanation and points to each of the requested changes. Note that the answers are not sorted chronologically (due to space limit in the rebuttal).
>
> > My main concern is that the quantitative gains on datasets with human annotations are somewhat overstated. In Table 1, the proposed method is often comparable to the human-annotation baseline, but the differences are usually small, and in some cases the method performs worse, for example on VOC 2012. I think the fairer conclusion is that the framework can replace human annotations without a major loss in quality, rather than that it clearly improves over human annotations.
>
> We largely **agree with the interpretation written in the review**. Our intended conclusion from Table 1 is that model-generated annotations can replace human annotations with little or no degradation in explanation quality, rather than that they consistently outperform human annotations. In fact, this was the intended narrative of the discussion. For example, we explicitly state that “we do not observe any significant degradation in explanation quality when using model-annotated data to compute compositional explanations”. The statement highlighted by the reviewer specifically refers to the CUB setting, where the framework does outperform the human-annotation baseline, rather than to the overall trend across the datasets in Table 1.
>
> Nevertheless, we agree that some parts of the discussion could be interpreted more strongly than intended. **To improve clarity, we revised the text in several places**. In particular, we changed the discussion of Table 1 from “comparable or better” to “comparable, with the exception of VOC”, and revised the conclusion to better distinguish between settings where human annotations are available and settings where they are not. We hope these revisions better communicate the intended message. If there are additional passages that the reviewer feels remain too strongly phrased, we would be happy to further revise the wording.
>
> > The user study is useful, but some scores are hard to interpret. For example, a generic explanation such as "bird" can receive a good precision score on CUB even though it is not very informative for a fine-grained bird classifier. This suggests that the relevance score may be more meaningful than the alignment or precision scores, and the paper should clarify which parts of the user study are most important for its conclusions.
>
> We thank the reviewer for this observation and agree that the importance of the different user-study metrics depends on the task being analyzed. **We have clarified this point in the revised manuscript by expanding the discussion of the user-study results.**
>
> Specifically, in the Place365 setting, all methods rely on concept sets that are relevant to the task. Therefore, we argue that alignment and precision are the most informative metrics, since they help identify hallucinations and concepts that are either too coarse or too specific for the dataset. Conversely, in the CUB setting, we agree that relevance is the primary metric of interest. We explicitly state in the revised text that alignment and precision are more difficult for participants to evaluate in this setting due to our user-study design, while relevance better captures whether the explanation identifies concepts that are truly useful for the task.
> We have updated the discussion to clarify which metrics are most meaningful in each setting and to avoid over-interpreting alignment and precision scores in fine-grained classification tasks.
>
> > There are also minor presentation issues, such as inconsistent equation references and a Figure 2 caption that could more clearly explain the different granularity levels.
>
> Thank you for pointing out these issues. **We fixed the references to equation 10 and expanded the caption of Figure 2 (now Figure 4)**.

---

> > ### Author Response · Authors · 2026-06-15
> > **Rebuttal Part 2**
> >
> > > the experiments do not fully show how stable the explanations are when changing the segmentation backbone or when using a simpler concept vocabulary.
> >
> > We thank the reviewer for raising this point. Regarding the complexity of the concept vocabulary, we would like to clarify that the experiments already cover a broad range of vocabulary sizes, ranging from 19 concepts to more than 800 concepts depending on the dataset. We agree that this was not emphasized in the original manuscript. To address this issue, **we explicitly report the size of each concept set in the appendix and clarify the broad range in the main text**.
> >
> > Regarding the dependence on the segmentation model, **we added new experiments to investigate and discuss it in Section A**. The first experiment (Figure 5) compares explanations generated using different segmentation models through the same categorization procedure introduced in Section 4. The results show that the differences between explanations generated by different segmentation models are comparable to the differences observed between explanations generated from human annotations and model-generated annotations. More specifically, over half of the concepts appearing in the explanations are identical, approximately 30% correspond to granularity variations or semantically related concepts, and only around 20% correspond to concepts with limited semantic overlap. Overall, these results suggest that explanations generated from different segmentation models are typically highly similar.
> >
> > To further investigate the differences, we performed an additional analysis. Let $E_A$ be the explanation generated using annotations produced by segmentation model $A$, and $E_B$ the explanation generated using annotations produced by segmentation model $B$. When $E_A$ and $E_B$ differ, we recompute the alignment score of $E_B$ using the annotations generated by model $A$ and compare it with the score of $E_A$. The purpose of this experiment is to determine whether the disagreement arises from substantial annotation errors or from the presence of multiple explanations with very similar alignment scores and small variations in the segmentation masks.
> >
> > The results (Figure 6) show a low average IoU often below or close to 0.01. This difference is small and indicates that explanations selected by different segmentation models remain highly ranked when evaluated under the annotations generated by the alternative model most of the time. Consequently, the observed disagreements are generally not caused by gross annotation errors but rather by the existence of multiple explanations with comparable alignment scores (and small variation in concepts masks by individual segmentors), a phenomenon that is expected in the presence of superposition and overlapping semantic factors [1].
> >
> > Throughout our experiments we did not observe cases in which an explanation achieved a high alignment score under one segmentation model and a substantially lower score under another. Overall, these results suggest that the framework is robust to the choice of segmentation backbone and that explanations produced by different segmentation models generally capture genuine and consistent alignments.
> >
> >
> > > Finally, some parts of the analysis would benefit from more robustness checks. A short sensitivity analysis would make these claims more convincing.
> >
> > To address this concern, **we expanded both Section 4 and Appendix E and added two additional analyses**. In the first experiment (Table 17), we evaluated the impact of the ontology-based unification procedure by comparing the disagreement statistics before and after concept unification. The results show that the overall conclusions remain largely unchanged, indicating that the analysis is not strongly dependent on the ontology-merging procedure. Instead, the role of the ontology is primarily to help characterize and interpret the source of disagreement between explanations rather than to drive the conclusions themselves.
> >
> > In the second analysis (Table 16), we provide some metrics to monitor when unifying the concept set and choosing the excluded high-level nodes. Specifically, we characterize the relation between nodes merged by the procedure when nodes are added to or removed from the exclusion list. We report these results in the appendix together with a more detailed discussion of the rationale behind the selected nodes. More generally, we would like to emphasize that the ontology-guided procedure and the categorization are intended as an analysis tool rather than an optimization procedure. Consequently, our goal is not to identify a universally optimal categorization or a single correct set of excluded nodes. The objective is to make the assumptions behind the analysis explicit while providing users with a framework for understanding the origin of explanation disagreements.

---

### Review · Reviewer_Prip · 2026-06-02

**Summary Of Contributions:**

The paper builds off of prior work aimed at explaining the preferred response features of neurons as logical expressions of base concepts, where the alignment between the two is based on the similarity between the neuron’s activation maps and and segmentation masks for the logical expression, both computed on the same “probing” dataset. The work primarily differs from past methods in that it neither relies on human-generated segmentation masks for concepts on the probing dataset (expensive), nor on a “closed vocabulary” of concepts coming from a pre-trained “closed vocabulary” segmentation model (limited, especially in terms of concept generality across different datasets and in terms of concept granularity). Empirically, the proposed method offers more flexibility than competing methods, at lower costs in terms of manual labour, and with greater performance according to quantitative and qualitative evaluation metrics.

**Audience:**

Yes

**Audience Explanation:**

I don’t have a good sense of how active this subfield of research is at the moment. The main work that the authors build on was from 2020, and since then approaches to explainability have changed significantly. As the authors state a few times, their approach differs quite fundamentally from the standard SAE approach of today’s mechanistic interpretability. To the extend that the segmentation approach to explainability is still of interest to the community today, I think that the authors’ method constitutes a meaningful improvement over prior work and that there would be some interested readers. *But I think that it is up to the authors to make and support the claim that these approaches are actually used by ML practitioners*. Are there works, since 2020 (and hopefully more recently), that use these kinds of approaches to derive useful insight? Can the authors point to real-world applications that have been tried out, and not simply hypothetical proposals? Given that these sorts of methods have been around for long enough, I think that such examples are necessary to justify the relevance of the work.

An important limitation, unmentioned by the authors as far as I can tell, is that the proposed method can only be applied to CNNs. It relies on the translational equivariance of neurons in a CNNs convolutional layers so that one can derive activation masks from them, but there is no such translational equivariance in Transformers (i.e., every neuron is different, and we cannot link them across some notion of 2D space). Given that Transformers are by far the dominant class of vision models at this point in time, I think it certainly harms the proposed method that it is specific to CNNs, and will make it far less interesting to the interpretability community.

**Broader Impact Concerns:**

I don’t think it’s necessary for this paper.

**Claims And Evidence:**

Yes

**Claims Explanation:**

Somewhat.

On the positive side, once one understands the prior methods that have been proposed in the literature, it is easy to see how and why the proposed one strictly improves on them. I think that especially in terms of quantitative metrics, the authors provide strong enough results to convince me that their method is superior.

On the negative side, I find that many of the methods described in Sections 3 and 4 are poorly explained. Actually, at a high-level, I was pretty sure that I understood the method quite well after having read everything up to Section 3, and then Section 3 introduced a great deal of notation and definitions that were difficult to follow such that I think I was *more* confused by the end of it. As for Section 4, I did not really understand what was going on and how it fit into the authors’ study.

On a more fundamental level, I think that it’s worth noting that that IoU is very low (~<10%), both for the proposed metric and for the baselines. I personally think that this raises significant concerns for this class of methods as a whole, and that it probably suggests that individual neuron responses cannot be explained using logical expressions of linguistic concepts (which seems natural, given that neural networks learn complex, messy, and distributed codes). However, given that there is an active literature around these kinds of methods and that it does seem like it’s possible to make some progress in them (such as the proposed method), it seems unfair for me to recommend rejection (even if I personally think these sorts of approaches are likely dead ends).

A claim that I think is unsupported is that the custom granularity of the proposed method is useful. This is the focus of Section 6.1 and Figure 2, but based on the figure I'm not convinced the granularity is that useful. Judging by the figure alone, it just looks like the neuron represents "hummingbird-like features", and doesn't clearly represent things at a finer level of granularity (if I simply look at the masks).

**Requested Changes:**

- Section 3 needs to be improved for clarity. I recommend re-writing it, focusing less on mathematical notation and more on clear explanations that are easy to understand on a single read (and especially concrete examples). It’s currently not even clear from Section 3 alone what a concept $C_k$actually means or why it’s a “subset” rather than just a string, even though the notion of a concept is foundational for this paper.
    - The section would also benefit from a figure illustrating the different steps; given that the methods have a fair bit of geometry/masks, I think that they can be more clearly illustrated in a figure than in math.
    - As an aside regarding Section 3, shouldn't a lot of what you're calling "sets" actually be lists, since ordering matters (e.g., you have to match the ordering of the activations and the concept masks across images in the probing dataset).
    - Another aside, I think that equation 4 should use $\mathbb{D}$ (probably a typo).
    - In equation 8, it’s probably not a good idea to reuse the index $k$ for neurons, given that it was already used to index concepts above.
    - “The shape of $a_{k,k}$ depends on the neuron type” is confusing at this point, because a neuron strictly speaking is a scalar. I understand that in the case of a CNN the translation equivariance means that a neuron is actually a 2D map, but actually up to this point in the paper it probably isn’t clear that the methods are specific to CNNs.
- Section 4 didn’t make sense to me, and I think it too needs to be thoroughly improved for clarity. I’m not sure I completely understand its purpose, and I certainly could not describe the algorithm at even a high-level after trying to read it a few times. I don’t even understand if the purpose is to come up with a final metric for "misalignment between explanations", and whether it is something like a number (as in a distance metric), or something else.
    - Also, where is the knowledge graph coming from? Is it provided?
    - What is meant by “the frequency of annotated data”?
    - Regarding “information from the KG”, I thought that the concepts had not been associated to a node in the knowledge graph yet, so it's not clear to me how the knowledge graph is being used on its own to match a node with a concept without synonyms.
    - What do you mean by "we ignore the negative side of explanations"? What does "ignore" mean in this context? To account for all possible logical equivalences, one certainly can't ignore negation (e.g., modus ponens).
    - Regarding “we consider an ancestor found only if it is not one of the highest-level nodes in the tree”, it seems quite arbitrary and difficult to have a cutoff for "highest-level nodes". I think that this needs to be made more precise and justified.
    - You mention that the algorithm is iterative, but it isn’t clear to me how.
- Are there works, since 2020 (and hopefully more recently), that use these kinds of approaches to derive useful insight? Can the authors point to real-world applications that have been tried out, and not simply hypothetical proposals? Given that these sorts of methods have been around for long enough, I think that such examples are necessary to justify the relevance of the work, but currently no case is made.
- It should be clearly and obviously highlighted that the proposed method is specific to CNNs, and cannot easily be generalized to Transformers (unless the authors believe I’m wrong on this point).
- In Section 3, there are multiple activation ranges following the K-Means step, but the reset of Section 3 just discusses a single binary activations mask. It only becomes clear later in the experiments that one can construct the activation masks from multiple ranges (not just the highest one), and I think that this can be clarified from the start in Section 3.
- “For our framework, we identify a multi-granularity concept set obtained through refinements and task-specific information”. More needs to be said here in the main text; we need to have at least some intuition for where your multi-granularity concept is coming from.
- “Given a randomly sampled set of activation masks […] we define a concept as aligned […]”. Why "a concept"? Don't you mean "a compositional explanation", specifically the one that was discovered as the best description of the neuron's activation mask?
- “However, it fails to provide the appropriate level of granularity in ADE20K”. I don’t see ADE20K in Table 3, which I think is the cited results table for this segment. Am I missing something, or is this a mistake?
- Are the activation distribution really clustered and multi-modal? If so, did you use an automated process to identify the correct $k$ for K-Means? Otherwise, it seems like the purpose of the K-Means is just to quantize the activations into $k$ ranges, and you may as well just use the quantiles from the empirical distribution directly instead of using K-Means. I understand that this is what was used in some prior work, but I’m not sure I understand its purpose. I think that it needs more explanation/justification in the main text.

---

> ### Author Response · Authors · 2026-06-15
> **Part 1 (Requested Changes)**
>
> We thank the reviewer for the time spent reading the paper and providing useful feedback. We implemented the requested changes in the new revision. Below and in the next replies, the reviewer can find the explanation and points to each of the requested changes. In the last reply, the reviewer can find additional clarification for other comments in the review beyond the requested changes.
>
> > “we define a concept as aligned […]”. Why "a concept"? Don't you mean "a compositional explanation"?
>
> The term “concept” is used intentionally here and refers to the individual concepts appearing within a compositional explanation, not to the full logical formula itself. Specifically, quoting from the paper, participants were asked to evaluate “how many concepts in the compositional explanations generated by each method were aligned, precise, and relevant”. For example, if the explanation is “A OR B OR C”, users evaluate whether each individual concept (A, B, C) satisfies the corresponding property. Therefore, the sentence “a concept is aligned if ...” refers to the evaluation of each constituent concept within the compositional explanation. **We slightly revised the sentence (page 10) to make it more explicit** in the following way:  100 participants were asked to rate, on a scale from 1 (none) to 5 (all), how many **individual** concepts in  **each** of the compositional explanations generated by each method were aligned, precise, and relevant, as defined below . We hope the new wording removes any ambiguity.
>
> > For our framework, we identify a multi-granularity concept set obtained through refinements and task-specific information”. More needs to be said here in the main text; we need to have at least some intuition for where your multi-granularity concept is coming from.
>
> We agree that moving additional intuition regarding the multi-granularity concept sets to the main text would improve readability. Therefore, **we added a brief summary of this process and the structure of the concept set** to the main text and improved the references to the appendix discussion. The reviewer can find the new information in the last paragraph of Page 10.
>
> > I don’t see ADE20K in Table 3, which I think is the cited results table for this segment. Am I missing something, or is this a mistake?
>
> Thank you for pointing this out. This was a typo in the discussion of Table 3. **We corrected the sentence** to properly refer to the Places365 setting rather than ADE20K.
>
> > Are the activation distribution really clustered and multi-modal? If so, did you use an automated process to identify the correct k  for K-Means? O you may as well just use the quantiles from the empirical distribution directly instead of using K-Means.
>
> The purpose of the K-Means step is to identify distinct activation ranges that may correspond to different semantic behaviors of the neuron, as previously observed in the compositional explanation literature [2]. In this work, we follow the same setup (i.e., activations and probed models) and hyperparameters (e.g., k in k-means) adopted in prior work to ensure comparability. Ablation studies on the k are reported in prior work. Our framework itself is agnostic to the specific strategy used to define activation ranges and independent of whether these ranges are obtained through clustering or quantiles. **We clarified this motivation and this property in the main text (Section 3, Page 5**, immediately after K-means) to address the reviewer’s concern.
> Additionally, as noted by the reviewer, alternative approaches such as quantile-based thresholds are possible. In fact, the new transformer experiments (Section A.2) **added in the revision use quantile-based ranges** and produce results consistent with the conclusions of the paper.
>
> > In Section 3, there are multiple activation ranges following the K-Means step, but the reset of Section 3 just discusses a single binary activations mask. [..] I think that this can be clarified from the start in Section 3.
>
> To clarify this point, **we revised the first paragraph of Section 3** to explicitly state that compositional explanations can be computed either using quantiles or clusters. We **also** clarified **in the paragraph "alignment computation” (Section 3)** that explanations are computed independently for different activation ranges, each producing a corresponding binary activation mask.

---

> ### Author Response · Authors · 2026-06-15
> **Part 2 (Requested Changes)**
>
> > It should be clearly and obviously highlighted that the proposed method is specific to CNNs, and cannot easily be generalized to Transformers (unless the authors believe I’m wrong on this point).
>
> To address the reviewer’s concern, **we added a dedicated discussion in the limitations section (Section B)** clarifying the applicability conditions, limitations, and open challenges inherited from the broader family of compositional explanations. **We also added experiments on transformer-based architectures in the appendix (Section A.2)**.
>
> More generally, we would like to clarify that the challenge highlighted by the reviewer is not introduced by our framework, but is instead inherited from the compositional explanation paradigm itself, which our work extends without modifying its underlying alignment formulation. As discussed in the revised limitations section and in Section 3, compositional explanations (and therefore our framework) require a mechanism that projects activations into the annotation space so that alignment can be measured. From this perspective, the framework is not restricted to CNNs. In principle, it can be applied whenever activations and annotations can be represented in a common space. Indeed, prior work has already explored compositional explanations beyond convolutional architectures, including NLP settings with LSTMs [3] and reinforcement learning settings in actor-critic networks [4].
>
> Clearly, applicability does not guarantee that a meaningful compositional explanation setting exists. Many activations may not encode the spatial semantics captured by compositional explanations. The semantics, scale, and interpretation of these representations (e.g., attention maps) differ substantially from the established settings in which compositional explanations have been studied. Consequently, the main challenge is not whether the framework can technically be applied, but rather how to define appropriate projection mechanisms, representations, and evaluation settings that preserve semantic alignment (if any).
> For this reason, we added an applicability discussion in the limitations section. We believe that identifying optimal projection mechanisms and semantic interpretations for other settings is an important research direction, but it is orthogonal to the contribution of this paper and falls outside its scope. Our goal is not to establish the universal applicability of compositional explanations across all possible combinations of annotation-activation spaces, but rather to study how the proposed framework modifies the annotation side of this family of methods within well-established compositional explanation settings.
>
> Finally, to further address the reviewer’s concern, we added experiments on transformer-based architectures in the appendix (Section A.2). The results are consistent with the conclusions observed in the CNN settings.
>
> > Are there works, since 2020 (and hopefully more recently), that use these kinds of approaches to derive useful insight? I think that such examples are necessary to justify the relevance of the work, but currently no case is made.
>
>
> While compositional explanations remain a niche, several works since the original proposal have used them to study properties of learned representations. For example, prior work has used compositional explanations to analyze neuron polysemanticity, investigate how neurons encode different concepts at different activation ranges, study the extent to which convolutional neural networks exploit relative spatial relationships between objects, characterize whether neurons encode specialized or abstract concepts, and explore the relationship between representation interpretability and model performance. This line of work has also inspired several successive approaches in neuron explanations building on the findings and observations emerging from this area of research (see the literature building on [1] and [3]). To make this clearer, **we expanded the Related Work section and explicitly discussed these applications together with the corresponding references**.
>
> We agree that the number of downstream applications is smaller than for more established interpretability techniques. **We view this as a consequence of the limitations of existing compositional explanation methods, particularly their dependence on densely annotated concept datasets**. One of the motivations of our work is exactly to reduce this barrier and allow the application of compositional explanations in settings where such annotations are unavailable. We hope that increasing the applicability and flexibility of these explanations will facilitate their adoption in a broader range of future analyses.

---

> ### Author Response · Authors · 2026-06-15
> **Part 3 (Requested Changes)**
>
> > Section 3 needs to be improved for clarity. [..]
>
> We thank the reviewer for the detailed suggestions regarding the clarity of Section 3. **We implemented the proposed improvements in the revised version of the paper**.
> Specifically, we added a new figure illustrating the full pipeline and the different processing steps of the framework. We added a concrete example in the concept-set identification stage to better clarify the meaning of concept subsets and their role within the framework. We corrected the typo in Equation 4, changed the neuron indexing notation to avoid reusing the same index, and introduced additional notation clarifications to distinguish between unordered sets and structures where sample correspondence must be preserved across the probing dataset. Finally, we introduced the clarifications about neurons and activation shapes at the beginning of Section 3, as suggested by the reviewer.
>
> Regarding the comment about neurons and activation shapes, we agree that the previous wording could be confusing. We revised this discussion to clarify that the framework operates on activation representations associated with neurons, whose dimensionality depends on the architecture and representation being analyzed. In the vision settings explored in this paper, we follow the established compositional explanation literature and focus on bidimensional spatial activations extracted from convolutional representations. Finally, to further improve readability, we added examples at every step throughout Section 3.
>
> > Section 4 didn’t make sense to me, and I think it too needs to be thoroughly improved for clarity. I’m not sure I completely understand its purpose, and I certainly could not describe the algorithm at even a high-level after trying to read it a few times. I don’t even understand if the purpose is to come up with a final metric for "misalignment between explanations", and whether it is something like a number (as in a distance metric), or something else. Also, where is the knowledge graph coming from? Is it provided? What is meant by “the frequency of annotated data”? Regarding “information from the KG”, I thought that the concepts had not been associated to a node in the knowledge graph yet, so it's not clear to me how the knowledge graph is being used on its own to match a node with a concept without synonyms. What do you mean by "we ignore the negative side of explanations"? What does "ignore" mean in this context? To account for all possible logical equivalences, one certainly can't ignore negation (e.g., modus ponens). You mention that the algorithm is iterative, but it isn’t clear to me how.
>
> We thank the reviewer for the detailed feedback. We agree that the original version of Section 4 did not sufficiently explain the purpose of the proposed process, the role of the knowledge graph, and the meaning of its outputs. To address these concerns, **we substantially revised Section 4 (and added further info in section E)**.
>
> Specifically, we added a figure illustrating the process (Figure 2), a high-level description of the process and explicitly stated its objective, namely to identify whether two explanations differ and to characterize the origin of such differences (e.g., whether they stem from granularity mismatches). We also expanded the discussion of the semantic knowledge graph, clarifying its structure, assumptions, and how concepts are associated with nodes.
>
> Regarding the individual steps, we expanded most of the descriptions, clarified how concepts without synonyms are handled, specified the output of each step, and made the iterative nature of the procedure explicit. We also added a new paragraph describing the characterization of misalignment and moved part of the discussion previously reported in the analysis section to this paragraph. This addition clarifies how the process is used to distinguish disagreements caused by differences in granularity from other forms of disagreement and how the remaining misalignments are categorized.
>
> We believe these changes improve the clarity of the section and address the points raised by the reviewer.

---

> ### Author Response · Authors · 2026-06-15
> **Part 4 (Requested Changes) and References**
>
> > Regarding “we consider an ancestor found only if it is not one of the highest-level nodes in the tree”, it seems quite arbitrary and difficult to have a cutoff for "highest-level nodes". I think that this needs to be made more precise and justified.
>
> Since this procedure is intended as an analysis tool rather than an optimization procedure, our goal and intention is not to identify a universally optimal cutoff, but rather to make the assumptions behind the analysis explicit and interpretable. Specifically, the purpose of this step is to identify meaningful granularity differences between explanations. Because every pair of concepts in a semantic hierarchy shares a common ancestor, using all ancestors would merge a large number of concepts that are semantically unrelated for the analysis at hand. Therefore, some form of filtering is necessary.
>
> We do not believe that there exists a universally correct cutoff. The notion of a meaningful ancestor depends on the ontology being used, the task under analysis, and the objective of the study. For example, an ancestor that is informative in a fine-grained biological taxonomy may be too coarse for scene understanding, while the opposite may hold in other domains. To make this point more precise, **we revised the Section 4 and the Appendix to explicitly state that the excluded nodes are ontology- and task-dependent. We also added a discussion in the appendix describing the rationale used in our experiments.** Specifically, we distinguish between highly abstract nodes near the root of the hierarchy, visually meaningful but excessively general concepts, and task-specific concepts that, although valid ancestors, would merge distinctions that are important for the downstream analysis. We additionally report the exact nodes excluded in the Places365 experiments together with representative examples motivating each choice. **We also provided some metrics to monitor during the process**, clarifying that those should be looked at just as a reference of major problems rather than an optimization measure, and that the analysis should drive the process.
>
> References used in the previous replies:
>
> [1] David Bau, Jun-Yan Zhu, Hendrik Strobelt, Agata Lapedriza, Bolei Zhou, and Antonio Torralba. Understanding the role of individual units in a deep neural network. Proceedings of the National Academy of Sciences (2020).
>
> [2] Biagio La Rosa, Leilani H. Gilpin, and Roberto Capobianco. 2023. Towards a fuller understanding of neurons with clustered compositional explanations. In Proceedings of the 37th International Conference on Neural Information Processing Systems (NIPS '23). Curran Associates Inc., Red Hook, NY, USA, Article 3082, 70333–70354.
>
> [3] Jesse Mu and Jacob Andreas. 2020. Compositional explanations of neurons. In Proceedings of the 34th International Conference on Neural Information Processing Systems (NIPS '20). Curran Associates Inc., Red Hook, NY, USA, Article 1439, 17153–17163.
>
> [4] Jiang, Zeyu, Hai Huang, and Xingquan Zuo. "Compositional Concept-Based Neuron-Level Interpretability for Deep Reinforcement Learning." arXiv preprint arXiv:2502.00684 (2025).

---

> ### Author Response · Authors · 2026-06-15
> **Further clarifications**
>
> Below, some further clarifications on other points mentioned in the rebuttal but not related to the requested changes.
>
> >A claim that I think is unsupported is that the custom granularity of the proposed method is useful. This is the focus of Section 6.1 and Figure 2, but based on the figure I'm not convinced the granularity is that useful. Judging by the figure alone, it just looks like the neuron represents "hummingbird-like features", and doesn't clearly represent things at a finer level of granularity (if I simply look at the masks).
>
> > On a more fundamental level, I think that it’s worth noting that that IoU is very low (~<10%), both for the proposed metric and for the baselines.
>
> We thank the reviewer for this observation. We realize that the purpose of the granularity analysis may not have been sufficiently clear from the original text. We do not claim that a particular granularity is better than another. The purpose of Section 6.1 is instead to illustrate that analyzing different granularities separately can provide **additional** information about the semantic content encoded by a neuron.
>
> In the example shown in Figure 2, the aggregated explanation suggests that the neuron is aligned with hummingbird-related features, as the reviewer suggests. However, the fine-grained analysis reveals that the neuron is also similarly aligned with specific bird parts (the IoU score associated with the explanation). Rather than contradicting the aggregated explanation, these observations complement it by providing a more detailed view of the semantic factors associated with the activations. This behavior is consistent with the phenomenon of superposition, where multiple semantic features are encoded within the same neuron and activation range. In this case, some activations appear to be aligned with hummingbird-specific characteristics, while others exhibit comparable alignment with particular bird parts. It also compensates for the limited length of the explanation.
>
> This observation explains and it is related also to the observation about the IoU values. Neuron-level alignment scores in Network Dissection [1] and related work, including compositional explanations, operate in a very different way than standard metrics, since the neuron context is non-traditional. In this literature, IoU values above 0.04 are considered evidence of meaningful semantic alignment [1], and scores typically lie in the range of 0.00-0.20. The reason is that neurons are not expected to behave as perfect detectors that activate if and only if a single concept is present. Instead, activations often parse and encode multiple overlapping or unrelated semantic factors, resulting in partial rather than perfect spatial alignment. Indeed, previous work has shown that excessively high alignment scores can themselves be indicative of issues in the interpretation of neuron behavior [2]. Consequently, we do not interpret the values reported in Figure 2 as evidence of weak explanations. Rather, they are consistent with the alignment levels commonly observed in the compositional explanation [3] and Network Dissection literature [1].

---

> > ### Comment · Reviewer_Prip · 2026-06-15
> >
> > So far I've read the rebuttals, which seem to meaningfully engage with my feedback. This is just a note that I'll look at the updated paper in the coming days and reply accordingly.

---

> > > ### Comment · Reviewer_Prip · 2026-06-22
> > >
> > > I think that my clarity concerns in the methods sections (Sections 3 and 4) have been meaningfully addressed and that the paper is much improved from it. Thanks to the authors for taking my specific suggestions into consideration. In addition, several of my smaller technical concerns were addressed, and others are now at least acknowledged in the paper.
> > >
> > > I think that some of my larger concerns still hold.
> > >
> > > 1. The IoU of similar compositional explanation methods might indeed be similarly low, but my criticism wasn't so much about the comparison to competing methods. My concern was more broadly about whether it is even reasonable to attempt to interpret neurons in this way if it accounts for so little of what they actually respond to. Low IoUs across the benchmarks I think lends more support to this concern.
> > > 2. I'm still not convinced about the utility of the "multiple levels of granularity" in the authors' method. They argue that these different granularities can reveal complementary semantics, but at least from the figure I still just see the neuron as representing "hummingbird/bird-region features". It would have been nice to see a different example that gives better qualitative insight into what the neuron is doing, if the authors wanted to truly convince readers that their method shows us something we can't get from inspecting just a single level of granularity.
> > > 3. I don't agree that the method can be applied to general ViT-style Transformers, specifically because one cannot come up with a meaningful projection to the same space as the semantic segmentation map. After the first layer, a ViT-style neuron simply doesn't have any reliable spatial semantics. For different inputs it can represent different regions of the image, due to the image-wide attention. The authors seem to agree with me in certain parts of the paper when discussing limitations, but I don't completely know what to make of this given that the rebuttal argues that this is more of a concern for future work to come up with good projections to annotation-space. The problem is that for a real Transformer (the dominant vision architecture), neither the current method nor a simple modification of it can be used. I don't know how to interpret the current Transformer experiments that were added, given that I don't see why we can pick some arbitrary reshaping and projection of the activations tensor and claim that it aligns with a semantic segmentation map.
> > > 4. Regarding relevance, I argued in my original review that it would be nice to see a practical, real-world application of compositional explanations, given that they have been around for several years now. While the authors provided some interpretability papers that have used compositional explanations, I don't think this constitutes substantial evidence that they have made an impact. Combined with my concerns (1) and (3) above, I remain worried that this class of methods may not be leveraged in future work.
> > >
> > > All things considered, it is my opinion that this paper is ready for publication. In the end, despite my concerns, it makes meaningful progress in improving an established method, and the clarity has been improved substantially such that most readers will understand the method and be able to reproduce it. I think this meets the bar for acceptance.